# Hijacking of transcriptional condensates by endogenous retroviruses

Vahid Asimi[1,2,14], Abhishek Sampath Kumar[1,3,14], Henri Niskanen[1,14], Christina Riemenschneider[1,3,14], Sara Hetzel[1], Julian Naderi[1], Nina Fasching[4], Niko Popitsch[4,5], Manyu Du[6,7], Helene Kretzmer[1], Zachary D. Smith[8,9], Raha Weigert[1], Maria Walther[1], Sainath Mamde[1], David Meierhofer[10], Lars Wittler[11], René Buschow[12], Bernd Timmermann[13], Ibrahim I. Cisse[6,7], Stefan L. Ameres[4,5], Alexander Meissner[1,8,9] and Denes Hnisz[1]✉

**Most endogenous retroviruses (ERVs) in mammals are incapable of retrotransposition; therefore, why ERV derepression is associated with lethality during early development has been a mystery. Here, we report that rapid and selective degradation of the heterochromatin adapter protein TRIM28 triggers dissociation of transcriptional condensates from loci encoding super-enhancer (SE)-driven pluripotency genes and their association with transcribed ERV loci in murine embryonic stem cells. Knockdown of ERV RNAs or forced expression of SE-enriched transcription factors rescued condensate localization at SEs in TRIM28-degraded cells. In a biochemical reconstitution system, ERV RNA facilitated partitioning of RNA polymerase II and the Mediator coactivator into phase-separated droplets. In TRIM28 knockout mouse embryos, single-cell RNA-seq analysis revealed specific depletion of pluripotent lineages. We propose that coding and noncoding nascent RNAs, including those produced by retrotransposons, may facilitate 'hijacking' of transcriptional condensates in various developmental and disease contexts.**

ERVs make up around 10% of mammalian genomes, and ERVs are repressed by multiple mechanisms including heterochromatin, DNA methylation and modification of their RNA transcripts[1–13]. One of the best-studied repressive pathways involves the TRIM28 heterochromatin corepressor that is recruited by KRAB-ZFP transcription factors to ERVs in pluripotent embryonic stem cells (ESCs), where it recruits the histone H3 K9 methyltransferase SETDB1 and the heterochromatin protein HP1α that together establish a repressive chromatin environment[7,9,14–17]. ERV derepression is associated with lethality at various embryonic stages and in ESCs deficient for the TRIM28-HP1α pathway[7–10], although most ERVs in mice and humans have lost their ability to undergo retrotransposition[1–6], and deletion of entire clusters of KRAB-ZFP factors does not lead to elevated transposition rates in mice[18]. These findings suggest that RNA transcripts produced by ERVs may contribute to developmental phenotypes associated with ERV derepression.

RNA has long been recognized as a component of phase-separated biomolecular condensates, including stress granules, splicing speckles and Cajal bodies[19], and recent studies indicate that RNA may make important contributions to nuclear condensates formed by transcriptional regulatory proteins[20]. During transcription, nascent RNA is thought to promote formation of transcriptional condensates enriched in RNA polymerase II (RNAPII) and the Mediator

coactivator through electrostatic interactions that contribute to phase separation[21]. ERV RNAs can be transcribed at hundreds, if not thousands, of genomic loci, and many nuclear noncoding RNAs localize to the loci where they are produced[22]. These data lead us to hypothesize that ERV RNA transcripts may impact the genomic distribution of transcriptional condensates in cells deficient for ERV repression.

Here, we test the model that ERV RNA transcripts contribute to lethality associated with ERV derepression through disrupting the genomic distribution of transcriptional condensates. We found that RNAPII-containing condensates, typically associating with SE-driven pluripotency genes, are hijacked by transcribed ERV loci upon acute perturbation of the machinery responsible for ERV repression in ESCs. Condensate association was dependent on ERV RNAs and was rescued by ERV RNA knockdown or forced expression of pluripotency transcription factors. The results highlight an important role of ERV RNA transcripts in nuclear condensates in pluripotent cells.

## Results

**Rapid and selective degradation of TRIM28 in mESCs.** ERVs, including intracisternal A-type particles (IAPs), are bound by members of the TRIM28-HP1α pathway and marked by H3K9me3 in murine ESCs (mESCs)[7,9,14–17] (Fig. 1a,b and Supplementary

[1]Department of Genome Regulation, Max Planck Institute for Molecular Genetics, Berlin, Germany. [2]Institute of Chemistry and Biochemistry, Freie Universität Berlin, Berlin, Germany. [3]Institute of Biotechnology, Technische Universität Berlin, Berlin, Germany. [4]Institute of Molecular Biotechnology (IMBA), Vienna BioCenter (VBC), Vienna, Austria. [5]Department of Biochemistry and Cell Biology, Max Perutz Labs, University of Vienna, Vienna BioCenter (VBC), Vienna, Austria. [6]Department of Physics, Massachusetts Institute of Technology (MIT), Cambridge, MA, USA. [7]Department of Biological Physics, Max Planck Institute of Immunobiology and Epigenetics, Freiburg, Germany. [8]Broad Institute of MIT and Harvard, Cambridge, MA, USA. [9]Department of Stem Cell and Regenerative Biology, Harvard University, Cambridge, MA, USA. [10]Max Planck Institute for Molecular Genetics, Mass Spectrometry Facility, Berlin, Germany. [11]Department of Developmental Genetics, Max Planck Institute for Molecular Genetics, Berlin, Germany. [12]Microscopy Core Facility, Max Planck Institute for Molecular Genetics, Berlin, Germany. [13]Sequencing Core Facility, Max Planck Institute for Molecular Genetics, Berlin, Germany. [14]These authors contributed equally: Vahid Asimi, Abhishek Sampath Kumar, Henri Niskanen, Christina Riemenschneider. ✉e-mail: hnisz@molgen.mpg.de

Fig. 1a–g). In contrast, heterochromatin components tend not to occupy enhancers bound by the pluripotency transcription factors (TFs) OCT4, SOX2 and NANOG that drive the cell-type-specific transcriptional program of mESCs[7,23,24] (Fig. 1a,b and Supplementary Fig. 1h–j).

Resolving the direct consequences of ERV derepression has been impeded, in part, by limitations of classic gene disruption strategies and the essential nature of the TRIM28-HP1α pathway in mESCs[7–10]. To overcome these challenges, we generated an mESC line that encodes degradation-sensitive TRIM28-FKBP alleles using the dTAG system (Fig. 1c and Supplementary Fig. 2a,b)[25]. Directed differentiation and tetraploid aggregation assays confirmed that TRIM28-FKBP mESCs maintained pluripotency and a gene expression profile similar to parental V6.5 mESCs (Supplementary Fig. 2d–n). Endogenously tagged TRIM28 experienced reversible, ligand-dependent proteolysis with near-complete degradation after 6 h of exposure to the dTAG-13 ligand (Fig. 1d and Supplementary Fig. 2b). Quantitative mass spectrometry confirmed that TRIM28 degradation was highly selective up to 24 h of dTAG-13 treatment (Supplementary Fig. 2c). Short-term (up to 24 h) TRIM28 degradation did not substantially alter the protein levels of pluripotency markers (for example, OCT4, SOX2, SSEA-1) (Fig. 1d and Supplementary Fig. 2j-l), suggesting that acute TRIM28 degradation did not markedly alter the pluripotent state.

**Reduced SE transcription in TRIM28-degraded mESCs.** To monitor changes in transcriptional activity upon acute TRIM28 degradation, we used TT-SLAM-seq, a recently developed genome-wide nascent transcription readout[26]. TT-SLAM-seq combines metabolic labeling and chemical nucleoside conversion (SLAM-seq)[27] with selective enrichment of newly synthesized RNA (TT-seq)[28] to detect nascent RNA transcription with high temporal resolution and sensitivity (Supplementary Fig. 3a–c). Consistent with previous reports[7,29], we observed derepression of several main classes of ERVs, including IAPs, MMERVK10C and MMERVK9C elements in TRIM28-degraded ESCs (Fig. 1e and Supplementary Fig. 4a–f) and loss of H3K9me3 at these sites (Supplementary Fig. 4g). Derepression of ERVs was also confirmed with extended TRIM28 degradation for 96 h and RNA-seq (Fig. 1e and Supplementary Fig. 4a–e). The TT-SLAM-seq data revealed around 250 genes whose transcription was significantly induced and around 300 genes whose transcription was significantly reduced upon 24 h of TRIM28 degradation (greater than twofold, false discovery rate (FDR) < 0.05) (Fig. 1f,g). The downregulated genes were enriched for SE-associated pluripotency genes (NES = −1.6, $P < 10^{-3}$) (Fig. 1h). Downregulation of these genes was associated with the

reduction of nascent transcription at the SEs (Fig. 1i and Extended Data Fig. 1a–d), which tended to precede the reduction of transcription at the SE-driven gene (Fig. 1f and Extended Data Fig. 1a–c). These results were unexpected, as TRIM28 binds to ERVs in mESCs and is not bound at enhancers or SEs (Fig. 1b and Supplementary Fig. 1i,j). These data reveal the direct transcriptional response to the loss of TRIM28 and suggest that acute TRIM28 degradation leads to reduction of SE transcription in ESCs.

**Reduced SE-condensate association in TRIM28-degraded mESCs.** Components of the transcription machinery, for example, RNAPII and the Mediator coactivator, form biomolecular condensates that associate with SEs in ESCs[30–33], and the presence of RNAPII condensates at genomic sites correlates with elevated transcriptional activity[32]. We thus hypothesized that reduction of SE transcription in TRIM28-degraded ESCs may be caused by reduced association of transcriptional condensates with SE loci. To test this idea, we visualized the genomic region containing the well-studied SE at the *miR290-295* locus using nascent RNA-fluorescence in situ hybridization (FISH) and transcriptional condensates using immunofluorescence (IF) against RNAPII[31]. RNAPII puncta consistently colocalized with the *miR290-295* locus in control ESCs, and the colocalization was reduced after 24 h of TRIM28 degradation (Fig. 1j and Extended Data Fig. 2a), while the overall level of RNAPII did not change (Extended Data Fig. 2b,c). Similar results were observed at the *Fgf4* SE locus (Extended Data Fig. 2d,e). These data indicate that transcriptional condensates associate less with SE loci in TRIM28-degraded ESCs.

To further probe colocalization between RNAPII condensates and SEs, we used live-cell super-resolution photoactivated localization microscopy (PALM)[32]. We used an mESC line that encodes 24 copies of an MS2 stem-loop integrated into the SE-driven *Sox2* gene, a transgene encoding the MCP MS2-binding protein with a SNAP tag (MCP-SNAP) and Rpb1 RNAPII subunit endogenously tagged with the Dendra2 photoconvertible fluorophore. We integrated the degradation-sensitive FKBP tag into the *Trim28* locus in these cells, enabling acute TRIM28 degradation (Extended Data Fig. 2f). In this system, the MCP-SNAP protein can be used to visualize nascent RNA produced by the *Sox2* gene, and the Dendra2 tag can be used to track RNAPII clusters[32]. We then visualized RNAPII clusters for 2 min in live mESCs using PALM and measured the size and distance of the RNAPII cluster nearest to the *Sox2* locus. We found that 24 h dTAG treatment led to a significant reduction in the size of the RNAPII cluster nearest to the *Sox2* locus ($P = 2 \times 10^{-4}$, Wilcoxon–Mann–Whitney test) (Fig. 1k) and an increase in the distance between the locus and the nearest RNAPII cluster ($P = 0.04$,

**Fig. 1 | TRIM28 degradation leads to the reduction of SE transcription and loss of transcriptional condensates at SEs in mESCs. a**, Models of the TRIM28/HP1α pathway and enhancers. **b**, Heatmap of ChIP–seq read densities within a 2-kb window around full-length IAP ERVs and enhancers in mESC. The genomic elements were length normalized. Enhancers include the constituent enhancers of SEs and typical enhancers. Rpm, reads per million. **c**, Scheme of the dTAG system to degrade TRIM28 in mESCs. **d**, Western blot validation of the FKBP degron tag and its ability to degrade TRIM28. **e**, FC in read density of TT-SLAM-seq and RNA-seq data after the indicated duration of dTAG-13 treatment, normalized to the level in the DMSO control. Data are presented as mean values ± s.d. from three biological replicates. *P* values are from unpaired two-sided *t*-tests. \*\**P* < 0.01. **f**, Genome browser tracks of ChIP–seq data (H3K27Ac, OCT4, SOX2, NANOG) in control mESCs and TT-SLAM-seq data upon 0 h, 2 h, 6 h and 24 h dTAG-13 treatment at the *Klf4* locus. Chr, chromosome. **g**, FC of gene transcription (TT-SLAM-seq data) upon dTAG-13 treatment. The number of significantly deregulated genes (DESeq2) and example pluripotency genes are highlighted. **h**, Gene set enrichment analysis: genes are ranked according to their FC in transcription (TT-SLAM-seq) after 24 h of dTAG-13 treatment. SE genes are marked with black ticks. *P* denotes a nominal *P* value. **i**, Log$_2$ FC in TT-SLAM-seq read density at SEs and typical enhancers upon dTAG-13 treatment normalized to untreated control mESCs. *P* values are from two-sided Wilcoxon–Mann–Whitney tests. \*\*\*\**P* = 5 × 10$^{-8}$, \*\*\**P* = 5 × 10$^{-4}$. **j**, Representative images of individual z-slices (same z) of RNA-FISH and IF signal, and an image of the merged channels. The nuclear periphery determined by DAPI staining is highlighted as a white contour (scale bars, 2.5 μm). Also shown are averaged signals of either RNA-FISH or RNAPII IF centered on the FISH foci or randomly selected nuclear positions (scale bars, 0.5 μm). *r* denotes a Spearman's correlation coefficient. **k**, Live-cell PALM imaging of Dendra2-RNAPII and nascent RNA transcripts of *Sox2*-MS2 in mESCs after 24 h dTAG-13 treatment. Left, size of the nearest RNAPII cluster around *Sox2*; middle left, distance between the *Sox2* locus and the nearest RNAPII cluster; middle right, average RNAPII cluster size globally; right, number of RNAPII clusters per cell. Data are presented as mean values ± s.d. *P* values are from Wilcoxon–Mann–Whitney tests.

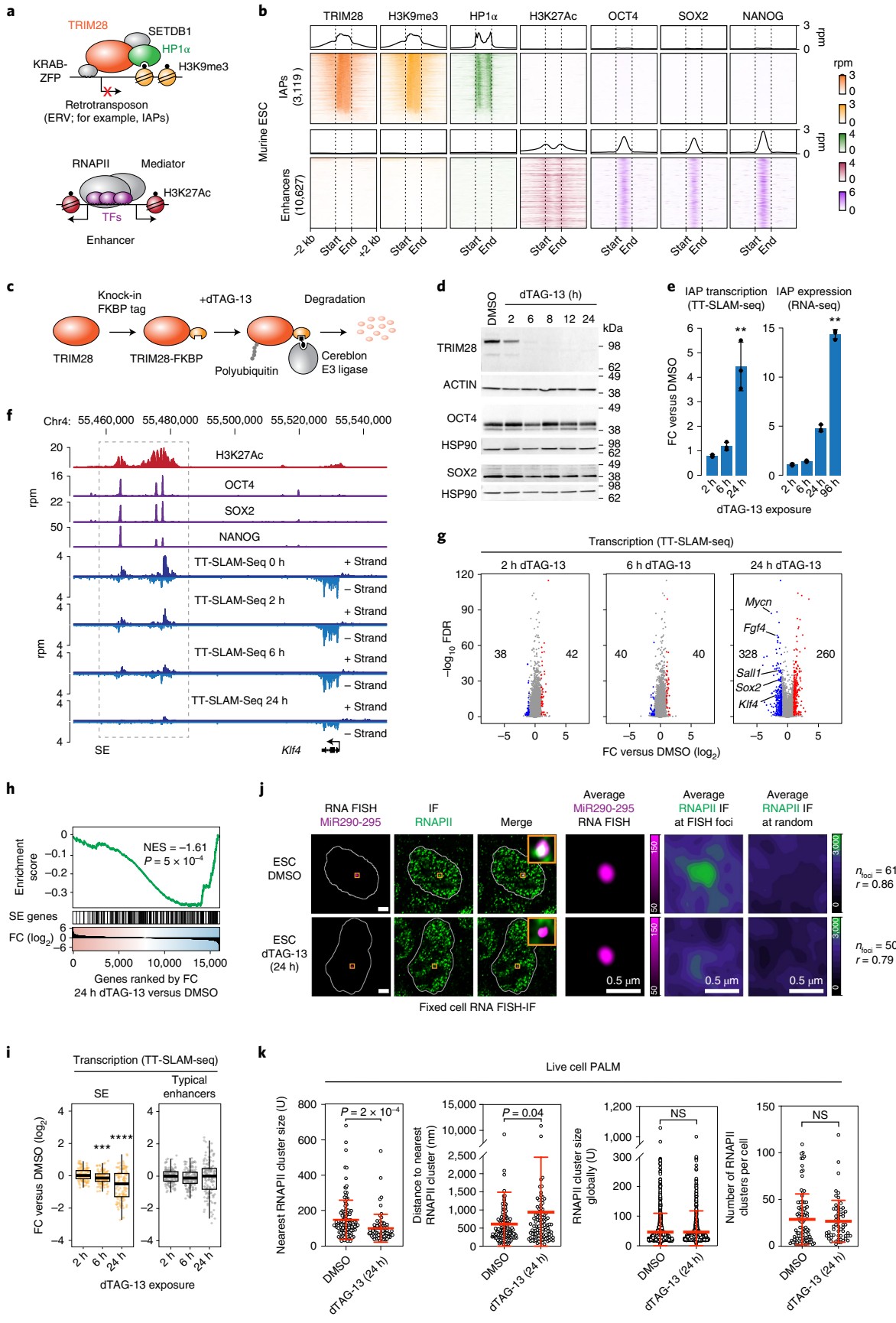

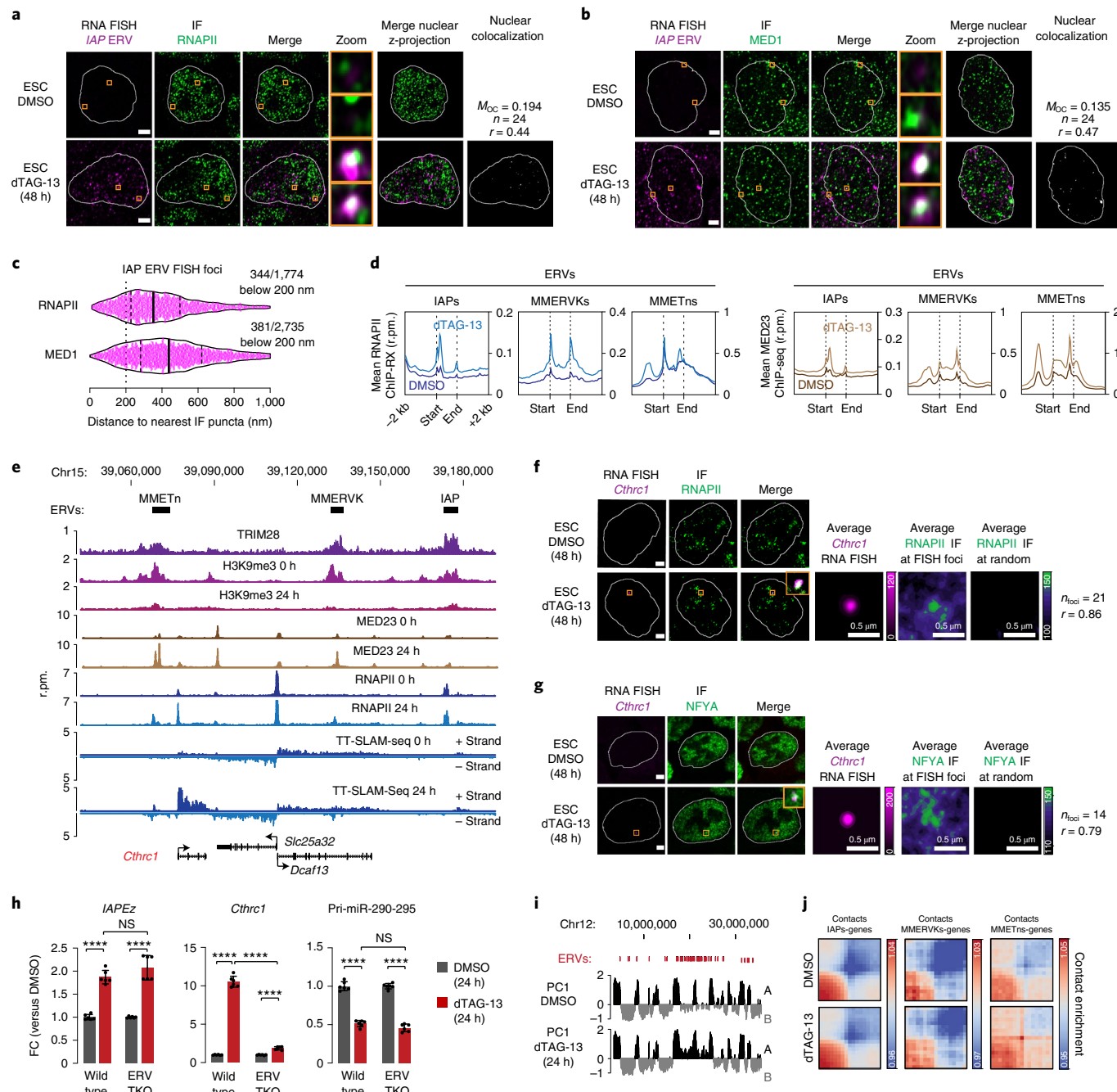

**Fig. 2 | Derepressed IAPs form nuclear foci that associate with RNAPII condensates and incorporate nearby genes. a**, Representative images of individual z-slices (same z) of RNA-FISH and RNAPII IF signal, and an image of the merged channels. The nuclear periphery determined by DAPI staining is highlighted as a white contour. The zoom column displays the region of the images highlighted in a yellow box (enlarged for greater detail). Merge of the nuclear z-projections is displayed, and overlapping pixels between the RNA-FISH and IF channels are highlighted in white. Displayed $M_{OC}$ and Pearson's correlation coefficient ($r$) values are an average obtained from 24 analyzed nuclei. Scale bars, 2.5 μm. **b**, Same as **a**, except with MED1 IF. **c**, Distance of *IAP* RNA-FISH foci to the nearest RNAPII or MED1 IF puncta. Each dot represents one *IAP* RNA-FISH focus. **d**, Meta representations of RNAPII ChIP with reference exogenous genome (ChIP-RX) (left) and MED23 ChIP–seq (right) read densities at IAP, MMERVK and MMETn ERVs in control (DMSO-) and dTAG-13 (24 h)-treated mESCs. The mean read densities are displayed ±2 kb around the indicated elements. The genomic elements were length normalized. **e**, Genome browser tracks at the *Cthrc1* locus. Note the independent transcription initiation events at *Cthrc1* and MMETn, ruling out that the MMETn acts as an alternative *Cthrc1* promoter. Rpm, reads per million. **f**, Representative images of individual z-slices (same z) of RNA-FISH and IF signal, and an image of the merged channels. The nuclear periphery determined by DAPI staining is highlighted as a white contour (scale bars, 2.5 μm). Also shown are averaged signals of either RNA-FISH or IF centered on the *Cthrc1* FISH foci or randomly selected nuclear positions (scale bars, 0.5 μm). *r* denotes a Spearman's correlation coefficient. **g,** Same as **f**, except with NFYA IF. **h**, qRT–PCR data for *IAP* RNA, *Cthrc1* mRNA and the *Pri-miR-290-295* transcript in control and ERV-triple knockout (TKO) cells. Data are presented as mean values ± s.d. from six biological replicates. *P* values are from two-tailed *t*-tests. ****$P < 1 \times 10^{-4}$. **i**, Principal component (PC) plot of Hi-C interactions at an ERV-rich locus on chromosome 12. **j**, Pile-up analysis of contacts between IAPs, MMERVKs, MMETns and transcribed genes in wild-type and TRIM28-degraded mESCs.

Wilcoxon–Mann–Whitney test) (Fig. 1k), while the global size of RNAPII clusters in the cells and the average number of RNAPII clusters per cell did not change (Fig. 1k). These data indicate reduced association of RNAPII condensates at the *Sox2* SE locus upon acute TRIM28 degradation in live cells.

**Derepressed IAP RNA foci overlap RNAPII condensates.** To investigate whether transcriptional condensates colocalize with derepressed ERVs, we visualized IAP ERV loci with RNA-FISH. Nuclear *IAP* foci became progressively apparent after 24–48h of TRIM28 degradation (Fig. 2a and Extended Data Fig. 3a), and some nuclear *IAP* foci colocalized with RNAPII puncta visualized with IF (mean Manders' overlap coefficient ($M_{OC}$), 0.193; $n = 24$ cells) (Fig. 2a and Extended Data Fig. 3a). Colocalization of *IAP* foci was similarly observed with Mediator puncta visualized with IF using antibodies against the MED1 ($M_{OC}$, 0.135; $n = 24$ cells) (Fig. 2b) and MED23 Mediator subunits (Extended Data Fig. 3b). Overall, ~20% of *IAP* foci were located within 200nm of an RNAPII or MED1 puncta, a distance range compatible with regulatory interactions[32] (Fig. 2c). Consistent with the colocalization of transcriptional condensates with *IAP* foci and their reduced colocalization with SEs, the occupancy of RNAPII, Mediator and the transcription-associated H3K27Ac chromatin mark increased at various ERV families already after 24h of TRIM28 degradation, while their enrichment was reduced at SEs (Fig. 2d and Supplementary Fig. 5a,b).

As transcriptional condensates may associate with multiple distant DNA sites, we explored the possibility that condensates associating with ERVs incorporate ERV-proximal genes. To this end, we visualized the *Cthrc1* locus using nascent RNA-FISH, as *Cthrc1* was among the top upregulated genes in the TT-SLAM-seq data after 24h of TRIM28 degradation and is located within 100kb of three ERVs (Fig. 2e). We found that the *Cthrc1* locus colocalized with RNAPII puncta in TRIM28-degraded ESCs (Fig. 2f and Extended Data Fig. 4a). The locus also colocalized with puncta formed by the NFY TF, whose motif is highly enriched in the long terminal repeat (LTR) of IAPs and other ERVs (Fig. 2g and Extended Data Fig. 4b,c), but not with puncta of a control TF NRF1 (Extended Data Fig. 4d). Transient (30 min) treatment of the cells with 1.5% 1,6 hexanediol (1-6 HD)—a short chain aliphatic alcohol that dissolves various biomolecular condensates including RNAPII condensates[32] (Extended Data Fig. 3c,d)— reduced the level of *Cthrc1* nascent RNA (twofold, $P < 0.05$, $t$-test) in TRIM28-degraded cells, indicating that RNAPII condensates contribute to the upregulation of this gene (Extended Data Fig. 3e). We then used CRISPR–Cas9 to delete the three ERVs at the *Cthrc1* locus and found that, in the absence of the three ERVs, induction of *Cthrc1* and other genes in the locus was compromised upon TRIM28 degradation (Fig. 2h and Supplementary Fig. 6a–e). To further probe contacts between derepressed ERVs and genes, we performed in situ Hi-C in control and TRIM28-degraded ESCs. We found that 24h of TRIM28 degradation did not lead to marked genome-wide changes in chromatin contacts (Supplementary Figs. 7a,b and 8a,b) but did lead to a shift of the most-induced ERV taxa from the inactive 'B' towards the active 'A' compartment (Fig. 2i and Supplementary Fig. 7c) and a moderate increase in the contact frequency of ERVs with transcribed genes and SEs (Fig. 2j and Supplementary Fig. 7d). These results demonstrate that transcriptional condensates may incorporate genes proximal to derepressed ERVs.

**SE-enriched TFs rescue condensate localization.** RNAPII- and Mediator-containing condensates are thought to be anchored at SEs by TFs that are enriched at these sites[34]. One would thus expect that overexpression of SE-enriched TFs rescues the reduced association of transcriptional condensates with SEs in TRIM28-degraded cells. To test this idea, we generated degradation-sensitive TRIM28-FKBP alleles in an induced pluripotent stem cell (iPSC) line that contains

integrated transgenes encoding OCT4, SOX2, KLF4 and MYC under a doxycycline-inducible promoter (Fig. 3a,b and Extended Data Fig. 5a,b)[35]. The OCT4, SOX2 and KLF4 TFs are highly enriched at SEs in ESCs[36]. TRIM28 degradation in the iPSCs led to the appearance of IAP foci as revealed by IAP RNA-FISH (Fig. 3c,d). Overexpression of OCT4, SOX2, KLF4 and MYC substantially reduced the fraction of iPSCs containing *IAP* foci (Fig. 3c,d) and overall *IAP* RNA level in the cell population (Fig. 3e). Furthermore, OCT4, SOX2, KLF4 and MYC overexpression rescued the extent of colocalization of RNAPII puncta with the *miR290-295* SE locus in TRIM28-degraded cells (Fig. 3f) while the overall levels of RNAPII subunits did not change (Extended Data Fig. 5c,d). OCT4, SOX2, KLF4 and MYC overexpression also partially rescued the downregulation of the miR290-295 SE RNA and Pri-miR290-295 transcript in TRIM28-degraded cells (Fig. 3g) and nascent transcript levels at the *Klf4*, *Fgf4*, *Oct4* and *Mycn* SE loci (Extended Data Fig. 5e–h). These results suggest that forced expression of SE-binding TFs prevents the loss of transcriptional condensates at the *miR290-295* SE locus and attenuates *IAP* induction in TRIM28-degraded cells.

**Roles of ERV RNA in condensate formation and localization.** RNA is a key component of numerous biomolecular condensates[19] and nascent RNA can enhance phase separation of transcriptional regulatory proteins[21]. Therefore, we hypothesized that RNA produced at ERV loci may contribute to the genomic localization of RNAPII-containing condensates. To test this idea, we knocked down various ERV RNAs in TRIM28-degraded cells. Expression of shRNAs targeting the four most prominent ERV families (IAPs, MMERVK10Cs, MMERVK9Cs, MMETns) partially rescued the downregulation of SEs and their associated genes after 24h of TRIM28 degradation while knocking down IAPs alone did not (Extended Data Fig. 6a–d). However, expression of the shRNAs for 24h before inducing TRIM28 degradation (for 24h) almost entirely rescued the upregulation of ERV transcript levels (Fig. 4a,b and Extended Data Fig. 6e,g), the appearance of IAP RNA-FISH foci (Extended Data Fig. 6h), reduced transcription at SEs and their associated genes (Fig. 4b,c and Extended Data Fig. 6f) and the reduced association of RNAPII condensates at the *mir290-295* SE locus (Fig. 4d and Extended Data Fig. 6i–j). These results indicate that knockdown of ERV RNAs rescues the decrease of SE transcription and reduced condensate localization at SEs in TRIM28-degraded cells.

To further dissect the relationship between ERV RNA and transcriptional condensates, we performed in vitro reconstitution experiments. We purified recombinant, mCherry-tagged C-terminal domain (CTD) of RNAPII, which was shown previously to form condensates in vitro[30,33], and mixed it with fluorescein-labeled in vitro transcribed IAP RNA fragments. The IAP RNA fragments facilitated RNAPII CTD droplet formation in a dose-dependent manner (Fig. 4e,f and Extended Data Fig. 7a), and the IAP RNA was enriched within RNAPII CTD droplets in a dose-dependent manner (Fig. 4e,g). IAP RNA also facilitated condensation of the intrinsically disordered region (IDR) of the MED1 Mediator subunit (Extended Data Fig. 7b–e)—a frequently used in vitro model of Mediator[31,33,34]. Furthermore, IAP RNA enhanced droplet formation of purified recombinant HP1α (an in vitro model of heterochromatin[37]), but the optimal concentration of the RNA for HP1α was about fivefold lower than that for MED1 IDR in this in vitro system (Fig. 4h and Extended Data Fig. 8a,b). As expected, various other RNAs for example SE RNA[21] and RNA from main satellite repeats[38] also enhanced droplet formation of MED1 IDR and HP1α in vitro, but the difference in the optimal RNA concentration stayed consistently about fivefold (Extended Data Fig. 8a,b). Moreover, IAP RNA fragments facilitated partitioning of both the MED1 IDR and NFYC-IDR into IAP-RNA-containing heterotypic droplets (Fig. 4i,j and Extended Data Fig. 7f–i). These results indicate that IAP RNA can enhance droplet formation of key transcriptional regulatory

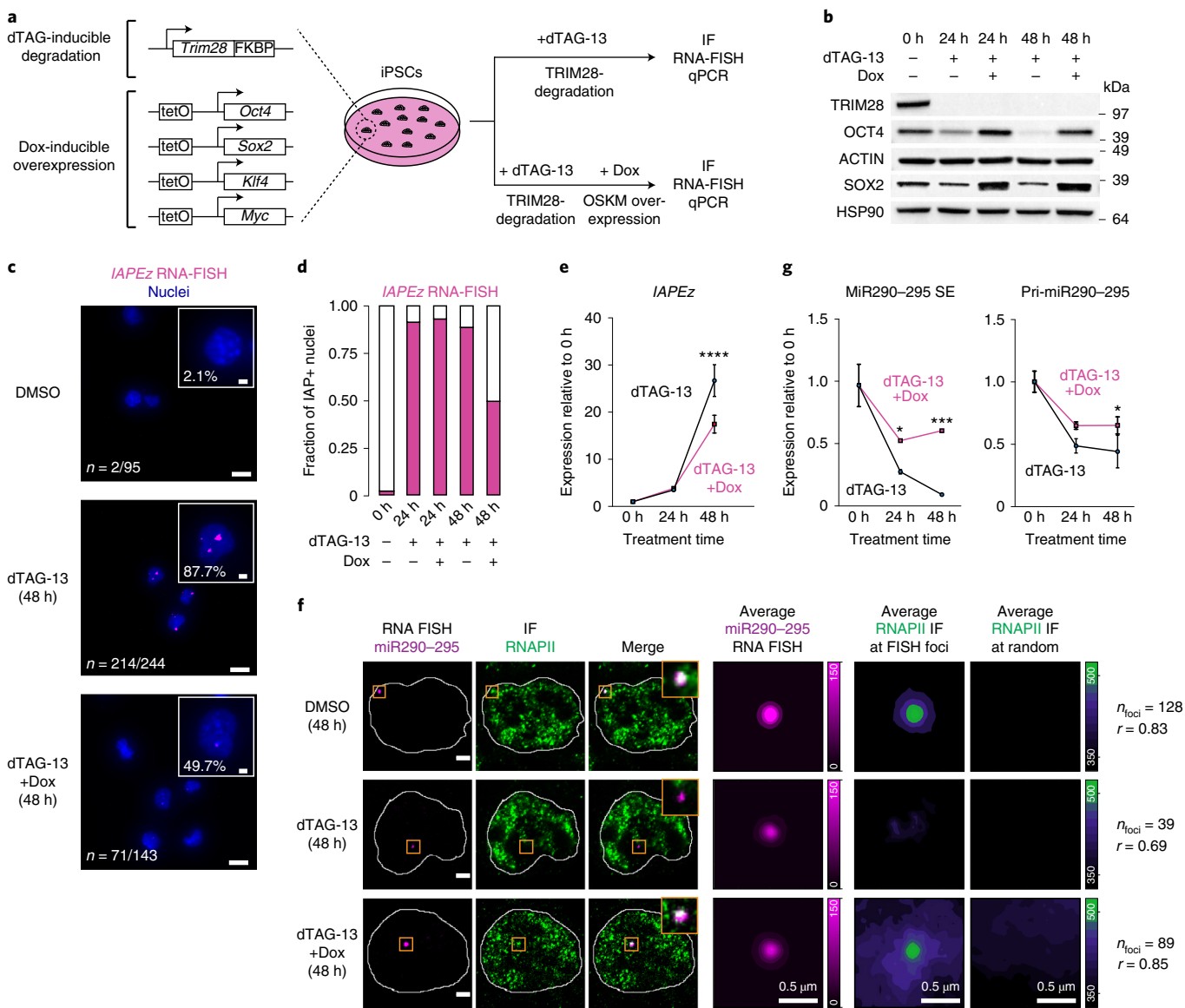

**Fig. 3 | SE-enriched TFs rescue condensate localization in TRIM28-degraded mESCs. a**, Genotype of the iPSC line and scheme of the experimental setup. The iPSC line contains degradation-sensitive *Trim28-FKBP* alleles and doxycycline-inducible *Oct4*, *Sox2*, *Klf4* and *c-Myc* (OSKM) transgenes. **b**, Western blot validation of the FKBP degron tag and OSKM ectopic expression in iPSCs. **c**, Representative images of IAP RNA-FISH staining. The number and percentage refer to cells with detectable *IAP* foci, pooled from two biological replicates. Scale bars, 10 μm; inset scale bars, 2 μm. **d**. Quantification of cells with detectable *IAP* foci (IAP⁺ cells) at the indicated treatment regimes. **e**, IAP RNA expression is reduced in TRIM28-degraded iPSCs that ectopically express OSKM factors. The line plot shows qRT–PCR data of IAP RNA levels normalized to 0 h of dTAG-13 treatment. Data are from three independent biological replicates (three wells on a tissue culture plate) and are presented as mean values ± s.d. The experiment was repeated three times, and data from one representative experiment are shown. *P* value is from two-tailed *t*-tests. ***$P < 1 \times 10^{-4}$. **f**, Colocalization between the nascent RNA of *miR290-295* and RNAPII puncta in TRIM28-degraded iPSCs that ectopically express OSKM factors. Separate images of individual z-slices (same z) of the RNA-FISH and IF signal are shown along with an image of the merged channels. The nuclear periphery determined by DAPI staining is highlighted as a white contour (scale bars, 2.5 μm). Also shown are averaged signals of either RNA-FISH or RNAPII IF centered on the miR290-295 RNA FISH foci or randomly selected nuclear positions. *r* denotes a Spearman's correlation coefficient (scale bars, 0.5 μm). **g**, Elevated levels of miR290-295 SE transcript and Pri-miR290-295 nascent transcript in TRIM28-degraded iPSCs that ectopically express OSKM factors. qRT–PCR data was normalized to the 0 h of dTAG-13 treatment. Data are from three independent biological replicates (three wells on a tissue culture plate) and are presented as mean values ± s.d. The experiment was repeated three times, and data from one representative experiment are shown. *P* values are from two-tailed *t*-tests. *$P = 0.027$, ***$P = 1 \times 10^{-4}$.

proteins, and suggest a mechanistic basis for the difference of the effect of RNA on heterochromatin and transcriptional condensates.

**Transgenic ERVs compete with SEs for activators.** Derepressed ERV loci seem to compete for transcriptional condensates with SEs, in part through producing RNA that facilitates condensation

of transcriptional activators. To probe this competition directly, we investigated whether simultaneously activated transcription at repetitive loci (for example, ERVs) could compromise transcription at SEs. First, we attempted to activate IAPs using CRISPRa[39], but targeting a dCas9-VP64 protein to IAPs with several guide RNAs failed to produce meaningful transcription at those elements. We then

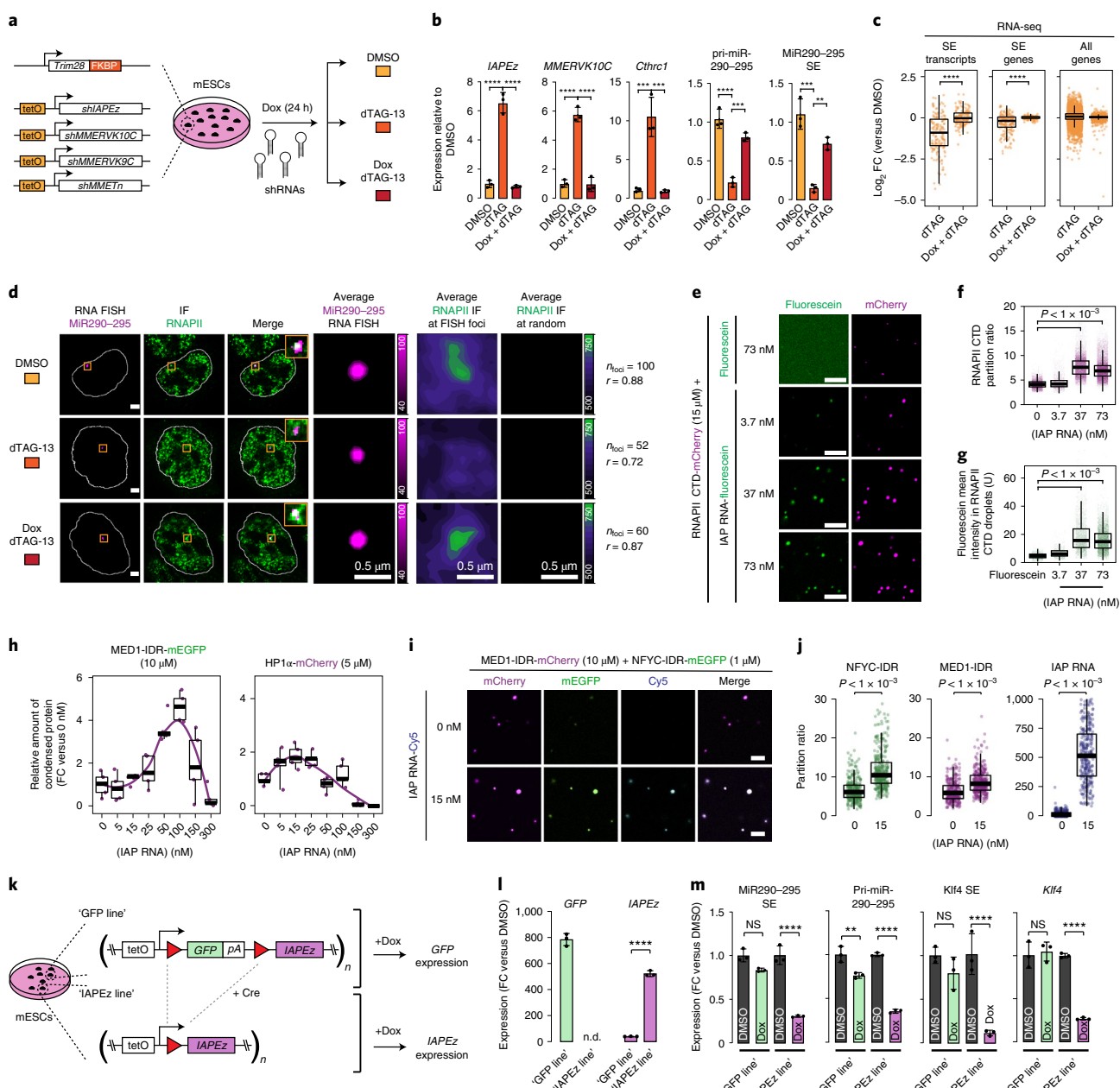

**Fig. 4 | Contributions of IAP RNA to condensate localization in vivo and condensate formation in vitro. a**, Schematic model of the ERV shRNA knockdown experiments. **b**, qRT–PCR data as FC-normalized to the DMSO treatment control. Data are presented as mean values ± s.d. from three biological replicates. *P* values are from two-tailed *t*-tests. ****$P < 1 \times 10^{-4}$, ***$P < 1 \times 10^{-3}$, **$P < 1 \times 10^{-2}$. **c**, Log$_2$ FC values in total RNA-seq data at intergenic SEs and genes. Data are from three biological replicates. *P* values are from two-sided Wilcoxon–Mann–Whitney tests. ****$P < 1 \times 10^{-4}$. **d**, Representative images of individual z-slices (same z) of RNA-FISH and IF signal, and an image of the merged channels. The nuclear periphery determined by DAPI staining is highlighted as a white contour (scale bars, 2.5 μm). Also shown are averaged signals of either RNA-FISH or RNAPII IF centered on the *miR290-295* FISH foci or randomly selected nuclear positions (scale bars, 0.5 μm). *r* denotes a Spearman's correlation coefficient. **e**, Representative images of mixtures of fluorescein-labeled *IAP* RNA and purified recombinant RNAPII CTD-mCherry in droplet formation buffer. Scale bar, 5 μm. **f**, Partitioning ratio of RNAPII CTD-mCherry into droplets at the indicated IAP RNA concentrations. Every dot represents a detected droplet. *P* values are from two-sided *t*-tests. **g**, Quantification of the enrichment of fluorescein-labeled IAP RNA in RNAPII CTD-mCherry droplets. *P* values are from two-sided *t*-tests. **h**, Quantification of the partitioning of (left) MED1 IDR and (right) HPIα into droplets in the presence of IAP RNA. Values are normalized against the partition ratio at no RNA added. Corresponding images are found in Extended Data Fig. 8a. The displayed quantification is the same as displayed in Extended Data Fig. 8b. **i**, Representative images of droplet formation by purified NFYC-IDR-mEGFP (1 μM) and MED1 IDR-mCherry (5 μM) fusion proteins in the presence of in vitro transcribed Cy5-labeled IAP RNA fragment. Scale bar, 5 μm. **j**, Partitioning ratio of NFYC-IDR-mEGFP, MED1 IDR-mCherry and IAP RNA into droplets at the indicated IAP RNA concentrations. Every dot represents a detected droplet. All pairwise *P* values $<2.2 \times 10^{-16}$ (Welch's *t*-test). **k**, Schematic model of the experiment mimicking *IAPEz* transcription. **l**, qRT–PCR data as FC-normalized to the DMSO control treatment. n.d., not detectable. Data are presented as mean values ± s.d. *P* values are from two-tailed *t*-tests. ****$P < 1 \times 10^{-4}$. **m**, qRT–PCR data as FC-normalized to the DMSO control treatment. Data are presented as mean values ± s.d. from three biological replicates. *P* values are from two-tailed *t*-tests. ****$P < 1 \times 10^{-4}$, ***$P < 1 \times 10^{-3}$, **$P < 1 \times 10^{-2}$, NS, not significant. In panels **f**, **g**, **h** and **j**, data for quantification were acquired from at least five images of two independent image series per condition.

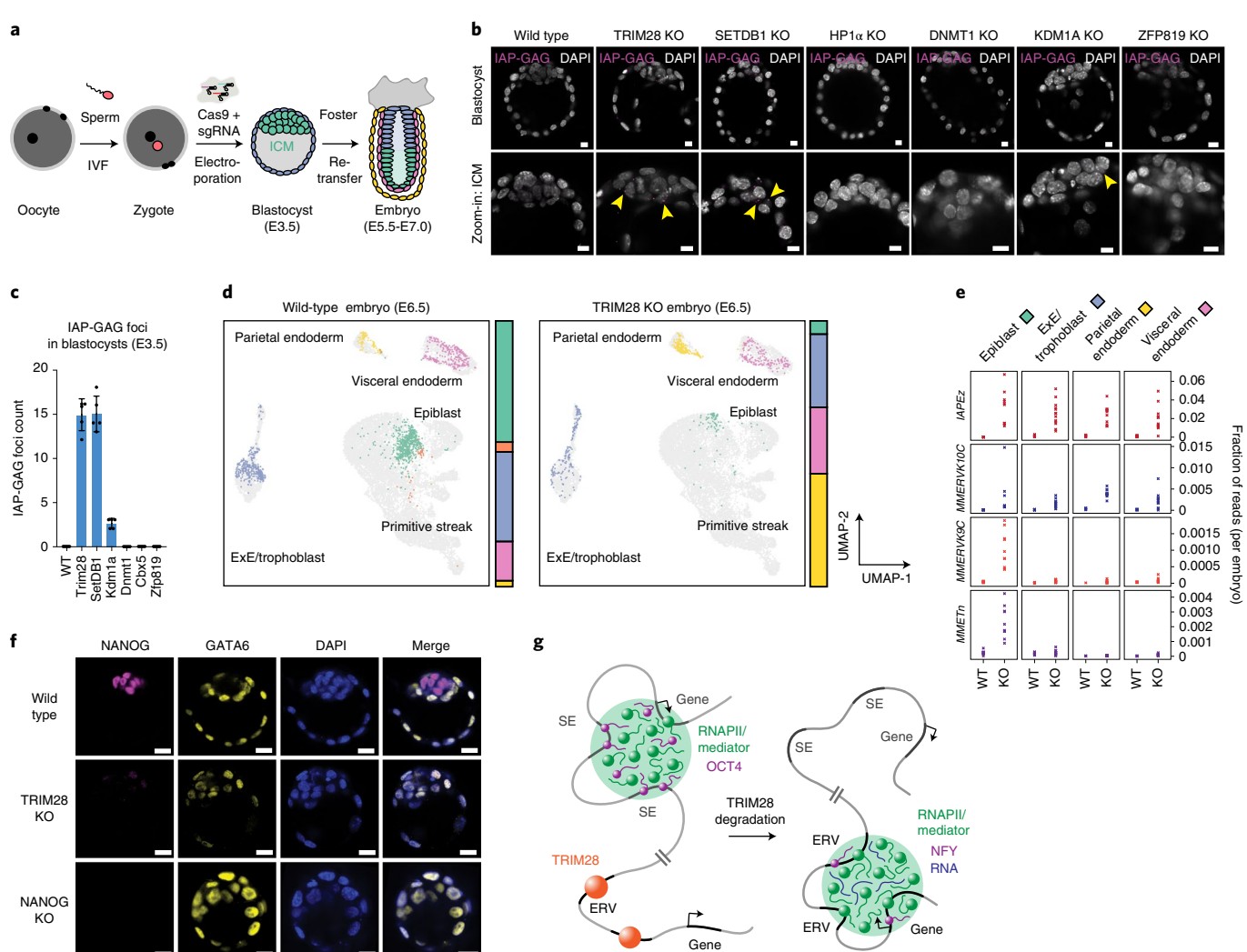

**Fig. 5 | Early ERV activation correlates with depletion of pluripotent lineages in mouse embryos. a**, Scheme of the zygotic CRISPR–Cas9 perturbation platform. **b**, IF images of mouse E3.5 blastocysts stained for the GAG protein produced by IAPs. Nuclei are counterstained with DAPI. Note the magenta IAP GAG foci highlighted with yellow arrowheads. Scale bar, 10 μm. **c**, Quantification of IAP GAG foci in multiple embryos of the indicated genotype across three independent perturbation experiments. Five embryos were picked from the pool of embryos from each genotype for the staining. Each dot represents the GAG foci from an individual embryo. Data are presented as mean values ± s.d. **d**, Epiblast cells are depleted in TRIM28 KO embryos. Uniform manifold approximation and projection (UMAP) of E6.5 wild-type and E6.5 TRIM28 KO embryos mapped on the combined reference cell state map. The proportions of cells that belong to the individual cell states are indicated as a bar on the right of the UMAP plots. Exe, extraembryonic ectoderm. **e**, Lineage-specific ERV derepression in TRIM28 KO mouse embryos. The plot shows the fraction of RNA-seq reads that map to the displayed ERV taxa in the indicated cell types in wild-type (WT) and TRIM28 KO embryos in the scRNA-seq data. Each 'x' represents a single embryo. **f**, The inner part of TRIM28 KO blastocysts is populated by GATA6-expressing, NANOG-negative cells. Displayed are representative IF images of NANOG and GATA6 in E3.5 wild-type, TRIM28 KO and NANOG KO blastocysts across two independent perturbation experiments with around 20 embryos per condition. Scale bars, 20 μm. **g**, Condensate hijacking model. In pluripotent cells, transcriptional condensates associate with SEs bound by pluripotency TFs (for example, OCT4). In the absence of TRIM28, transcriptional condensates are lost from SEs and associate with derepressed ERVs.

mimicked the effect of simultaneous ERV induction by generating an mESC line containing multiple copies of an integrated PiggyBac transposon (Extended Data Fig. 9a–d). The transposon encoded a Dox-inducible green fluorescent protein (GFP) transgene with a polyA between two loxP sites, and ~900 bp fragments of IAPEz ERVs (Fig. 4k). Transfection of a plasmid encoding a Cre-recombinase enabled the generation of an isogenic line ('IAPEz line') encoding Dox-inducible *IAPEz* transgenes with the same copy number and insertion sites as *GFP* in the parental mESC line ('GFP line') (Fig. 4k). Quantitative PCR with reverse transcription (qRT–PCR) analyses confirmed induction of either *GFP* or *IAPEz* transcription upon Dox treatment in the respective lines (Fig. 4l). Moreover, induction of *IAPEz* transcription led to a rapid reduction of SE

transcription at the *miR290-295*, *Klf4* and *Fgf4* loci, and reduced transcript levels of the associated genes, whereas it generally did not affect transcript levels of typical enhancer-associated genes (Fig. 4m and Extended Data Fig. 9e). In contrast, induction of GFP transcription had only a mild effect on SEs (Fig. 4m and Extended Data Fig. 9e). Consistent with the specific effect of IAPEz RNA induction, cellular fractionation experiments revealed that about twice as much of the *IAPEz* RNA is retained in the nuclear fraction compared with the *GFP* RNA (Extended Data Fig. 9f). Similar results were observed in a second pair of mESC lines in which the *IAPEz* fragment was substituted with fragments of *MMERVK10C* ERVs of around 900 bp (Extended Data Fig. 9g–i). Induction of *MMERVK10C* transcription from a PiggyBac transposon compromised transcription at SEs

and their associated genes (Extended Data Fig. 9g–i). These results demonstrate that simultaneous activation of transgenic ERVs may compromise SE transcription in mESCs, and the ERV RNA seems to play an important role in this process.

**ERV derepression correlates with loss of pluripotent cells.** The above results suggest that pluripotent stem cells fail to maintain transcription of SE-driven genes when ERV repression is compromised. This model predicts that the amount of ERV products would correlate with the inability of embryos to maintain a pluripotent compartment. To test this model in vivo, we used our recently developed zygotic perturbation platform (Fig. 5a)[40–42]. We generated zygotic deletion mutants of TRIM28, SETDB1, HP1α and other epigenetic regulators implicated in ERV repression, and assayed the timing and amount of the GAG protein produced by IAPs (Fig. 5b). IAP GAG foci were detected in E3.5 blastocysts of TRIM28, SETDB1 and KDM1A knockout (KO) mutants, and these mutations were lethal at around E6.5 in embryos (Fig. 5b,c) suggesting that early appearance of IAP GAG foci may correlate with the onset of embryonic lethality.

To probe which cell types are affected by derepression of ERVs, we used single-cell RNA-seq (scRNA-seq). We created a reference cell state map of an early mouse embryo spanning a window of developmental stages (E5.5 to E7.0) that encompass the onset of lethality in TRIM28 KO mice (around E6.5) (Extended Data Fig. 10a–f,i)[7,43,44]. We then generated TRIM28-deficient embryos using Cas9/sgRNA delivery into zygotes (Extended Data Fig. 10a–c) and mapped cell states using scRNA-seq (Fig. 5d and Extended Data Fig. 10g). The E6.5 TRIM28 KO scRNA-seq data revealed a dramatic scarcity of epiblast cells normally derived from pluripotent cells of the inner cell mass (ICM) (Fig. 5d), and an abundance of extraembryonic lineages (for example, parietal endoderm) (Fig. 5e and Supplementary Fig. 9a,b). Quantifying the fraction of reads mapping to various ERV taxa revealed that IAPs and MMERVK10Cs were derepressed in all cell types, while MMERVK9Cs and MMETns were derepressed specifically in epiblast cells, resulting in a higher total fraction of reads from ERVs (Fig. 5e and Extended Data Fig. 10h). These data indicate that the amount of ERV RNA transcripts is especially high in pluripotent cells in TRIM28 KO embryos.

IF imaging corroborated specific depletion of pluripotent cells in early embryos. The pluripotency factors NANOG, OCT4, SOX2 and KLF4 were already virtually absent in the ICM of TRIM28 KO E3.5 blastocysts (Fig. 5f and Supplementary Fig. 10a-b), and the inner part of the blastocysts was instead populated by cells expressing the endoderm marker GATA6 (Fig. 5f). This phenotype is reminiscent of NANOG knockout embryos, in which the pluripotent ICM is replaced by GATA6-expressing parietal endoderm-like cells (Fig. 5f)[45]. These data are consistent with upregulation of endoderm markers observed in TRIM28-degraded ESCs (Supplementary Fig. 11a-c). Overall, these findings suggest that extended ERV derepression results in the loss of expression of pluripotency genes and consequent depletion of pluripotent cells in early mouse embryos.

## Discussion

The results presented here support a model in which ERV retrotransposons have the capacity to hijack biomolecular condensates formed by key transcriptional regulatory proteins in pluripotent cells (Fig. 5g). This model may help explain why thousands of transposition-incapable ERVs are repressed in mammals and how their reactivation could alter cellular fates in the absence of retrotransposition[7,9,18,46–50].

Derepressed ERVs seem to compete with SEs for transcriptional condensates in pluripotent cells, in part through the RNA transcripts they produce. In vitro, various RNA species facilitate phase separation of transcriptional regulators and heterochromatin proteins, primarily via engaging in electrostatic interactions[21,38]. In ESCs, forced transcription of ERV RNA from multiple loci led to a profound decrease in SE transcription while transcription of *GFP* RNA had a moderate effect (Fig. 4k–m and Extended Data Fig. 9g–i), suggesting sequence contribution to the impact of ERV RNAs on RNAPII-containing condensates in vivo. Recent studies reported that m6A methylation plays an important role in repressing ERV transcripts[11–13]. The contribution of ERV RNA to the hijacking of transcriptional condensates from SEs may explain the in vivo role and importance of ERV RNA modifications.

Many nuclear noncoding RNAs are known to localize to the loci where they are produced[22], but their functions are mysterious. Transient expression of ERVs and nuclear IAP foci have been described in early-stage human and mouse embryos[48,49,51,52] and adult immune cells[53], suggesting that ERV RNAs may play important roles in transcriptional programs during mammalian development. Consistently, previous studies suggested that derepressed transposons can act as enhancers and alternative promoters of cellular genes[29,48,53–55]. RNA transcripts produced by ERVs may thus contribute to the genomic distribution of condensates in various developmental contexts.

Condensate hijacking by ERVs may contribute to disease. *Trim28* haploinsufficiency is associated with obesity[56] and predisposes to Wilms' tumor[57]. Some ERVs may function as enhancers in acute myeloid leukemia[58], and ERV transcription is associated with neurological diseases[59] such as amyotrophic lateral sclerosis[60] and schizophrenia[61]. The capacity of ERV RNAs to hijack transcriptional condensates may shed light on the molecular basis of these conditions.

## Online content

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

## Methods

**Licenses.** All animal procedures were performed in our specialized facility, following all relevant animal welfare guidelines and regulations, approved by the Max Planck Institute for Molecular Genetics and LAGeSo, Berlin (license number, G0247/18-SGr1; and Harvard University (IACUC protocol 28-21)). S2 work was performed following all relevant guidelines and regulations, approved by the Max Planck Institute for Molecular Genetics and the local authorities LAGeSo, Berlin (license number, 222/15-17a).

**Cell culture.** The V6.5 mESCs and iPSCs were cultured on irradiated primary mouse embryonic fibroblasts (MEFs) under standard serum/leukemia inhibitory factor (LIF) conditions (KO DMEM containing 15% fetal bovine serum (FBS), supplemented with 1× penicillin/streptomycin, 1× GlutaMAX supplement, 1× nonessential amino acids, 0.05 mM β-mercaptoethanol (all from Gibco) and 1,000 U ml⁻¹ LIF).

For ChIP–seq, TT-SLAM–seq and RNA-seq experiments, mESCs were depleted from MEFs by incubating them on gelatin-coated cell culture plates for 45 min at 37 °C, allowing MEFs to attach while mESCs remain in suspension. MEF depletion was performed twice, after which mESCs were seeded on gelatin-coated plates and maintained in serum/LIF conditions with 2,000 U ml⁻¹ LIF.

For RNA-FISH combined with IF, MEF-depleted cells were grown on round 18-mm glass coverslips (Roth LH23.1). Coverslips were coated with 5 µg ml⁻¹ of poly-L-ornithine (Sigma-Aldrich, catalog no. P4957) for 30 min at 37 °C and with 5 µg ml⁻¹ of Laminin (Corning, catalog no. 354232) overnight at 37 °C.

To perturb RNAPII condensates, cells were treated for 30 min with 1.5% 1-6 HD (Sigma) in serum/LIF conditions with 2,000 U ml⁻¹ LIF (Extended Data Fig. 3c,d).

**Generation of the TRIM28-FKBP ESC line.** To knock in the degradation-sensitive FKBP^F36V tag at the N-terminus of TRIM28, a repair template containing homology arms spanning upstream and downstream of the target site was cloned into a pUC19 vector (NEB) (Supplementary Fig. 2a). The repair template included a mRuby2 fluorescent protein sequence, P2A linker and the FKBP tag sequence (Supplementary Fig. 2a)[25]. mRuby2 sequence was amplified from the mRuby2-N1 plasmid (Addgene, catalog no. 54614), and the P2A-FKBP sequence was amplified from the PITCh dTAG donor vector (Addgene, catalog no. 91792). A guide RNA (Supplementary Table 1b) targeting the N-terminus of TRIM28 was cloned into the sgRNA-Cas9 vector pX458 (Addgene, catalog no. 48138). The repair template and the sgRNA-Cas9 vector were transfected into V6.5 mESCs and iPSCs by nucleofection using Amaxa 4D Nucleofector X Unit (Lonza) according to the manufacturer's instructions. To screen for positive integrations, the transfected cells were sorted for mRuby2 fluorescent protein expression with flow cytometry. The sorted cells were seeded as single cells and expanded for a few days. Single colonies were picked and genotyped for the correct integration with Western blot.

**TRIM28 degradation.** Before treatment, cells were seeded on 0.2% gelatin-coated plates after two rounds of MEF depletion. For degradation of TRIM28, 500 nM of dTAG-13 compound[25] was mixed with mESC medium (supplemented with 2,000 U mL⁻¹ LIF) and incubated for the time indicated; medium was changed daily for fresh dTAG-13.

**RNA-FISH combined with IF.** RNA-FISH combined with IF was performed essentially as described[31]. For IF, dTAG-13- or DMSO- treated cells were fixed in 4% paraformaldehyde for 10 min at RT and stored in PBS at 4 °C. All buffers and antibodies were diluted in RNase-free PBS (Thermo Fisher, catalog no. AM9624). Cells were permeabilized with 0.5% Triton X-100 (Thermo Fisher, catalog no. 85111) for 10 min at RT, followed by three consecutive 5 min PBS washes. Cells were then incubated in the primary antibody (RNAPII (Abcam, catalog no. ab817) at 1:500, NFY-A (Santa Cruz, catalog no. sc-17753 X) at 1:250, NRF1 (Abcam, catalog no. ab55744) at 1:500, MED1 (Abcam, catalog no. ab64965) at 1:500 and MED23 (Bethyl Labs, catalog no. A300-425A) in PBS overnight. After two 5 min PBS washes, cells were incubated in the secondary antibody (Invitrogen, goat anti-mouse Alexa Fluor 488 (catalog no. A-11001) or goat anti-rabbit Alexa Fluor 488 (catalog no. A-11008)) at 1:500 for 60 min at room temperature. Cells were washed twice in PBS for 5 min and re-fixed with 4% paraformaldehyde in PBS for 10 min at room temperature. Following two 5 min PBS washes, cells were washed once with 20% Stellaris RNA-FISH Wash Buffer A (Biosearch Technologies, catalog no. SMF-WA1-60) and 10% deionized formamide (EMD Millipore, catalog no. S4117) in RNase-free water (Invitrogen, catalog no. 10977035) for 5 min at RT. Cells were hybridized with 90% Stellaris RNA-FISH Hybridization Buffer (Biosearch Technologies, catalog no. SMF-HB1-10), 10% deionized formamide and 12.5 or 25 µM Stellaris RNA-FISH probes. Probes were hybridized in a humidified chamber overnight at 37 °C. Cells were washed with Wash Buffer A for 30 min at 37 °C and stained with 0.24 µg ml⁻¹ 4,6-diamidino-2-phenylindole (DAPI) in Wash Buffer A for 3 min at room temperature. Cells were washed with Stellaris RNA-FISH Wash Buffer B (Biosearch Technologies, catalog no. SMF-WB1-20) for 5 min at RT, mounted onto glass microscopy slides with Vectashield mounting medium (Vector Laboratories, catalog no. H-1900) and sealed using transparent nail polish. Images were acquired with LSM880

Airyscan microscope equipped with a Plan-Apochromat ×63/1.40 oil differential interference contrast objective or Z1 Observer (Zeiss) microscope with ×100 magnification with Zen 2.3 v.2.3.69.1016 (blue edition) or Zen (black edition). Images were processed with ZEN 3.1 (Zeiss) and ImageJ software v.2.1.0/1.53i (Figs. 1j, 2a–b, 2f–g, 3f and 4d and Extended Data Figs. 2a, 3a–b,d, 4d and 6h). ImageJ colocalization plugins were used for colocalization analysis of ERV IAP RNA-FISH with RNAPII and MED1 IF[62,63]. For nearest RNAPII cluster distance analysis in the miR290-295 RNA-FISH dataset, z-projections consisting of ±4.5 slices around the FISH spot were obtained in both channels and thresholded to allow detection of individual RNAPII clusters. Center of mass distances to the nearest cluster were calculated using FIJI (DiAna)[63]. RNA-FISH probes were designed and generated by Biosearch Technologies Stellaris RNA-FISH to target introns of miR290-295 primary transcript and Cthrc1, and IAPEz transcripts. Sequences of RNA-FISH probes are available in Supplementary Table 1a.

**Live-cell PALM.** Live-cell PALM imaging was carried out as described before[32,64,65]. mESCs used for live-cell PALM imaging were derived from R1 background, with the Sox2 gene tagged with 24 repeats of MS2 stemloops at its mRNA 3′ end, Rpb1 tagged with Dendra2 at its N-terminus, EF1α-NLS-MCP-SNAP inserted stably into the genome and both alleles of Trim28 tagged with the degradation-sensitive FKBP tag (Extended Data Fig. 2f). Cells were simultaneously illuminated with 1.3 W cm⁻² near UV light (405 nm) for photoconversion of Dendra2 and 3.2 kW cm⁻² (561 nm) for fluorescence detection with an exposure time of 50 ms. We acquired images of Dendra2-RNAPII for 100 s (2,000 frames) for quantification of Pol II clusters. For dual-color imaging, cells were incubated with 100 nM JF646 SNAP ligand for 20 min and washed with 2i medium, followed by 30 min incubation in 2i media without JF646-HaloTaq ligands, to wash out unbound SNAP ligands before fluorescence imaging in L-15 medium. We acquired 50 frames (2.5 s) with 642 nm excitation with a power intensity of 2.5 kW cm⁻² and quickly switched to simultaneous 405/561 imaging for PALM. Super-resolution images were reconstructed and analyzed using MTT[66] and qSR[67]. RNAPII cluster size was defined as the total number of localizations within the image acquisition time (100 s). The distance was calculated as the distance between the center of the MS2 nascent transcription site and the center of the nearest RNAPII cluster (Fig. 1k).

**TT-SLAM-seq.** TT-SLAM-seq was performed as described previously[26]. Briefly, cells were treated with DMSO or 500 nM dTAG-13 for 2, 6 or 24 h and subjected to 15 min of 4-thiouridine (4sU) labeling using 500 µM 4sU. Total RNA was extracted with Trizol (Ambion) and 24:1 chloroform:isoamylalcohol (Sigma) while using 0.1 mM dithiothreitol (DTT) in isopropanol precipitation and ethanol washes. For each sample, 50 µg of total RNA was fragmented with Magnesium RNA Fragmentation Module (NEB), and fragmentation buffer was removed from samples with ethanol precipitation in presence of 0.1 mM DTT. RNA was then resuspended in 350 µl RNase-free water, diluted in biotinylation buffer (200 mM HEPES pH 7.5, and 10 mM EDTA) and topped up with 5 µg MTS-Biotin (previously diluted to 50 µg ml⁻¹ in dimethylformamide) to reach a final volume of 500 µl. The biotinylation reaction was incubated for 30 min at room temperature while keeping samples in rotation and protected from light. Unbound biotin was removed with acid-phenol:chloroform extraction (125:24:1, Ambion) and isopropanol precipitation. Biotinylated RNA was resuspended in 100 µl RNase-free water, denatured in 65 °C for 10 min and then cooled on ice for 5 min. The biotinylated RNA was captured with 100 µl µMACS streptavidin beads (Miltenyi) by incubating for 15 min in rotation while keeping samples protected from light. µMACS columns were equilibrated on magnetic stand with nucleic acid equilibration buffer and two times with biotinylation buffer (20 mM HEPES, 1 mM EDTA, pH 8). Beads were transferred to columns and washed three times with wash buffer (100 mM Tris-HCl pH 7.5, 10 mM EDTA, 1 M NaCl and 0.1 % Tween 20), and labeled RNA was eluted twice with a total 200 µl of 100 mM DTT. RNA was cleaned up with RNeasy Minelute columns (Qiagen) and eluted to RNase-free water with 1 mM DTT. 4sU residues of RNA were alkylated with iodoacetamide treatment (10 mM iodoacetamide in 50 mM NaPO₄, pH 8 and 50 % DMSO) by incubating samples in 50 °C for 15 min, followed by quenching with 20 mM DTT. RNA samples were purified with ethanol precipitation and treated with Turbo DNase (Invitrogen). Sequencing libraries were prepared with NEBNext Ultra II Directional RNA Library Prep Kit and NEBNext Multiplex Oligos (NEB), according to manufacturer's instructions, except using 8 min incubation time in fragmentation step.

**Generating wild-type and mutant mouse embryos.** Zygotes were generated by in vitro fertilization (IVF) as previously described[68]. Briefly, B6D2F1 female mice aged 7–9 weeks were superovulated with two rounds of hormone injections (5 IU of pregnant mare serum gonadotrophin followed by 5 IU of human chorionic gonadotrophin after 46 h). Oocytes were isolated and cultured in pre-gassed KSOM medium before IVF. F1 (C57BL/6J × Castaneous) sperm isolated from the cauda epididymis were thawed and used for IVF. At 6 h after fertilization, zygotes were washed in M2 medium for multiple rounds and then prepared for electroporation. Alt-R CRISPR–Cas9 and guide RNAs ribonucleoproteins were prepared as described previously[40]. Guide RNAs used to target the genes are listed in Supplementary Table 1b. Zygotes were washed in three drops of OptiMEM

Reduced Serum Medium (Thermo Fisher Scientific) before electroporation. NEPA21 electroporator (NEPAgene) was used for electroporating zygotes with the following settings for a small chamber: four poring pulses of 34 V for 2.5 ms with an interval of 50 ms were used to generate pores in the zona pellucida layer. Voltage decay was set at 10% and (+) polarity. To enable intake of the ribonucleoproteins, five transfer pulses of 5 V were applied for 50 ms each with an interval of 50 ms. Voltage decay for the transfer was set at 40% with an alternating polarity of (+) and (−). Electroporated zygotes were washed in three drops of KSOM medium and cultured in pre-gassed KSOM drops until blastocyst stage under standard embryo culture conditions. Blastocysts were scored for viability and morphology and retransferred bilaterally in a clutch of 15 blastocysts per uterine horn into day 2.5 pseudopregnant CD-1 surrogate female mice. E6.5 embryos were dissected from the uterus in 1× Hanks' Balanced Salt Solution and used for further analysis. E5.5 wild-type embryos were generated with the setup, and mock electroporation with guide targeting GFP sequence was used.

**scRNA-seq of embryos.** E5.5 wild-type and E6.5 TRIM28 mutant embryos were dissected from the decidua in 1× Hanks' Balanced Salt Solution and then washed in 1× PBS. Reichert's membrane was removed carefully with sharp forceps and glass capillaries, and the embryos were washed in 1× PBS with 0.4% BSA. The embryos were disaggregated with TrypLE Express (Gibco) with gentle pipetting every 10 min up to a total of 40 min at 37 °C. The dissociated cells were counted for viability and then washed in 1× PBS with 0.4% BSA for a total of three washes at 4 °C and 1,200 r.p.m. for 5 min. The cells were subjected to scRNA-seq using a 10x Genomics Chromium Single Cell 3′ v.2 kit. Single-cell libraries were generated following the manufacturer's instructions with the exception of the cycle number used. Libraries were sequenced on a Novaseq6000 with asymmetric reads and a depth of 300–350 million fragments per library.

**Average image and radial distribution analysis.** The image analysis pipeline used for the colocalization analysis of RNA-FISH combined with IF was described previously[31]. Briefly, MATLAB scripts were used to identify RNA-FISH foci in z stacks through intensity thresholding (the same threshold was used for image sets shown on the same figure panels) and create RNA-FISH signal centroids ($x$, $y$, $z$) that were stitched together and positioned in a box of size $l = 1.5\,\mu$m. For identified FISH foci, signal from corresponding location in the IF channel was collected in the $l \times l$ square centered at the RNA-FISH focus at each corresponding z-slice. The IF signal centered at FISH foci for each FISH and IF pair were then combined to calculate an average intensity projection, providing averaged data for IF signal intensity within a $l \times l$ square centered at FISH foci. The same process was carried out for the FISH signal intensity centered on its own coordinates, providing averaged data for FISH signal intensity within a $l \times l$ square centered at FISH foci. As a control, this same process was carried out for IF signal centered at random nuclear positions generated using custom Python scripts. These average intensity projections were then used to generate two-dimensional contour maps of the signal intensity or radial distribution plots. Contour plots are generated using inbuilt functions in MATLAB. The intensity radial function (($r$)) is computed from the average data. For the contour plots of the IF channel, an intensity colormap consisting of 14 bins with gradients of black, violet and green was generated. For the FISH channel, black to magenta was used. The generated colormap was employed to 14 evenly spaced intensity bins for all IF plots. The averaged IF centered at FISH or at randomly selected nuclear locations were plotted using the same color scale. For the radial distribution plots, the Spearman correlation coefficients, $r$, were computed and reported between the FISH and IF (centered at FISH) signal. A two-tailed Student's $t$-test, comparing the Spearman correlation calculated for all pairs, was used to generate $P$ values (Figs. 1j, 2f–g, 3f and 4d and Extended Data Figs. 2a and 4d).

**Bioinformatics.** All analyses were carried out using R v.3.6.3 unless stated otherwise.

**TT-SLAM-seq processing.** Raw reads were trimmed by quality, Illumina adapter content and polyA content analogous to the RNA-seq samples and aligned with STAR with parameters '−outFilterMultimapNmax 50−outReadsUnmapped Fastx' to the SILVA database[69] (downloaded 6 March 2020) to remove rRNA content. Unaligned reads were afterwards reverse-complemented using the seqtk 'seq' command (https://github.com/lh3/seqtk, v.1.3-r106; parameters: -r). Reverse-complemented reads were processed using SLAM-DUNK[70] with the 'all' command (v.0.4.1; parameters: -rl 100 -5 0) with the GENCODE gene annotation (VM19) as '-b' option. Reads with a 'T > C' conversion representing nascent transcription were filtered from the BAM files using alleyoop (provided together with SLAM-DUNK) with the 'read-separator' command. Counts per gene were quantified based on the 'T > C'-converted reads using htseq-count (v.0.11.4; parameters:−stranded=yes,−nonunique=all)[71]. FPKM values were calculated based on the resulting counts. For genome-wide coverage tracks, technical replicates were merged using samtools 'merge'[72]. Coverage tracks for single and merged replicates were obtained using deepTools bamCoverage[72] (v.3.4.3; parameters:−normalizeUsing CPM) separately for the forward and reverse strand based on the 'T > C'-converted reads.

**Enhancer and SE annotation.** The annotation of SErs, enhancers and enhancer constituents was taken from Whyte et al.[73]. Coordinates were lifted from mm9 to mm10 using UCSC liftOver. These coordinates were used throughout this study for all enhancer-associated analyses (Supplementary Table 2).

**Retrotransposon element definition.** The genome-wide retrotransposon annotation of LTR, LINE and SINE elements was downloaded from Repbase[74]. Based on the Repbase classification system, we used the element annotation as LTR, LINE or SINE as the retrotransposon classes. Retrotransposon families considered in this study were L1 and L2 elements (LINE), ERV1, ERV3, ERVK, ERVL and MALR (LTR), as well as Alu, B2, B4 and MIR elements (SINE). Repeat subfamilies used in this study were subdivided into IAP, MMERVK and MMETn (ERVK) elements. IAPs and MMERVKs consist of multiple different subfamilies as annotated by Repbase (Supplementary Figs. 1a and 4a), which we summarized under these broader terms. The classification is consistent with retrotransposon classification described in previous studies[1,75,76].

Full-length retrotransposons were defined based on the Repbase repeat annotation. For full-length ERVK elements, we required the element to consist of an inner part with two flanking LTRs. First, elements annotated as inner parts (containing the keyword 'int') were merged if they belonged to the same subfamily and were located within maximal 200 bp of each other. Second, only the merged inner parts with an annotated ERVK LTR within a distance of, at most, 50 bp on each side were selected as full-length element candidates. For IAPs specifically, only LTRs that belonged to an IAP subfamily were considered. No size restrictions were applied on the inner parts or LTRs, which could lead to potential false positive candidates that are too truncated to be able to be transcribed, but, on the other hand, provides an unbiased definition of full-length repeat elements. The subfamily per element was defined based on the inner part. Inner parts flanked by one LTR were termed half-length elements. LTRs without an inner part were termed solo LTRs. To provide a broad overview of potential full-length L1 elements, only annotated elements with a size of greater than 6 kb were shown. The genomic coordinates of retrotransposons are listed in Supplementary Table 2a–e.

**scRNA-seq processing.** Fastq files for the wild-type timepoints E6.5 and E7.0 were downloaded from GEO (Supplementary Table 1c)[77]. For the wild-type time point E5.5 and the Trim28 KO, raw reads (fastq) were generated using Cell Ranger (https://support.10xgenomics.com/single-cell-gene-expression/software/downloads/latest) (v.4) from 10x Genomics Inc. with the command 'cellranger mkfastq.' Reads from all timepoints were aligned against the mouse genome (mm10), and barcodes and unique molecular identifiers were counted using 'cellranger count'. Multiple sequencing runs were combined using 'cellranger aggr.'

**Retrotransposon expression quantification.** Global repeat expression quantification from RNA-seq, TT-SLAM-seq and scRNA-seq (Fig. 1e and Supplementary Fig. 4b-e) was carried out as described[40]. Briefly, to estimate the expression for each retrotransposon subfamily without bias due to gene expression, only reads not overlapping any gene were considered for the analysis. Reads overlapping splice sites, as well as reads with a high polyA content, were removed. The remaining reads were counted per subfamily only if they aligned uniquely or multiple times to elements of the same subfamily. Here, any annotated element of a specific subfamily from Repbase was considered independent of our full-length ERVK annotations. Reads aligning to multiple elements were counted only once. For scRNA-seq samples, reads were counted per subfamily, sample and cell state. The number of reads per subfamily was normalized by library size for RNA-seq and TT-SLAM-seq samples and normalized by reads aligning to genes and repeats for scRNA-seq samples. Fold change (FC) was calculated with respect to the DMSO or wild-type samples.

**Statistical tests.** The statistical significance of the difference of IAP expression between DMSO control and dTAG timepoints for TT-SLAM-seq and RNA-seq was calculated using an unpaired two-sided $t$-test (Fig. 1e). Statistical significance of differences in FC (versus DMSO) in control versus 1-6 HD-treated cells was estimated with unpaired two-sided $t$-test (Extended Data Fig. 3e). All other tests are described in the figure legends.

**Definition of boxplot elements.** In Figs. 1i and 4c,f–h,j, Extended Data Figs. 7c,e and 8b and Supplementary Fig. 7c, elements depicted in boxplots are as follows: middle line, median; box limits, upper and lower quartile; whiskers, 1.5× interquartile range. In Extended Data Figs. 2b–d and 5c, elements depicted in dot plots are as follows: middle line, mean; whiskers, s.d.; points, all data points.

**Statistics and reproducibility.** For all RNA-FISH combined with IF experiments, the target combination of gene transcript and transcriptional activator was probed on one coverslip of mESCs and at least two viewpoints were acquired. The number of detected foci included in the radial plot analysis is indicated under $n_{foci}$ in Figs. 1j, 2f,g, 3f and 4d and Extended Data Figs. 2b,d, 4d and 6j. For Fig. 2a,b, $n$ indicates the number of analyzed nuclei collected from at least three viewpoints, whereas the total number of detected IAPez foci is indicated in Fig. 2c (1,774 for RNAPII and 2,735 for MED1). Colocalizing foci (distance <200 nm) from Fig. 2a,b are

indicated in Fig. 2c (344 of 1,774 for RNAPII and 381 of 2,735 for MED1). In Fig. 1h, enhancer constituents with significant transcription (FPKM > 1) are included (n = 117 for super-enhancers, n = 153 for typical enhancers).

IAP RNA-FISH–RNAPII IF experiments were repeated three times. Images and analysis of one representative experiment are displayed in Fig. 2a, and those from a second replicate experiments in Extended Data Fig. 3a. IAP RNA-FISH–MED1/MED23 IF images are from one biological replicate staining experiment (Fig. 2b and Extended Data Fig. 3b). IF images of 1-6 HD-treatment experiments (Extended Data Fig. 3d) and Cthrc1 RNA–NRF1 IF images are from one biological replicate staining experiment (Extended Data Fig. 4d).

For in vitro biochemistry experiments (Fig. 4e,i and Extended Data Figs. 7a,b,d,g,h and 8a), at least one independent slide containing the indicated mix was imaged and at least five independent viewpoints were acquired for each slide. Data are displayed as boxplots (Fig. 4f,g,h,j and Extended Data Figs. 7c,e,i and 8b), and each dot represents an individual droplet (n is the total number of droplets). In the boxplots, the lower box limit was set to the 25th percentile, upper box limit was set to the 75th percentile, the center line indicates the median and the whiskers represent the range within 1.5× interquartile. The following numbers of viewpoints and droplets were analyzed (formatted as condition per number of replicates per number of viewpoints per experiment per total number of droplets shown if the figure contains a boxplot): Fig. 4e,f,g, (IAP RNA: 0 nM; Fluorescein: 73 nM)/3/5/4,152, (3.7 nM)/3/5/1,337, (37 nM)/3/5/8,142, (73 nM)/3/5/11,018; Fig. 4h, (all conditions)/1/5/5 (each dot represents an image); Fig. 4i,j, (IAP RNA: 0 nM)/2/5/308, (15 nM)/2/5/332; Extended Data Fig. 7a–e, (IAP RNA: 0 nM; Fluorescein: 73 nM)/2/5/945, (3.7 nM)/2/5/293, (37 nM)/2/5/217, (73 nM)/2/5/321; Extended Data Fig. 7g, (all conditions)/2/5; Extended Data Fig. 7h,i, (mEGFP)/2/5/337, (NFYC-IDR-mEGFP)/2/5/434; Extended Data Fig. 8a,b, (IAP RNA-Cy5 all combinations)/1/5/5 images, (Maj Sat Repeat RNA-Cy5 all combinations)/1/10/10 images, (MiR290-295 SE RNA-Cy5 with MED1 IDR-mEGFP)/1/10/10 images and (MiR290-295 SE RNA-Cy5 with HP1α-mCherry)/1/20/20 images).

Sample sizes in Fig. 1k are as follows: left, sample size: 157 (DMSO), 112 (dTAG-13) cells; middle left, sample size: 157 (DMSO), 112 (dTAG-13) cells; middle, right, sample size: 2,591 (DMSO), 1,572 (dTAG-13) RNAPII clusters; right, number of RNAPII clusters per cell: sample size: 94 (DMSO), 59 (dTAG-13) cells.

Sample sizes in Fig. 4c are as follows: left panel shows SE constituents with FPKM > 0.05 (n = 163), middle panel contains SE-associated genes (n = 185) and right panel includes all active genes FPKM > 1 (n = 11,525).

**Reporting summary.** Further information on research design is available in the Nature Research Reporting Summary linked to this article.

## Data availability
All data are available in the Supplementary Information. Sequence data were deposited at GEO under the accession number GSE159468. Mass spectrometry data were deposited at ProteomeXchange under the accession ID PDX021895. Plasmids generated in the study are available at Addgene. Source data are provided with this paper.

## Code availability
All custom code used in this study was deposited along with the raw data at Zenodo (https://doi.org/10.5281/zenodo.6521914).

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

## Acknowledgements
We thank T. Aktas and A. Bulut-Karslioglu for comments on the manuscript, C. Haggerty (MPIMG) for S2 cells, G. Winter (CeMM, Vienna), M. Jäger (CeMM, Vienna), M. Biesaga (IRB, Barcelona) and X. Salvatella (IRB, Barcelona) for sharing RNAPII antibody, A. Dall'Agnese and E. Guo for advice on FISH-IF, S. Basu for helping with colocalization analysis, S. Mackowiak for help with code, C. Althoff for help with Western blots, B. Lukaszewsa-McGreal for help with mass spectrometry, C. Hillgardt, C. Franke and the MPIMG transgenic facility for help with mouse work and the MPI-MG Sequencing core for sequencing. This work was funded by the Max Planck Society and partially supported by grants from the NIH (1P50HG006193 to A.M.; 5R01GM134734 to I.I.C.), the Deutsche Forschungsgemeinschaft (DFG) Priority Program SPP 2202 Grant HN 4/1-1 and HN 4/3-1 (to D.H.), the Austrian Academy of Sciences and the European Research Council (ERC) under the European Union's Horizon 2020 research and innovation program (ERC-CoG-866166, RiboTrace) (to S.L.A.). H.N. is supported by fellowships from the Emil Aaltonen, Orion Research Foundation and Instrumentarium Science Foundation.

## Author contributions
V.A. conceived the study, generated mESC lines, designed and performed RNA-FISH-IF experiments, analyzed and interpreted data, supported TT-SLAM-seq experiments and revised the manuscript. A.S.K. conceived the study, generated and characterized mESC lines, performed directed differentiation experiments, scRNA-seq, repeat knockdown, IAP-gag RNA-FISH, in vivo KO experiments, tetraploid aggregation experiments, whole-embryo IF, flow cytometry experiments and analysis and revised the manuscript. H.N. conceived the study, performed ChIP–seq, TT-SLAM-seq and in situ Hi-C experiments, analyzed and interpreted the data from TT-SLAM-seq and Hi-C experiments, assisted in RNA-seq analysis, performed and supported qRT–PCR experiments, generated ERV-TKO cell lines and revised the manuscript. C.R. conceived the study, designed and performed OSKM ectopic expression and ERV overexpression experiments, generated and characterized iPSC-OSKM-TRIM28-FKBP, mESC-IAPEz and mESC-MMERVK10C cell lines and revised the manuscript. S.H. performed initial processing of RNA-seq, TT-SLAM-seq, ChIP–seq and scRNA-seq datasets, analyzed and interpreted RNA-seq, ChIP–seq and scRNA-seq data, assisted in the TT-SLAM-seq data analysis and revised the manuscript. J.N. performed protein purification and droplet assays. N.F., N.P. and S.L.A. supported TT-SLAM-seq methodology and data analysis. M.D. and I.I.C. designed, performed and analyzed time-correlated PALM experiments. H.K. assisted in scRNA-seq analysis. Z.D.S. and L.W. performed in vivo *Trim28* KO and tetraploid aggregation experiments. R.W. generated and performed NPC experiments. M.W. supported cloning and qRT–PCR experiments. R.B. performed quantification for IAP-gag RNA-FISH. S.M. wrote code for image analysis. D.M. performed mass spectrometry. B.T. supported next-generation sequencing methodology. A.M. conceived the study and supervised the work. D.H. conceived the study, supervised the work and wrote the manuscript.

## Funding

## Competing interests
S.L.A. declares competing interest based on a granted patent related to SLAMseq. S.L.A. is co-founder, advisor and member of the board of QUANTRO Therapeutics GmbH

## Additional information
**Extended data** is available for this paper at https://doi.org/10.1038/s41588-022-01132-w.

**Correspondence and requests for materials** should be addressed to Denes Hnisz.

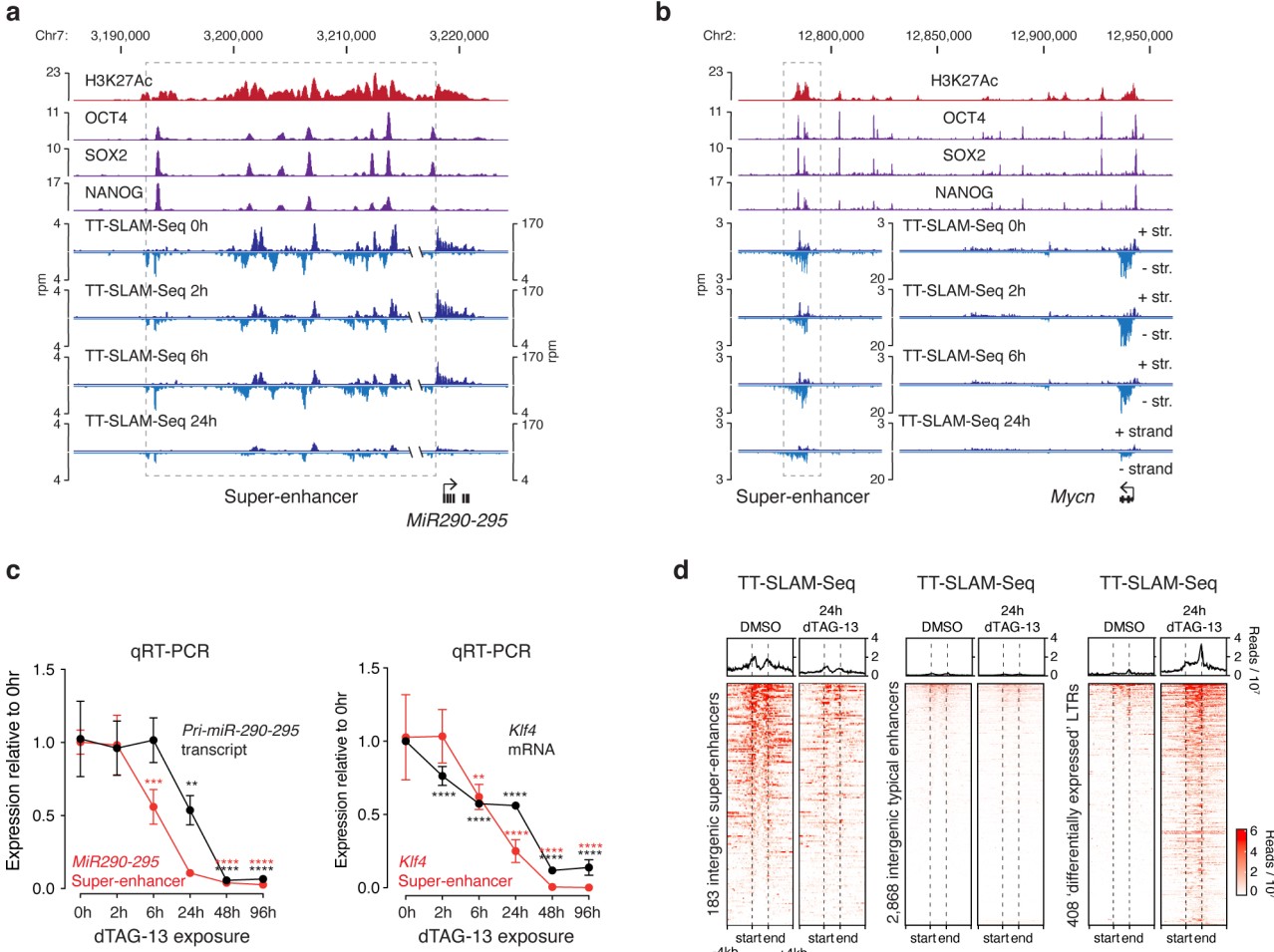

**Extended Data Fig. 1 | Reduction of super-enhancer transcription and the pluripotency circuit in TRIM28-degraded mESCs. a.** Acute reduction of transcription at the *miR290-295* super-enhancer locus upon TRIM28-degradation. Displayed are genome browser tracks of ChIP-seq data (H3K27Ac, OCT4, SOX2, NANOG) in control mESCs, and TT-SLAM-seq data upon 0 h, 2 h, 6 h and 24 h dTAG-13 treatment at the *miR290-295* locus. Rpm: reads per million. Co-ordinates are mm10 genome assembly co-ordinates. **b.** Acute reduction of transcription at the *Mycn* super-enhancer locus upon TRIM28-degradation. Displayed are genome browser tracks of ChIP-seq data (H3K27Ac, OCT4, SOX2, NANOG) in control mESCs, and TT-SLAM-seq data upon 0 h, 2 h, 6 h and 24 h dTAG-13 treatment at the *Mycn* locus. Rpm: reads per million. Co-ordinates are mm10 genome assembly co-ordinates. **c.** qRT-PCR validation of the TT-SLAM-seq data at the *miR290-295* and *Klf4* loci. Displayed are transcript levels after the indicated duration of dTAG-13 treatment. Values are displayed as mean ± SD from three independent experiments and are normalized to the level at 0 h. *P* values are from two-tailed *t*-tests. ****: $P < 10^{-4}$, ***: $P < 10^{-3}$, **: $P < 10^{-2}$, *: $P < 0.05$. **d.** Visualization of nascent transcripts at super-enhancers, enhancers and de-repressed LTR retrotransposons. Displayed are TT-SLAM-seq read densities from both strands within 4 kb around the indicated sites. The genomic features (the middle part of the plot) were length normalized. Meta-analyses of the mean signals are displayed above the heatmaps.

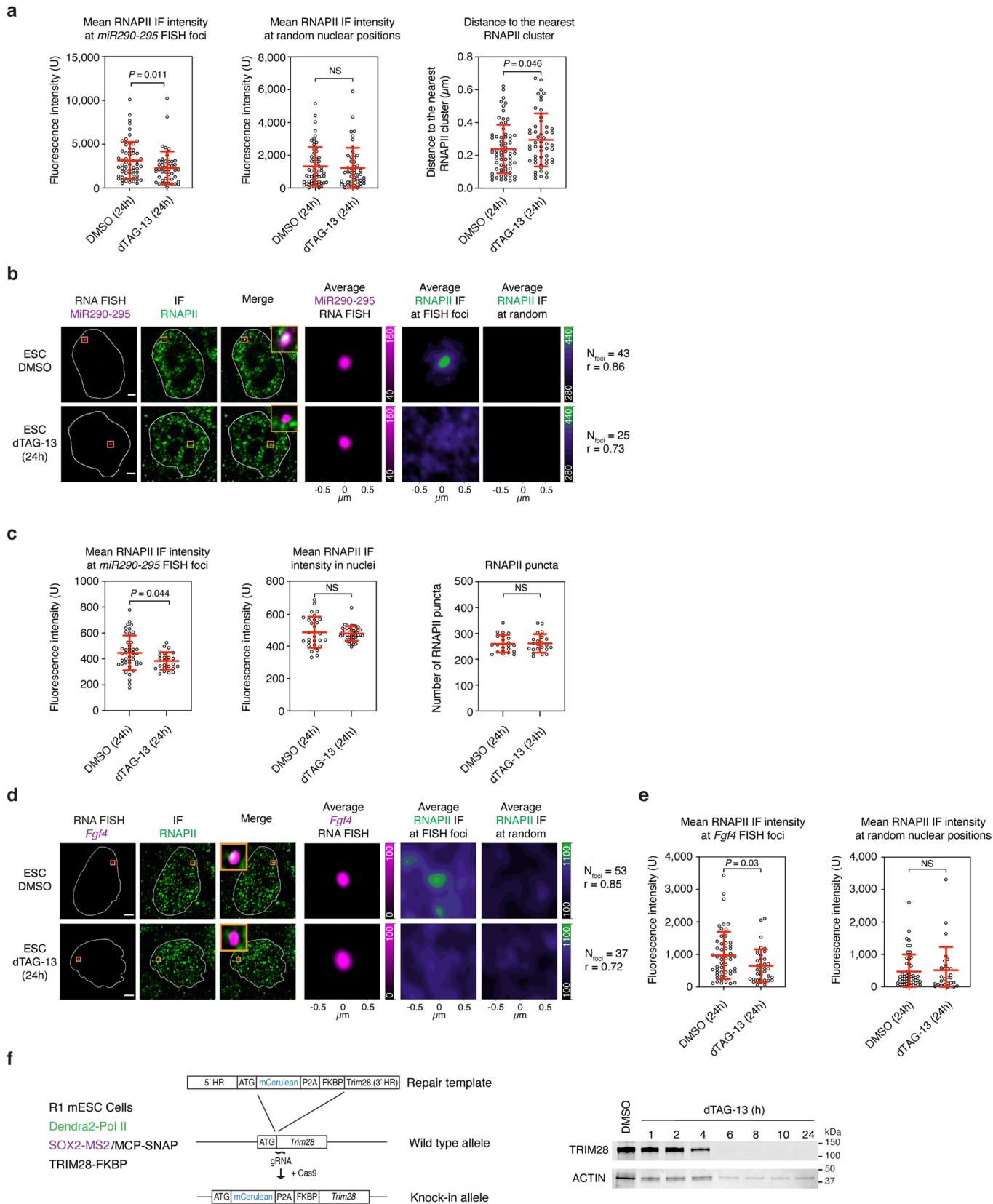

**Extended Data Fig. 2 | See next page for caption.**

**Extended Data Fig. 2 | Loss of SE-association with RNAPII puncta at super-enhancers. a**. Analyses of cells used in Fig. 1j. (left) RNAPII IF intensity at the *miR290-295* FISH foci ($n_{DMSO}=61$, $n_{dTAG-13}=50$). (middle) RNAPII mean fluorescence intensity at random nuclear positions ($n_{DMSO}=61$, $n_{dTAG-13}=50$). (right) Distance between the FISH focus and the nearest RNAPII puncta ($n_{DMSO}=67$, $n_{dTAG-13}=53$). Data presented as mean values $\pm$ SD from one staining experiment. *P* values are from two-sided Mann-Whitney tests. NS: not significant. **b**. Images of RNA-FISH and IF signal. Nuclear periphery determined by DAPI staining is highlighted as a white contour. Also shown are averaged signal of either RNA FISH or RNAPII IF centered on the *miR290-295* FISH foci or randomly selected nuclear positions. Data were collected as an independent replicate of experiments displayed in Fig. 1j. Scale bars: 2.5 μm. **c**. Analysis of cells used in panel '**b**'. (left) RNAPII IF intensity at the *miR290-295* FISH foci ($n_{DMSO}=43$, $n_{dTAG-13}=25$). (center) RNAPII mean fluorescence intensity ($n_{DMSO}=30$, $n_{dTAG-13}=40$). (right) Number of RNAPII puncta on a representative set of cells ($n_{DMSO}=22$, $n_{dTAG-13}=22$). Data are presented as mean values $\pm$ SD from one staining experiment. *P* values are from two-sided Mann-Whitney tests. NS: not significant. **d**. Images of individual z-slices (same z) of the *Fgf4* RNA-FISH and IF signal. Nuclear periphery determined by DAPI staining is highlighted as a white contour. Also shown are averaged signals of either RNA FISH or RNAPII IF centered on the FISH foci or randomly selected positions. Scale bars: 2.5 μm. **e**. Analyses of cells used in panel '**d**.' (left) RNAPII IF intensity at the *Fgf4* FISH foci ($n_{DMSO}=53$, $n_{dTAG-13}=37$). (right) RNAPII mean fluorescence intensity at random nuclear positions ($n_{DMSO}=53$, $n_{dTAG-13}=29$). Data presented as mean values $\pm$ SD from one staining experiment. *P* values are from two-sided Mann-Whitney tests. NS: not significant. **f**. (left) Scheme of FKBP knock-in strategy in the R1 mESCs used in the PALM experiments. (right) TRIM28 Western blot in the R1 mESCs. Western blot was done once.

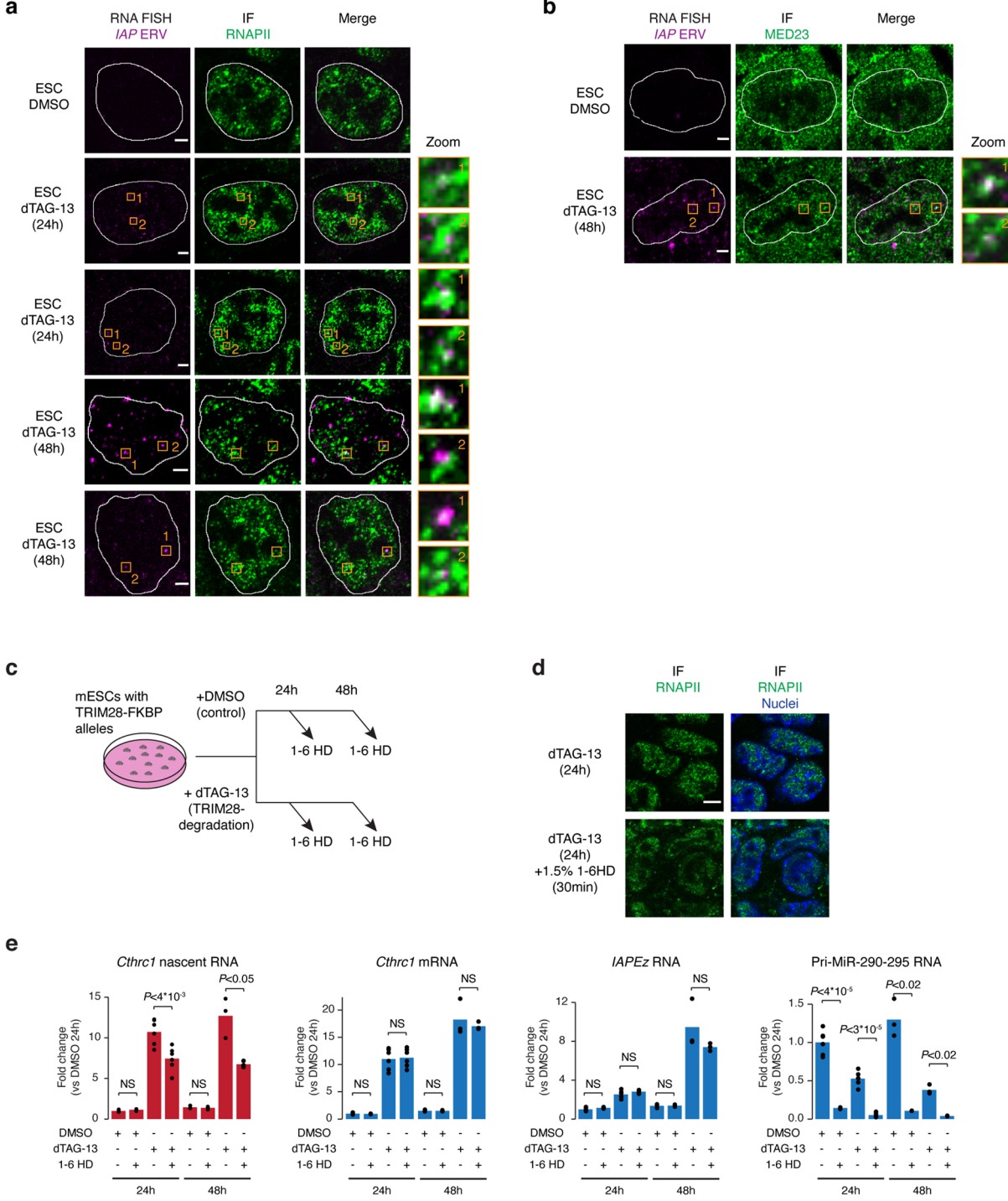

**Extended Data Fig. 3 | Condensate hijacking additional data. a, b**. Co-localization between the IAP RNA and **(a)** RNAPII puncta and **(b)** MED23 puncta in TRIM28-degraded mESCs. Displayed are separate images of the RNA-FISH and IF signal, and an image of the merged channels. The nuclear periphery determined by DAPI staining is highlighted as a white contour. The zoom column displays the region of the images highlighted in a yellow box zoomed in for greater detail. After 24 h dTAG-13 treatment, small nuclear puncta appear, and after 48 h of dTAG-13 treatment, large nuclear foci are visible. Scale bars: 2.5 μm. **c**. Scheme of the 1-6 hexanediol (1-6 HD) treatment experiments. **d**. Representative images of RNAPII immunofluorescence in control and 1-6 HD-treated cells. 1-6 HD partially dissolved the punctate localization of RNAPII. Scale bars: 5 μm. **e**. Transcription of the nascent *Cthrc1* RNA is reduced by 30 min 1% 1-6 hexanediol-treatment in TRIM28-degraded cells. The bar plots show qRT-PCR data as fold change normalized to the DMSO control across 6 and 3 biological replicates for 24 h and 48 h timepoints, respectively. Note that the IAP RNA does not contain introns; thus, the *IAP* RNA qRT-PCR detects the steady state pool of IAP RNAs. Each dot represents a data point, and bar indicates the mean. *P* values are from two-tailed *t* tests. NS: not significant.

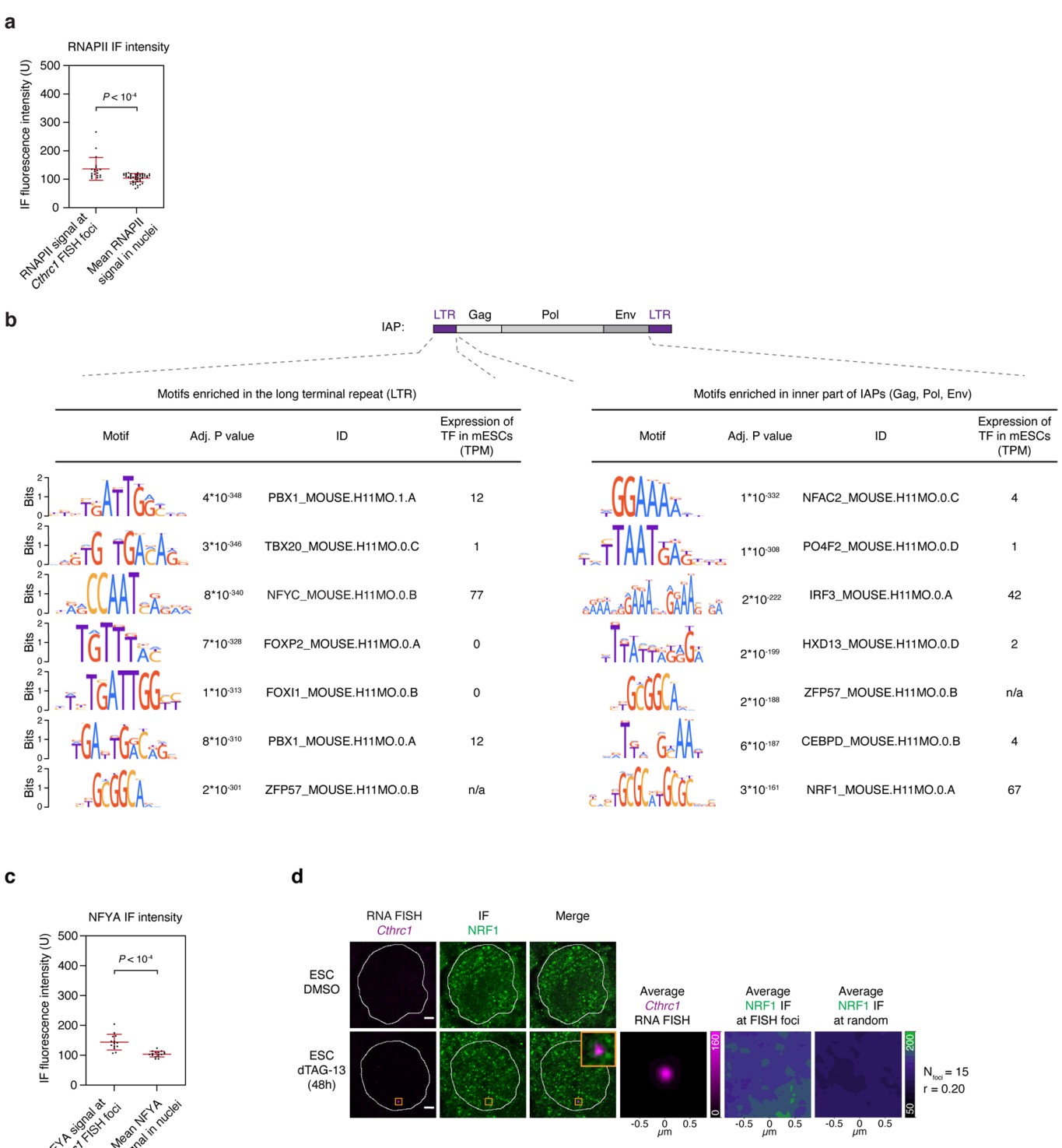

**Extended Data Fig. 4 | Additional characterization NFY and NRF1. a.** The RNAPII IF signal at *Cthrc1* FISH foci is higher than the average nuclear signal. Quantification of the RNAPII IF intensity at *Cthrc1* FISH foci (n = 20) and nuclei (n = 47) in the cells analyzed in Fig. 2f is shown. Data are presented as mean values ± SD from one staining experiment. *P* value is from a two-sided Mann-Whitney test. **b**. The sequence of IAPs is enriched for various TF binding motifs, including the motif of NFY. Top: schematic of an IAP element. Bottom: motif images, adjusted *P* values and motif IDs, and the expression level of the TF in mESC RNA-seq data. Displayed are the top-scoring motifs based on adjusted P-value. Motifs were filtered for redundancy. **c**. The NFYA IF signal at *Cthrc1* FISH foci is higher than the average nuclear signal. Quantification of the NFYA IF intensity at *Cthrc1* FISH foci (n = 14) and nuclei (n = 16) in the cells analyzed in Fig. 2g is shown. Data are presented as mean values ± SD from one staining experiment. *P* value is from a two-sided Mann-Whitney test. **d**. NRF1 puncta do not co-localize with the nascent RNA *Cthrc1* in TRIM28-degraded mESCs. Displayed are separate images of the RNA-FISH and IF signal, and an image of the merged channels. The nuclear periphery determined by DAPI staining is highlighted as a white contour. Also shown are averaged signals of either RNA FISH or NRF1 IF centered on the *Cthrc1* FISH foci or randomly selected nuclear positions. Scale bars: 2.5 μm.

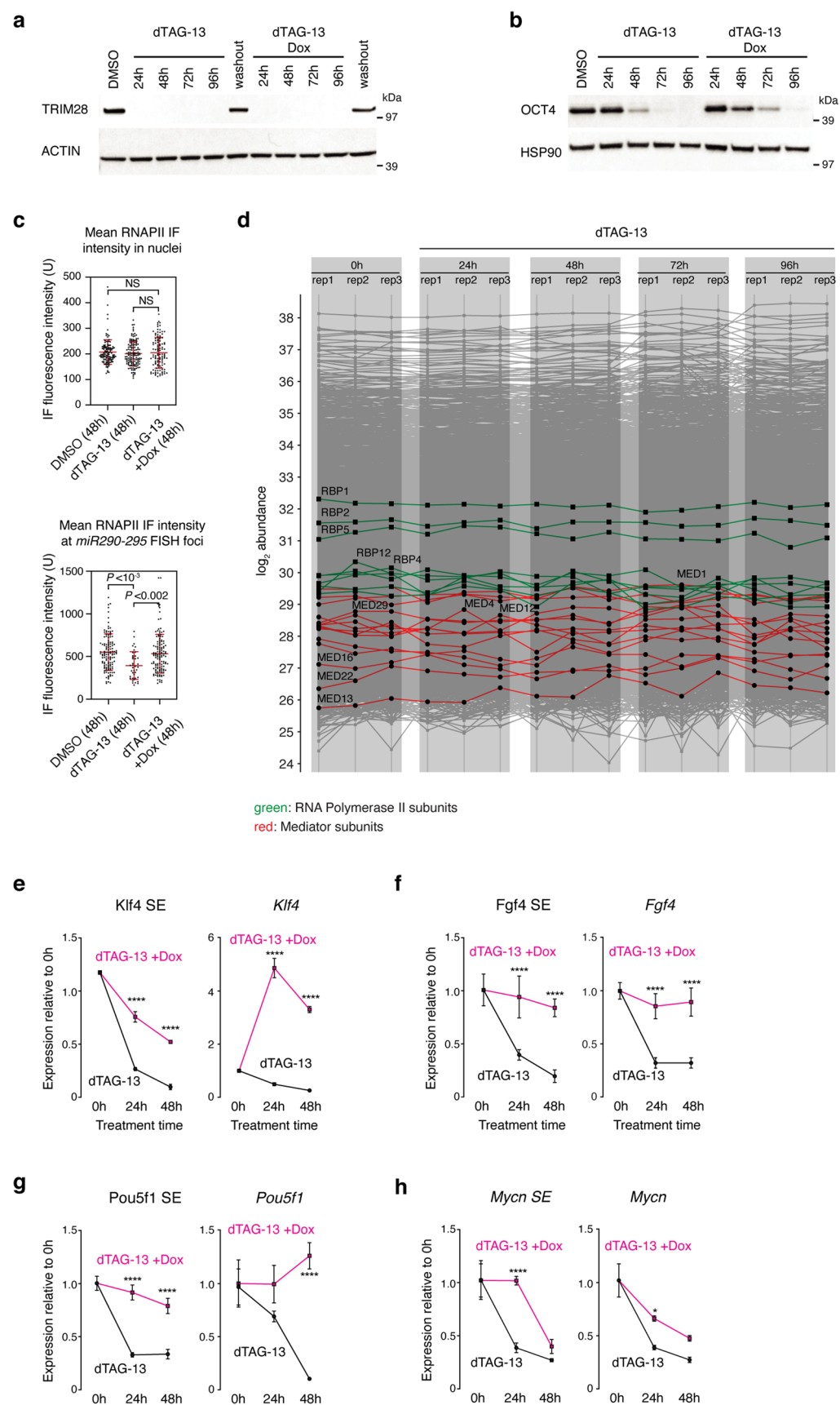

**Extended Data Fig. 5 | See next page for caption.**

**Extended Data Fig. 5 | Additional characterization of the OSKM/dTAG-13 experiments. a**. Western blot validation of the FKBP degron tag and its ability to degrade TRIM28 in iPSCs. Washout of the dTAG-13 ligand (24 h) indicates reversibility of degradation. Western blot experiments were performed twice and one representative image is shown. Actin is shown as the loading control. **b**. Western blot validation of the OSKM ectopic expression in the iPSC line. Western blot experiments were performed three times and one representative image is shown. HSP90 is shown as the loading control. **c**. dTAG-13 treatment leads to reduced RNAPII immunofluorescence signal at *miR290-295* FISH foci which is rescued by OSKM ectopic expression, while overall RNAPII levels do not change. (top) Quantification of RNAPII mean fluorescence intensity (n = 117 for DMSO, n = 138 for dTAG-13, n = 110 for Dox+dTAG-13) in the cells used in Fig. 3f. (bottom) Quantification of RNAPII IF intensities at the *miR290-295* FISH foci (n = 128 for DMSO, n = 39 for dTAG-13, n = 89 for Dox+dTAG-13) detected in the cells used in Fig. 3f. Data are presented as mean values ± SD from one staining experiment. *P* value is from a two-sided Mann-Whitney test. **d**. Mass spectrometry-detected protein abundance for three individual replicate samples after 0 h, 24 h, 48 h, 72 h and 96 h dTAG-13 treatment of mESCs. RNAPII subunits are highlighted in green. Mediator complex subunits are highlighted in red. **e–h**. qRT-PCR data normalized to the 0 h of dTAG-13 treatment. Data are from three independent biological replicates (that is, three wells on a tissue culture plate) and are presented as mean values ± SD. The experiment was repeated three times, and data from one representative experiment are shown. *P* values are from two-tailed *t*-tests. *: $P < 0.05$, ***: $P < 10^{-3}$, ****: $P < 10^{-4}$.

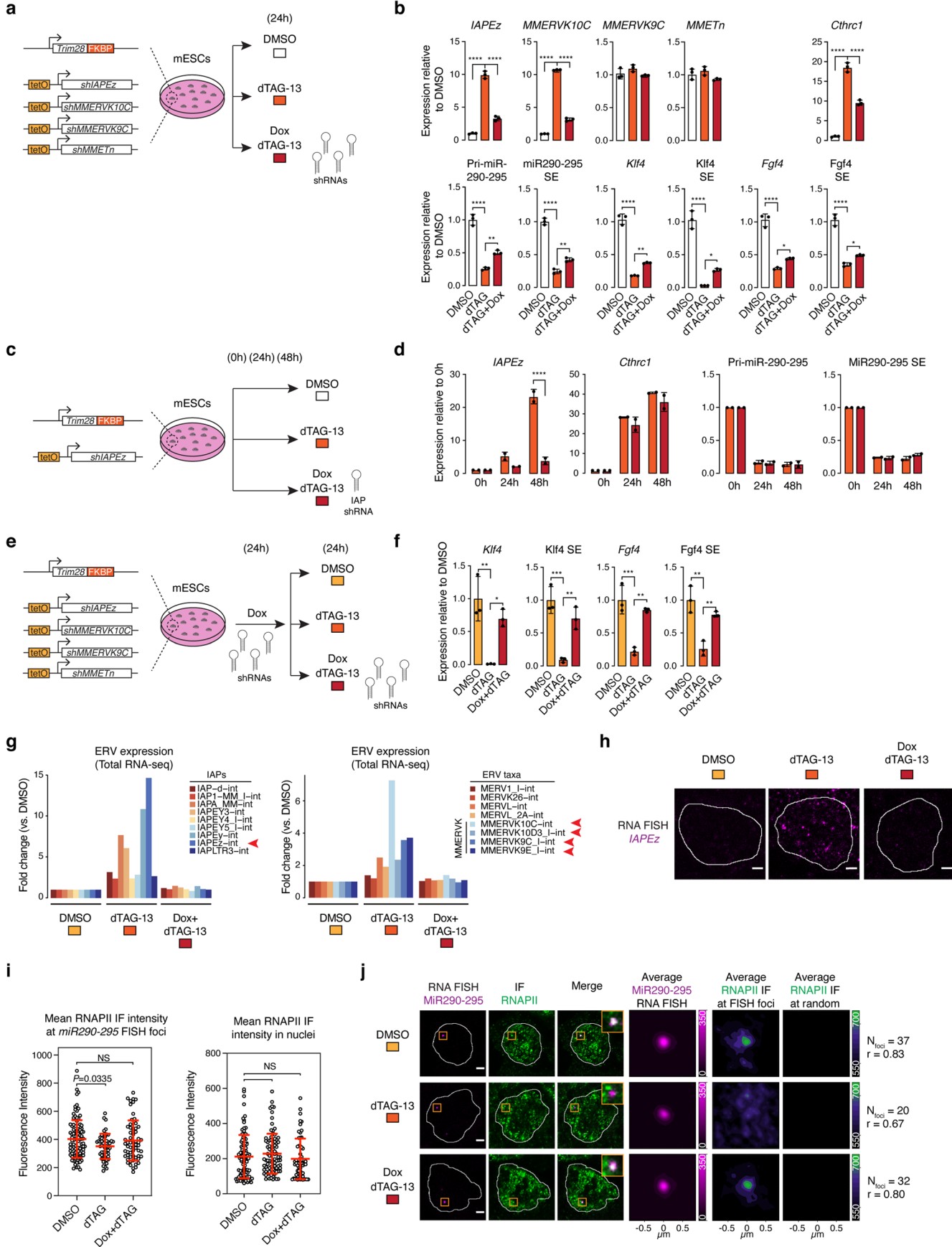

**Extended Data Fig. 6 | Knockdown of ERV RNA rescues super-enhancer transcription in TRIM28-degraded cells. a**. Scheme of knockdown experiments with simultaneous TRIM28 degradation. **b**. qRT-PCR analysis from three independent biological replicates. Data presented as mean values $\pm$ SD. Experiment was performed twice, and the data shown are from one representative experiment. *P* values are from two-tailed *t* tests. ****: $P < 10^{-4}$, ***: $P < 10^{-3}$, **: $P < 10^{-2}$, *: $P < 0.05$. **c**. Scheme of IAP knockdown experiments. **d**. qRT-PCR data displayed as fold change normalized to the 0 h control from three independent biological replicates. Data are presented as mean values $\pm$ SD. Each dot represents the mean of the three biological replicates of an individual experiment. *P* values are from two-tailed *t*-tests. ****: $P < 10^{-4}$, ***: $P < 10^{-3}$, **: $P < 10^{-2}$, *: $P < 0.05$. **e**. Scheme of the experiment in which shRNAs are induced for 24 h and then treated either with DMSO (yellow), dTAG-13 (orange) or with dTAG-13 and Dox (maroon) for additional 24 h. **f**. qRT-PCR data as fold change normalized to the Dox (24 h) treatment control from three independent biological replicates. Data are presented as mean values $\pm$ SD. Experiment was performed twice, and data from one representative experiment is shown. *P* values are from two-tailed *t*-tests. ****: $P < 10^{-4}$, ***: $P < 10^{-3}$, **: $P < 10^{-2}$, *: $P < 0.05$. **g**. RNA levels detected with total RNA-seq. Values from three biological replicates are normalized to levels detected at 0 h. Red arrowheads highlight the ERV taxa against whose sequences the shRNAs were designed. **h**. Representative images of IAP RNA FISH in cells described in panels (**e-f**). Scale bar: 2.5 µm. **i**. Analyses of cells used in Fig. 4d. (left) RNAPII IF intensities at the *miR290-295* FISH foci ($n_{DMSO} = 100$, $n_{dTAG-13} = 52$, $n_{Dox+dTAG-13} = 60$). (right) RNAPII mean fluorescence intensity at random nuclear positions ($n_{DMSO} = 97$, $n_{dTAG-13} = 88$, $n_{Dox+dTAG-13} = 60$). Data presented as mean values $\pm$ SD from one staining experiment. *P* values are from two-sided Mann-Whitney tests. NS: not significant. **j**. Images of individual z-slices (same z) of the RNA-FISH and IF signal. Nuclear periphery determined by DAPI staining is highlighted as a white contour. Also shown are averaged signals of either RNA FISH or RNAPII IF centered on the *miR290-295* FISH foci or randomly selected positions. Scale bars: 2.5 µm. The experiment is an independent biological replicate of the experiments shown in Fig. 4d.

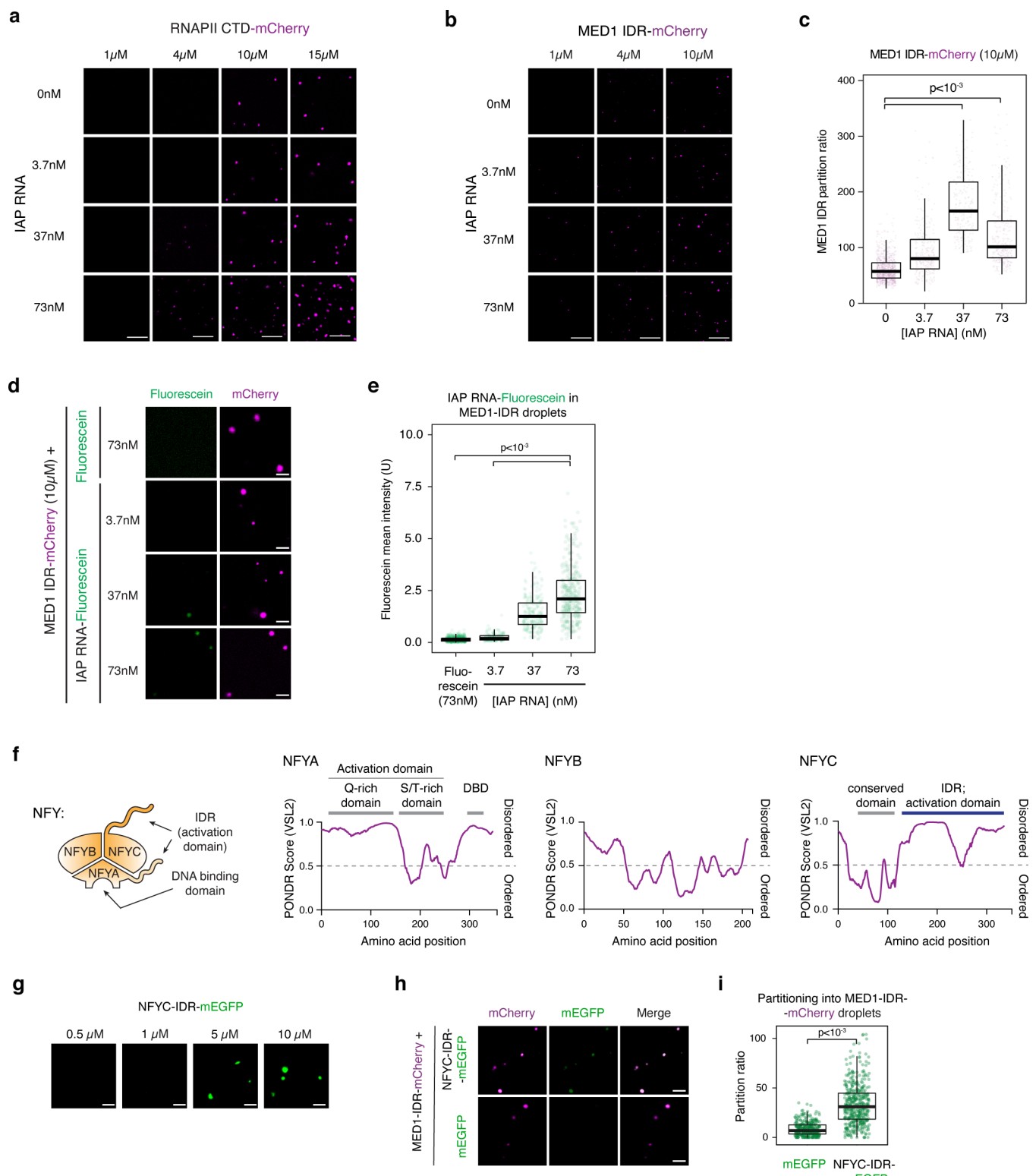

**Extended Data Fig. 7 | See next page for caption.**

**Extended Data Fig. 7 | IAP RNA facilitates droplet formation of transcriptional activators in vitro. a**. IAP RNA facilitates RNAPII CTD droplet formation in vitro. Displayed are representative images of droplet formation by purified RNAPII CTD-mCherry fusion proteins in the presence of in vitro transcribed IAP RNA fragments. Scale bar: 10 μm. **b**. IAP RNA facilitates MED1 IDR droplet formation in vitro. Displayed are representative images of droplet formation by purified MED1 IDR-mCherry fusion proteins in the presence of in vitro-transcribed IAP RNA. **c**. Partitioning ratio of MED1 IDR-mCherry into droplets at the indicated IAP RNA concentrations. Every dot represents a detected droplet. Data for the quantification was acquired from at least five images of two independent image series per condition. *P* value is a from a two-tailed *t* test. **d**. IAP RNA is enriched within MED1 IDR droplets. Displayed are representative images of the enrichment of fluorescein-labeled *IAP* RNA in MED1 IDR-mCherry droplets. **e**. Quantification of the enrichment of fluorescein-labeled IAP RNA in MED1 IDR-mCherry droplets. Data for the quantification was acquired from at least five images of two independent image series per condition. *P* value is from a two-tailed *t* test. **f**. (left) Schematic model of the heterotrimeric NFY transcription factor. (right) Graphs plotting intrinsic disorder in the NFYA, NFYB and NFYC proteins. The NFYC-IDR cloned for subsequent experiments is highlighted with a blue bar. **g**. Concentration-dependent droplet formation by purified recombinant NFYC IDR-mEGFP. Scale bar: 10 μm. **h**. Representative images of droplet formation by purified NFYC IDR-mEGFP and MED1 IDR-mCherry fusion proteins. MED1 IDR-mCherry mixed with purified mGFP is included as a control. Scale bar: 10 μm. **i**. Partitioning ratio of NFYC IDR-mEGFP or mEGFP in MED1 IDR-mCherry droplets. Every dot represents a detected droplet. Data for the quantification was acquired from at least five images of two independent image series per condition. *P* value is a from a two-tailed *t* test.

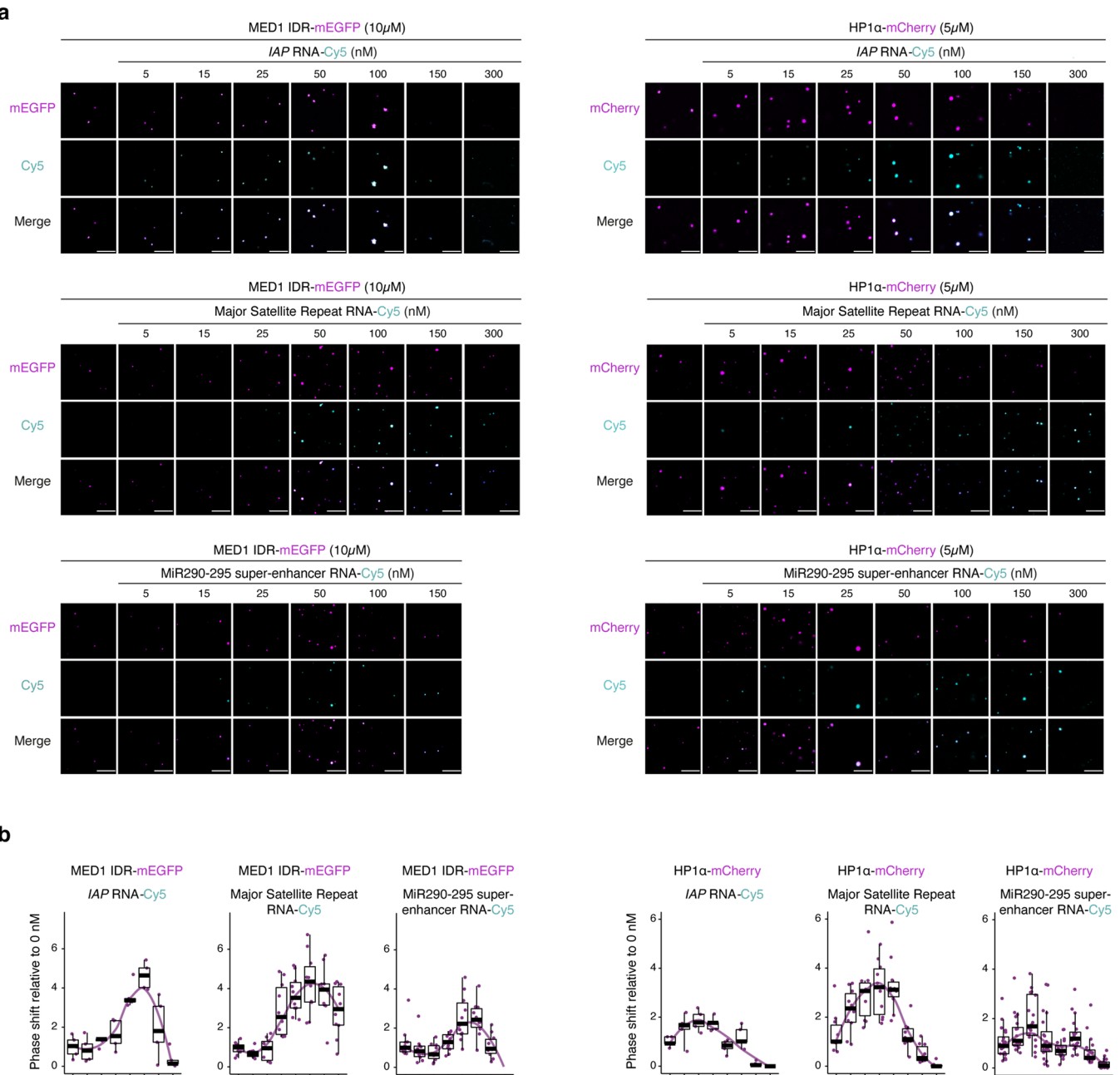

**Extended Data Fig. 8 | Effects of various RNAs on MED1 IDR and HPα droplets in vitro. a**. RNA facilitates MED IDR and droplet HPα formation in vitro. Displayed are representative images of droplet formation by purified (left) MED1 IDR-mEGFP fusion protein and (right) HPα-mCherry fusion protein in the presence of in vitro transcribed RNA fragments. Scale bar: 10 μm. **b**. Quantification of the partitioning of (left) MED1 IDR and (right) HPIα into droplets in the presence of the indicated RNA species. Values are normalized against the partition ratio at no RNA added. Data for the quantification was acquired from at least five images of two independent image series per condition. IAP RNA quantification is the same plot displayed in Fig. 4h.

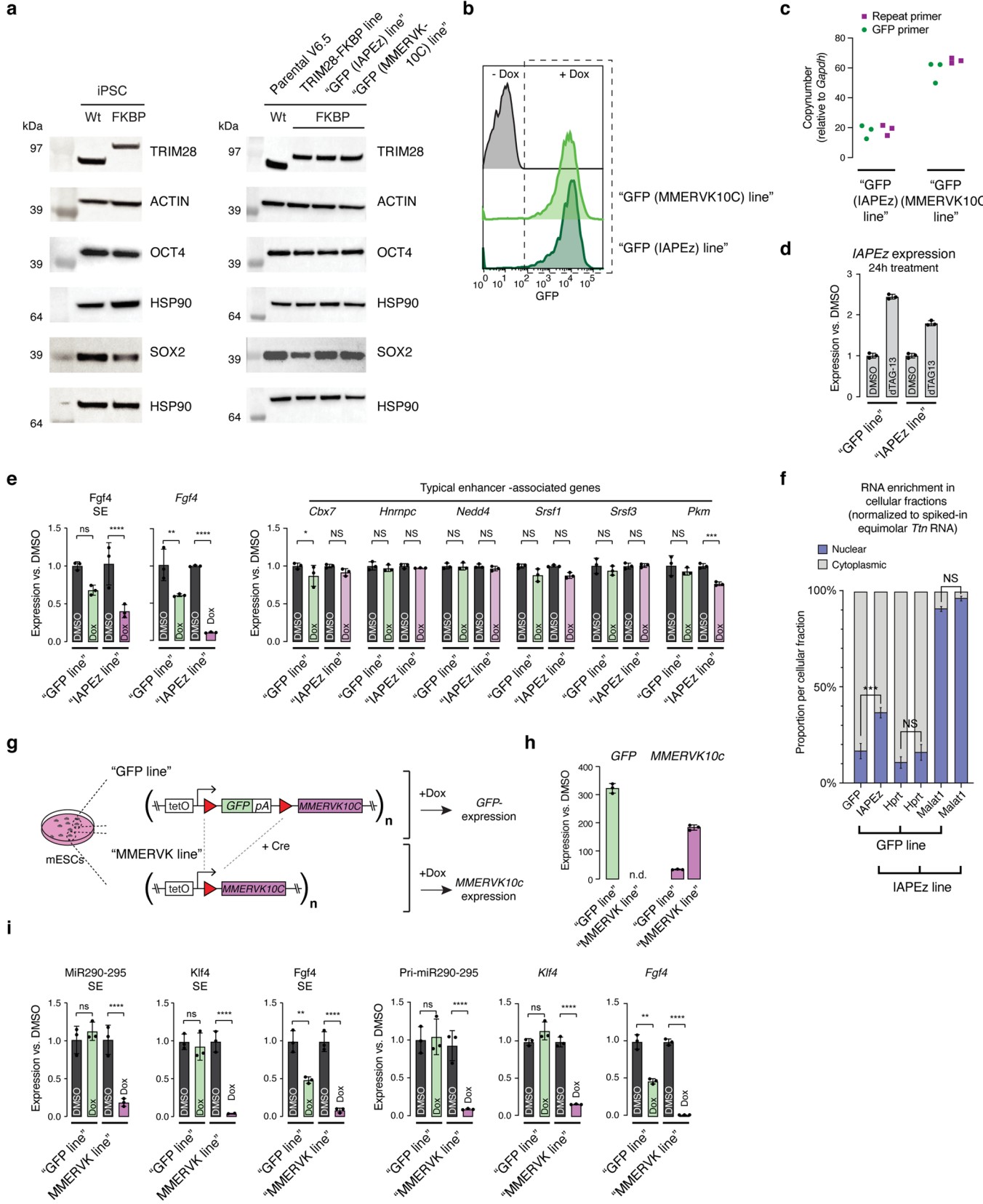

**Extended Data Fig. 9 | See next page for caption.**

**Extended Data Fig. 9 | Induction of ERV transcription compromises super-enhancer transcription. a**. Western blot of TRIM28, OCT4 and SOX2 in the indicated cell lines. Western blot experiments were performed once. **b**. FACS analysis of GFP in the 'GFP (IAPEz) line' and 'GFP (MMERVK10C) line'. **c**. Genotyping qRT-PCR of the 'GFP (IAPEz) line' and 'GFP (MMERVK10C) line.' Primer sets amplifying the transgenic GFP or repeat sequence (IAP or MMERVK10C) were used. Data are from triplicate experiments. **d**. qRT-PCR validation of IAPEz upregulation. Values are normalized against the IAPEz level in the corresponding DMSO condition. Data are presented as mean values ± SD from three biological replicates. **e**. Additional supporting data for the experiment in Fig. 4m. qRT-PCR data are shown as fold change normalized to the DMSO control treatment. Data are presented as mean values ± SD from three biological replicates. *P* values are from two-tailed *t*-tests. ****: $P < 10^{-4}$, ***: $P < 10^{-3}$, **: $P < 10^{-2}$, NS: not significant. **f**. The amount of *GFP*, *IAPEz*, *Hprt* and *Malat1* RNA were quantified in the nuclear and cytoplasmic fractions by qRT-PCR. The values are normalized against the amount of in vitro transcribed *Ttn* RNA that was spiked in at equimolar amount to the cytoplasmic and nuclear fractions. The expression values are then displayed as the percentage of the RNA in the nuclear and cytoplasmic fractions. *Hprt* is used as a control of a cytoplasmic mRNA, and *Malat1* is used as control RNA known to be enriched in the nucleus. Data are presented as mean values ± SD from three biological replicates. *P* values are from a two-way ANOVA Sidak's multiple comparison test. ***: $P < 10^{-3}$, NS: not significant. **g**. Schematic model of the experiment mimicking *MMERVK10C* transcription. mESC lines harboring ~61 copies of a PiggyBac transposon encoding a Dox-inducible GFP or MMERVK10C transgene were treated with Dox (to induce GFP or MMERVK10C expression). **h**. qRT-PCR validation of the 'GFP line' and 'MMERVK line'. The bar plots show qRT-PCR data as fold change normalized to the DMSO control treatment. Data are presented as mean values ± SD from three biological replicates. N.d.: not detectable. **i**. qRT-PCR data as fold change normalized to the DMSO control treatment. Data are presented as mean values ± SD from three biological replicates. *P* values are from two-tailed *t*-tests. ****: $P < 10^{-4}$, ***: $P < 10^{-3}$, **: $P < 10^{-2}$, NS: not significant.

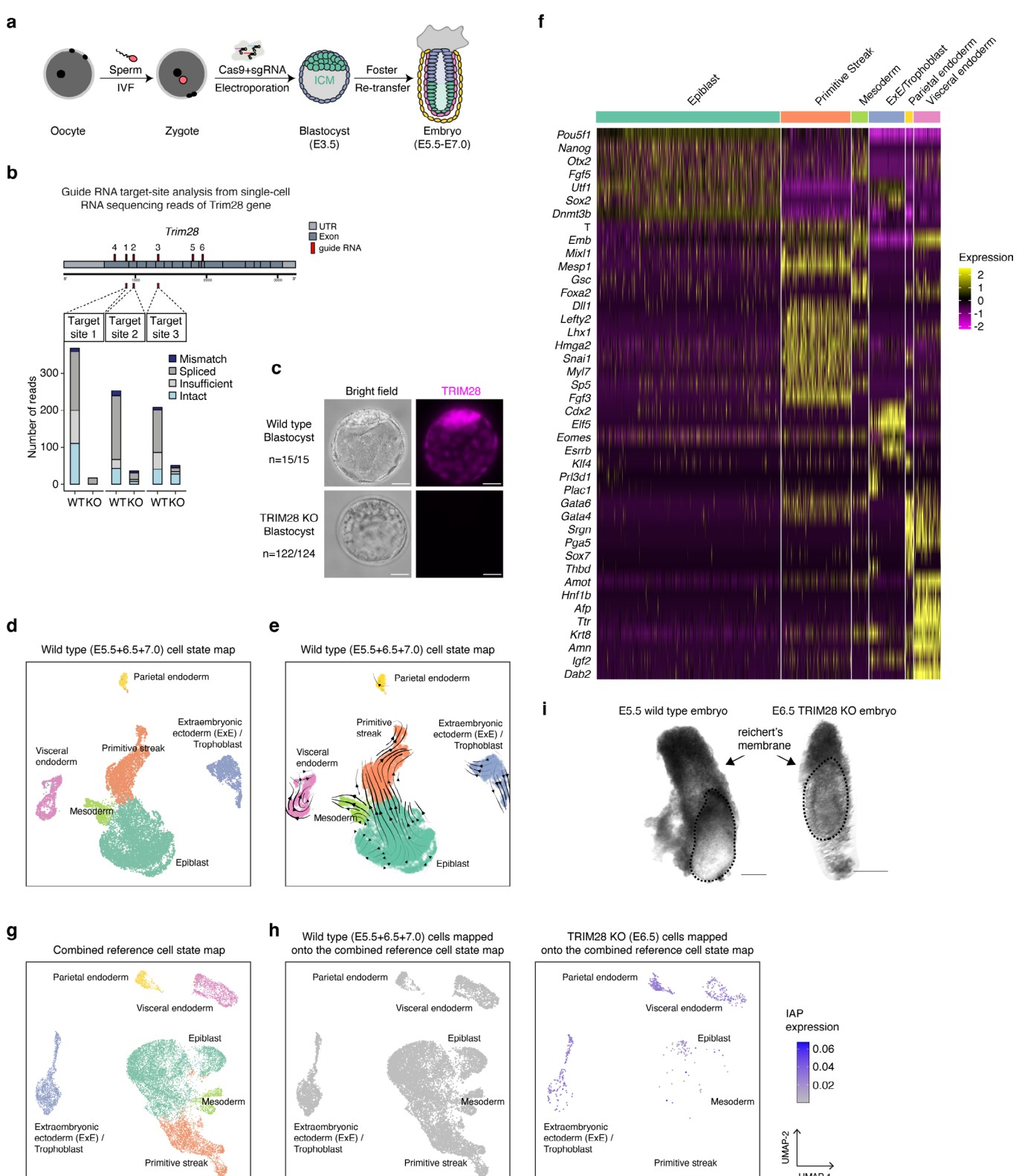

**Extended Data Fig. 10 | See next page for caption.**

**Extended Data Fig. 10 | Reference cell state maps in early mouse embryos. a**. Scheme of the zygotic CRISPR/Cas9 – scRNA-seq platform. **b**. Cut site analysis. The number, type and distribution of reads mapping to the sites targeted with the TRIM28 guide RNAs is quantified in the scRNA-seq data. **c**. Immunofluorescence verification of TRIM28 KO. Representative images are shown, with the number of embryos where the immunofluorescence confirmed the genotype per the total number of embryos analyzed. The knockout experiment was performed five times independently, and the pool of embryos from all experiments were used for staining. Scale bars: 20 μm. **d**. UMAP of wild type early mouse embryos spanning E5.5-E7.0 developmental window. The wild type cells used in scRNA-seq experiments from E5.5, E6.5 and E7.0 developmental stages were included. **e**. RNA velocity map of wild type early mouse embryos spanning E5.5-E7.0 developmental window. **f**. Heatmap representation of the expression levels of marker genes of each cluster/cell state. **g**. Combined cell state map. The wild type cells and TRIM28 KO cells used in scRNA-seq experiments were included. **h**. Elevated IAP expression in TRIM28 KO cells. (left) Wild type cells used in scRNA-seq experiments from E5.5, E6.5 and E7.0 developmental stages are projected on the combined reference map and are colored according to the *IAP* expression of the corresponding embryo and cell state. (right) TRIM28 KO E6.5 cells are projected on the combined reference map and are colored according to *IAP* expression of the corresponding embryo and cell state. **i**. Bright-field images of representative embryos from E5.5 wild type and E6.5 TRIM28 KO. Dotted lines represent the embryo that was dissected out from the dense Reichert's membrane. Scale bar is 100 μm.

# nature research

# Reporting Summary

Nature Research wishes to improve the reproducibility of the work that we publish. This form provides structure for consistency and transparency in reporting. For further information on Nature Research policies, see our Editorial Policies and the Editorial Policy Checklist.

## Statistics

For all statistical analyses, confirm that the following items are present in the figure legend, table legend, main text, or Methods section.

| n/a | Confirmed | |
|---|---|---|
| ☐ | ☒ | The exact sample size ($n$) for each experimental group/condition, given as a discrete number and unit of measurement |
| ☒ | ☐ | A statement on whether measurements were taken from distinct samples or whether the same sample was measured repeatedly |
| ☐ | ☒ | The statistical test(s) used AND whether they are one- or two-sided<br>*Only common tests should be described solely by name; describe more complex techniques in the Methods section.* |
| ☒ | ☐ | A description of all covariates tested |
| ☐ | ☒ | A description of any assumptions or corrections, such as tests of normality and adjustment for multiple comparisons |
| ☐ | ☒ | A full description of the statistical parameters including central tendency (e.g. means) or other basic estimates (e.g. regression coefficient) AND variation (e.g. standard deviation) or associated estimates of uncertainty (e.g. confidence intervals) |
| ☐ | ☒ | For null hypothesis testing, the test statistic (e.g. $F$, $t$, $r$) with confidence intervals, effect sizes, degrees of freedom and $P$ value noted<br>*Give P values as exact values whenever suitable.* |
| ☒ | ☐ | For Bayesian analysis, information on the choice of priors and Markov chain Monte Carlo settings |
| ☒ | ☐ | For hierarchical and complex designs, identification of the appropriate level for tests and full reporting of outcomes |
| ☐ | ☒ | Estimates of effect sizes (e.g. Cohen's $d$, Pearson's $r$), indicating how they were calculated |

*Our web collection on statistics for biologists contains articles on many of the points above.*

## Software and code

Policy information about availability of computer code

| Data collection | - Fluorescence images were collected with widefield and confocal microscopes using Zen Blue (2.3.69.1016) and Zen (black version) software (Zeiss)<br><br>- Western blot images were collected using Image Lab software (version 6.1.0 build 7) (Bio-Rad)<br><br>- Single-cell data was collected with 10X Cell Genomics Chromium System v2.0 chemistry (10X Genomics)<br><br>- FACS data was collected with FACS Diva software (BD Biosciences) and BD Aria II and BD Celesta instruments |
|---|---|
| Data analysis | - Fluorescence images were analyzed using:<br><br> Zeiss Zen Blue (2.3.69.1016)<br><br>Zeiss Zen (black version)<br><br>Fiji/ImageJ (2.1.0/1.53i)<br><br>Published MATLAB scripts (Sabari et al., 2018) were used for the analysis of RNA FISH with immunofluorescence. Custom python code was generated to curate random nuclear foci<br><br>- Fiji/ImageJ colocalization plugins (Bolte & Cordelieres, 2006, Gilles et al., 2017) were used for IAP FISH-IF analysis<br><br>- FACS data was analyzed with FlowJo (v10.7) |

- GraphPad Prism (v 9.2.0) was used for statistical analysis and barplot generation.

- PALM images were reconstructed and analyzed using MTT (Sergé et al., 2008) and qSR (Andrews et al., 2018).

- For NGS data analyses, the following softwares were used:

Seurat (version 3.2.3)

cutadapt (version 2.4)

STAR (version 2.7.5a)

StringTie (version 2.0.6)

Seqtk (version 1.3-r106)

SLAM-DUNK (version 0.4.1)

DeppTools bamCoverage (version 3.4.3)

bwa mem (version 0.7.17)

samtools (version 1.10)

gatk (version 4.1.4.1)

Spp (version 1.2.2)

MACS2 (version 2.1.2)

HOMER (version 4.10)

Treeview (version 3.0)

Ame (version 5.3.0)

Cell Ranger (version 4)

velocyto (version 0.1.18)

scanpy (version 1.4.3)

pheatmap (version 1.0.12)

Coolpup.py (version 0.9.5)

All software versions and parameters are listed in the Methods.

Custom code available under : https://doi.org/10.5281/zenodo.6521914

For manuscripts utilizing custom algorithms or software that are central to the research but not yet described in published literature, software must be made available to editors and reviewers. We strongly encourage code deposition in a community repository (e.g. GitHub). See the Nature Research guidelines for submitting code & software for further information.

## Data

Policy information about availability of data

All manuscripts must include a data availability statement. This statement should provide the following information, where applicable:
- Accession codes, unique identifiers, or web links for publicly available datasets
- A list of figures that have associated raw data
- A description of any restrictions on data availability

All data is available in the supplementary materials. Sequence data were deposited at GEO under the accession number GSE159468. Mass spectrometry data were deposited at ProteomeXchange under the accession ID PDX021895. Plasmids generated in the study are available at Addgene.

# Field-specific reporting

Please select the one below that is the best fit for your research. If you are not sure, read the appropriate sections before making your selection.

☒ Life sciences          ☐ Behavioural & social sciences          ☐ Ecological, evolutionary & environmental sciences

For a reference copy of the document with all sections, see nature.com/documents/nr-reporting-summary-flat.pdf

# Life sciences study design

All studies must disclose on these points even when the disclosure is negative.

| | |
|---|---|
| Sample size | No statistical methods were used to predetermine sample sizes. Sample sizes for qPCR, Mass spectrometry, RNA-FISH/IF, IF, FACS, ChIP-seq, RNA-seq, SLAM-TT-Seq, and other sequencing technologies are consistent with current standards for sample sizes included controls in the published literature. Sample sizes are indicated in the figure panels or legends or in the Methods. For droplet experiments we imaged at least 10 independent fields of a view for each experimental condition based on current methodology in the field (Sabari et al., 2018; Boija et al., 2018).For RNA FISH with IF we imaged at least two fields of view (yielding 20+ nuclei and the number of detected foci indicated in figure panels) for each condition based on current practices in the field (Sabari et al., 2018, Boija et al., 2018). |
| Data exclusions | In rare instances, out of focus images were excluded in FISH-IF experiments. |
| Replication | All results obtained and reported in this study were reproducible across the replicates examined. The number of replicates per experiment has been described in the figures, legends, tables, main text or supplementary materials. |
| Randomization | Not relevant in this study. Samples were allocated as either wildtype/control group or chemical/genetic treatment group. Cell culture samples for every experiment were collected without a preconceived selection strategy. Embryos were also collected without a preconceived selection strategy or priority. All experiments were performed with appropriate controls. Controls are included in the individual figure panels. |
| Blinding | Blinding was not relevant for the experiments. However, our analytical pipeline for each experiment followed uniform criteria applied to all samples, allowing us to analyze our data in an unbiased manner. |

# Reporting for specific materials, systems and methods

We require information from authors about some types of materials, experimental systems and methods used in many studies. Here, indicate whether each material, system or method listed is relevant to your study. If you are not sure if a list item applies to your research, read the appropriate section before selecting a response.

## Materials & experimental systems

| n/a | Involved in the study |
|---|---|
| ☐ | ☒ Antibodies |
| ☐ | ☒ Eukaryotic cell lines |
| ☒ | ☐ Palaeontology and archaeology |
| ☐ | ☒ Animals and other organisms |
| ☒ | ☐ Human research participants |
| ☒ | ☐ Clinical data |
| ☒ | ☐ Dual use research of concern |

## Methods

| n/a | Involved in the study |
|---|---|
| ☐ | ☒ ChIP-seq |
| ☐ | ☒ Flow cytometry |
| ☒ | ☐ MRI-based neuroimaging |

## Antibodies

| | |
|---|---|
| Antibodies used | - Immunofluorescence experiments:<br><br>TRIM28 (ab22553; 1:200, Abcam)<br>OCT4 (ab19857; 1:200, Abcam)<br>NANOG (REC-RCAB002P-F; 1:400, CosmoBio)<br>KLF4 (AF3158; 1:200, R&D Systems)<br>SOX2 (ab79351; 1:200, Abcam)<br>GATA6 (AF1700; 1:200, R&D Systems)<br>GATA4 (sc-25310; 1:200, Santa Cruz)<br>SOX17 (AF1924; 1:200, R&D Systems)<br>SSEA-1 antibody (BioLegend 125609/125608; 1:1000)<br>IAP-GAG (MBS8566075; 1:100, MyBioSource)<br>MED1 (IF: Abcam ab64965 lot GR3326781-1)<br>MED23 (IF: Bethyl A300-425A lot no. 1)<br>Pol II-CTD (IF: Abcam ab817 lot GR3216482-4)<br>NRF1 (IF: Abcam ab55744 lot GR3334322-3)<br>NFYA (IF: Santa Cruz sc-17753 X lot F0916)<br>Goat anti-Rabbit-Alexa 488 (Thermo A11008 lot 2179202)<br>Goat anti-Mouse-Alexa 488 (Thermo A11001 lot 2189178)<br>Donkey anti-Rabbit-Alexa 488 (A21206, Invitrogen)<br>Donkey anti-Rabbit-Alexa647 (711-605-152, Jackson Immuno)<br>Donkey anti-Mouse-Alexa 488 (A21202, Invitrogen)<br>Donkey anti-Mouse-Alexa 647 (715-605-150, Jackson Immuno)<br>Donkey anti-Goat-Alexa 594 (A11058, Invitrogen)<br>Donkey anti-Goat-Alexa 647 (705-605-147, Jackson Immuno) |

-Western blotting experiments:

TRIM28 (ab22553; 1:500, Abcam)
ACTB (ab8226; 1:1000, Abcam)
OCT4 (ab19857; 1:500, Abcam)
OCT4 (sc-5279; 1:500, Santa Cruz)
HSP90 (BD610419; 1:4000, BD Biosceinces)
SOX2 (ab79351; 1:500, Abcam)
HRP secondary (Jackson Immuno #115-035-174, 211-032-171)

- FACS experiments:

SSEA-1 antibody (BioLegend 125609/125608; 1:1000)
PDGFRA (BioLegend 135923; 1:1000)

-ChIP experiments:

Spike-in antibody (61686, Active Motif)
H3K9me3 (ab8898, Abcam)
H3K27ac (ab4729, Abcam)
Pol II-CTD (8WG16, BioLegend)
MED23 (A300-425A, Bethyl Lab)

Validation

Antibodies used in immunofluorescence experiments were validated by comparing WT mESCs and mouse embryos.
Antibody validation for IAP-GAG was performed with wild type 2C-8C stage mouse embryos.

All antibodies are validated by the provider and cited in numerous publications.

-Western blotting experiments:

TRIM28 (ab22553; 1:500, Abcam) - mouse
https://www.abcam.com/kap1-antibody-20c1-ab22553.html

ACTB (ab8226; 1:1000, Abcam) - mouse
https://www.abcam.com/beta-actin-antibody-mabcam-8226-loading-control-ab8226.html

OCT4 (ab19857; 1:500, Abcam) - mouse
https://www.abcam.com/oct4-antibody-ab19857.html

OCT4 (sc-5279; 1:500, Santa Cruz) - mouse
https://www.scbt.com/p/oct-3-4-antibody-c-10

HSP90 (BD610419; 1:4000, BD Biosceinces) - mouse
https://www.fishersci.com/shop/products/anti-hsp90-clone-68-bd-2/BDB610419

SOX2 (ab79351; 1:500, Abcam) - mouse
https://www.abcam.com/sox2-antibody-9-9-3-ab79351.html

- FACS experiments:

SSEA-1 antibody (BioLegend 125609/125608; 1:1000) - mouse
https://www.biolegend.com/ja-jp/products/alexa-fluor-488-anti-mouse-human-cd15-ssea-1-antibody-4820
https://www.biolegend.com/ja-jp/products/alexa-fluor-647-anti-mouse-human-cd15-ssea-1-antibody-4819

PDGFRA (BioLegend 135923; 1:1000) - mouse
https://www.biolegend.com/ja-jp/products/brilliant-violet-421-anti-mouse-cd140a-antibody-17921

-ChIP experiments:

Spike-in antibody (61686, Active Motif)
https://www.activemotif.com/catalog/1091/chip-normalization

H3K9me3 (ab8898, Abcam) - mouse
https://www.abcam.com/histone-h3-tri-methyl-k9-antibody-chip-grade-ab8898.html

H3K27ac (ab4729, Abcam) - mouse
https://www.abcam.com/histone-h3-acetyl-k27-antibody-chip-grade-ab4729.html

Pol II-CTD (8WG16, BioLegend) - mouse
https://www.biolegend.com/en-us/punchout/search-results/purified-anti-rna-polymerase-ii-rpb1-antibody-11666

MED23 (A300-425A, Bethyl Lab) - mouse
https://www.fortislife.com/products/primary-antibodies/rabbit-anti-crsp3-antibody/BETHYL-A300-423

# Eukaryotic cell lines

Policy information about cell lines

| | |
|---|---|
| Cell line source(s) | - V6.5 mouse embryonic stem cells (mESCs), source: Konrad Hochedlinger lab<br><br>- HEK293T, source: ATCC, Identifier: CRL-3216<br><br>- Trim28-FKBP mESC, source: This paper (see Methods "Generation of the TRIM28-FKBP ESC line")<br><br>- Trim28-FKBP mouse iPSC, source: This paper (see Methods "Generation of the TRIM28-FKBP iPSC line")<br><br>- Trim28-FKBP mouse iPSC, inactivation of GFP (Nanog::GFP), source: This paper (see Methods "Inactivation of NANOG::GFP in the TRIM28-FKBP iPSC line")<br><br>- Trim28-FKBP mESCs IAPEz knockdown, source: This paper (see Methods "Generation of shRNA knockdown mESC line for IAPEz and IAPEz/MMERVK10c/ /MMERVK9c/ MMETn")<br><br>- Trim28-FKBP mESCs ERV knockdown, source: This paper (see Methods "Generation of shRNA knockdown mESC line for IAPEz and IAPEz/MMERVK10c/ /MMERVK9c/ MMETn")<br><br>- Trim28-FKBP mESCs ERV-TKO, cource: This paper (see Methods "Deletion of ERVs in the TRIM28-FKBP mESC line")<br><br>- mESCs inducible ERVs, source: This paper (see Methods "Integration of PiggyBac transposon encoding Dox-inducible ERVs")<br><br>- Trim28-FKBP mEpiSCs, source: This paper (see Methods "Differentiation of TRIM28-FKBP mESCs to EpiSCs (epiblast stem cells)")<br><br>- Trim28-FKBP mNPCs, source: This paper (see Methods "Differentiation of TRIM28-FKBP mESCs to iXEN (induced extra-embryonic stem cells)", "Differentiation of TRIM28-FKBP mESCs to NPCs (neural progenitor cells)")<br><br>- Trim28-FKBP iXEN, source: This paper (see Methods "Differentiation of TRIM28-FKBP mESCs to iXEN (induced extra-embryonic stem cells)") |
| Authentication | The identity of HEK293T, parental V6.5 mESCs and iPSCs, and all cell lines derived from them has been validated using morphological characteristics, qPCRs, FACS, immunofluorescence, RNA-seq, and marker gene expression (where applicable), but have not been authenticated. |
| Mycoplasma contamination | All cell lines are negative for mycoplasma contamination and were regularly tested throughout the study. |
| Commonly misidentified lines<br>(See ICLAC register) | No commonly misidentified cell lines were used in this study. |

# Animals and other organisms

Policy information about studies involving animals; ARRIVE guidelines recommended for reporting animal research

| | |
|---|---|
| Laboratory animals | Oocytes were isolated from B6D2F1 strain female mice (age 7 to 9 weeks, Envigo), sperm was isolated from B6/CAST F1 male mice (>2months of age) which were generated by breeding C57BL/6J strain female mice with CAST/EiJ strain males. Blastocysts were transferred into Hsd:ICR (CD-1) strain female mice of age 9-12 weeks (21-25g, Envigo) which had been mated with Vasectomized SW strain males of >1 age (Envigo). All mice were kept under SPF-conditions in individually ventilated cages at a temperature of 22 +/- 2 degree celcius and a humidity of 55% +/- 10% with a 12hr light/dark cycle (6am-6pm). |
| Wild animals | Wild animals were not involved in this study. |
| Field-collected samples | No samples were collected from the field. |
| Ethics oversight | All procedures follow strict animal welfare guidelines as approved by the Max Planck Institute for Molecular Genetics (G0247/13-SGr1) and LAGeSO Berlin; Harvard University license numbers IACUC protocol (28-21). |

Note that full information on the approval of the study protocol must also be provided in the manuscript.

# ChIP-seq

## Data deposition

☒ Confirm that both raw and final processed data have been deposited in a public database such as GEO.

☒ Confirm that you have deposited or provided access to graph files (e.g. BED files) for the called peaks.

| | |
|---|---|
| Data access links<br>*May remain private before publication.* | Datasets generated in this study have been deposited in the Gene Expression Omnibus under accession number GSE159468 |

Files in database submission

PROCESSED DATA FILES

H3K27Ac_ESC_DMSO_rep1.bw
H3K27Ac_ESC_DMSO_rep2.bw
H3K27Ac_ESC_DMSO_rep3.bw
H3K27Ac_ESC_dTAG_24h_rep1.bw
H3K27Ac_ESC_dTAG_24h_rep2.bw
H3K27Ac_ESC_dTAG_24h_rep3.bw
H3K9me3_ESC_DMSO_rep1.bw
H3K9me3_ESC_DMSO_rep2.bw
H3K9me3_ESC_DMSO_rep3.bw
H3K9me3_ESC_dTAG_24h_rep1.bw
H3K9me3_ESC_dTAG_24h_rep2.bw
H3K9me3_ESC_dTAG_24h_rep3.bw
input_ESC_DMSO_rep1.bw
input_ESC_DMSO_rep2.bw
input_ESC_DMSO_rep3.bw
input_ESC_dTAG_24h_rep1.bw
input_ESC_dTAG_24h_rep2.bw
input_ESC_dTAG_24h_rep3.bw
Pol2_ESC_DMSO_rep1.bw
Pol2_ESC_DMSO_rep2.bw
Pol2_ESC_dTAG_24h_rep1.bw
Pol2_ESC_dTAG_24h_rep2.bw
Med23_ESC_DMSO_rep1.bw
Med23_ESC_DMSO_rep2.bw
Med23_ESC_dTAG_24h_rep1.bw
Med23_ESC_dTAG_24h_rep2.bw
input_ESC_DMSO_Pol2.bw
input_ESC_dTAG_24h_Pol2.bw
input_ESC_DMSO_Med23.bw
input_ESC_dTAG_24h_Med23.bw
H3K27Ac_ESC_DMSO_rep1_narrowPeak.bed.gz
H3K27Ac_ESC_DMSO_rep2_narrowPeak.bed.gz
H3K27Ac_ESC_DMSO_rep3_narrowPeak.bed.gz
H3K27Ac_ESC_dTAG_24h_rep1_narrowPeak.bed.gz
H3K27Ac_ESC_dTAG_24h_rep2_narrowPeak.bed.gz
H3K27Ac_ESC_dTAG_24h_rep3_narrowPeak.bed.gz
H3K9me3_ESC_DMSO_rep1_broadPeak.bed.gz
H3K9me3_ESC_DMSO_rep2_broadPeak.bed.gz
H3K9me3_ESC_DMSO_rep3_broadPeak.bed.gz
H3K9me3_ESC_dTAG_24h_rep1_broadPeak.bed.gz
H3K9me3_ESC_dTAG_24h_rep2_broadPeak.bed.gz
H3K9me3_ESC_dTAG_24h_rep3_broadPeak.bed.gz
Pol2_ESC_DMSO_rep1_narrowPeak.bed.gz
Pol2_ESC_DMSO_rep2_narrowPeak.bed.gz
Pol2_ESC_dTAG_24h_rep1_narrowPeak.bed.gz
Pol2_ESC_dTAG_24h_rep2_narrowPeak.bed.gz
Med23_ESC_DMSO_rep1_narrowPeak.bed.gz
Med23_ESC_DMSO_rep2_narrowPeak.bed.gz
Med23_ESC_dTAG_24h_rep1_narrowPeak.bed.gz
Med23_ESC_dTAG_24h_rep2_narrowPeak.bed.gz

RAW DATA FILES
H3K27Ac_ESC_DMSO_rep1_R1.fastq.gz
H3K27Ac_ESC_DMSO_rep2_R1.fastq.gz
H3K27Ac_ESC_DMSO_rep3_R1.fastq.gz
H3K27Ac_ESC_dTAG_24h_rep1_R1.fastq.gz
H3K27Ac_ESC_dTAG_24h_rep2_R1.fastq.gz
H3K27Ac_ESC_dTAG_24h_rep3_R1.fastq.gz
H3K9me3_ESC_DMSO_rep1_R1.fastq.gz
H3K9me3_ESC_DMSO_rep2_R1.fastq.gz
H3K9me3_ESC_DMSO_rep3_R1.fastq.gz
H3K9me3_ESC_dTAG_24h_rep1_R1.fastq.gz
H3K9me3_ESC_dTAG_24h_rep2_R1.fastq.gz
H3K9me3_ESC_dTAG_24h_rep3_R1.fastq.gz
input_ESC_DMSO_rep1_R1.fastq.gz
input_ESC_DMSO_rep2_R1.fastq.gz
input_ESC_DMSO_rep3_R1.fastq.gz
input_ESC_dTAG_24h_rep1_R1.fastq.gz
input_ESC_dTAG_24h_rep2_R1.fastq.gz
input_ESC_dTAG_24h_rep3_R1.fastq.gz
Pol2_ESC_DMSO_rep1_R1.fastq.gz
Pol2_ESC_DMSO_rep2_R1.fastq.gz
Pol2_ESC_dTAG_24h_rep1_R1.fastq.gz

Pol2_ESC_dTAG_24h_rep2_R1.fastq.gz
Med23_ESC_DMSO_rep1_R1.fastq.gz
Med23_ESC_DMSO_rep2_R1.fastq.gz
Med23_ESC_dTAG_24h_rep1_R1.fastq.gz
Med23_ESC_dTAG_24h_rep2_R1.fastq.gz
input_ESC_DMSO_Pol2_R1.fastq.gz
input_ESC_dTAG_24h_Pol2_R1.fastq.gz
input_ESC_DMSO_Med23_R1.fastq.gz
input_ESC_dTAG_24h_Med23_R1.fastq.gz
H3K27Ac_ESC_DMSO_rep1_R2.fastq.gz
H3K27Ac_ESC_DMSO_rep2_R2.fastq.gz
H3K27Ac_ESC_DMSO_rep3_R2.fastq.gz
H3K27Ac_ESC_dTAG_24h_rep1_R2.fastq.gz
H3K27Ac_ESC_dTAG_24h_rep2_R2.fastq.gz
H3K27Ac_ESC_dTAG_24h_rep3_R2.fastq.gz
H3K9me3_ESC_DMSO_rep1_R2.fastq.gz
H3K9me3_ESC_DMSO_rep2_R2.fastq.gz
H3K9me3_ESC_DMSO_rep3_R2.fastq.gz
H3K9me3_ESC_dTAG_24h_rep1_R2.fastq.gz
H3K9me3_ESC_dTAG_24h_rep2_R2.fastq.gz
H3K9me3_ESC_dTAG_24h_rep3_R2.fastq.gz
input_ESC_DMSO_rep1_R2.fastq.gz
input_ESC_DMSO_rep2_R2.fastq.gz
input_ESC_DMSO_rep3_R2.fastq.gz
input_ESC_dTAG_24h_rep1_R2.fastq.gz
input_ESC_dTAG_24h_rep2_R2.fastq.gz
input_ESC_dTAG_24h_rep3_R2.fastq.gz
Pol2_ESC_DMSO_rep1_R2.fastq.gz
Pol2_ESC_DMSO_rep2_R2.fastq.gz
Pol2_ESC_dTAG_24h_rep1_R2.fastq.gz
Pol2_ESC_dTAG_24h_rep2_R2.fastq.gz
Med23_ESC_DMSO_rep1_R2.fastq.gz
Med23_ESC_DMSO_rep2_R2.fastq.gz
Med23_ESC_dTAG_24h_rep1_R2.fastq.gz
Med23_ESC_dTAG_24h_rep2_R2.fastq.gz
input_ESC_DMSO_Pol2_R2.fastq.gz
input_ESC_dTAG_24h_Pol2_R2.fastq.gz
input_ESC_DMSO_Med23_R2.fastq.gz
input_ESC_dTAG_24h_Med23_R2.fastq.gz

| Genome browser session (e.g. UCSC) | UCSC track hub: http://ngs.molgen.mpg.de/ngsuploads/dept_meissner/Trim28_Submission.txt |

## Methodology

| Replicates | ChIP-Seq experiments were performed with 2 replicates for RNAPII and MED23 and with 3 replicates for H3K27Ac and H3K9me3. PolyA, Total RNA-seq, and SLAM-TT-seq experiments were performed with 3 biological replicates. |

| Sequencing depth | READ LENGTH<br>100 bp, paired-end<br><br>READS GENERATED<br>410687668  H3K27Ac_ESC_DMSO_rep1<br>110979600  H3K27Ac_ESC_DMSO_rep2<br>100900068  H3K27Ac_ESC_DMSO_rep3<br>88916272  H3K27Ac_ESC_dTAG_24h_rep1<br>95370374  H3K27Ac_ESC_dTAG_24h_rep2<br>196984348  H3K27Ac_ESC_dTAG_24h_rep3<br>74250624  H3K9me3_ESC_DMSO_rep1<br>134359096  H3K9me3_ESC_DMSO_rep2<br>98332334  H3K9me3_ESC_DMSO_rep3<br>127067384  H3K9me3_ESC_dTAG_24h_rep1<br>111877386  H3K9me3_ESC_dTAG_24h_rep2<br>142408192  H3K9me3_ESC_dTAG_24h_rep3<br>52207990  input_ESC_DMSO_rep1<br>54628918  input_ESC_DMSO_rep2<br>48538948  input_ESC_DMSO_rep3<br>21402648  input_ESC_dTAG_24h_rep1<br>50348032  input_ESC_dTAG_24h_rep2<br>59192212  input_ESC_dTAG_24h_rep3<br>52504118  Pol2_ESC_DMSO_rep1<br>60827162  Pol2_ESC_DMSO_rep2<br>52535240  Pol2_ESC_dTAG_24h_rep1<br>23526768  Pol2_ESC_dTAG_24h_rep2<br>61338540  Med23_ESC_DMSO_rep1<br>116714578  Med23_ESC_DMSO_rep2 |

86450084  Med23_ESC_dTAG_24h_rep1
72728384  Med23_ESC_dTAG_24h_rep2
13488990  input_ESC_DMSO_Pol2
8964686  input_ESC_dTAG_24h_Pol2
22051410  input_ESC_DMSO_Med23
10273856  input_ESC_dTAG_24h_Med23

READS UNIQUELY ALIGNED MOUSE (mm10)
344164541  H3K27Ac_ESC_DMSO_rep1
96585004  H3K27Ac_ESC_DMSO_rep2
79503843  H3K27Ac_ESC_DMSO_rep3
72703368  H3K27Ac_ESC_dTAG_24h_rep1
79715499  H3K27Ac_ESC_dTAG_24h_rep2
171498224  H3K27Ac_ESC_dTAG_24h_rep3
57983745  H3K9me3_ESC_DMSO_rep1
111284700  H3K9me3_ESC_DMSO_rep2
76982969  H3K9me3_ESC_DMSO_rep3
102970188  H3K9me3_ESC_dTAG_24h_rep1
89778418  H3K9me3_ESC_dTAG_24h_rep2
114868841  H3K9me3_ESC_dTAG_24h_rep3
45427465  input_ESC_DMSO_rep1
46429874  input_ESC_DMSO_rep2
38190218  input_ESC_DMSO_rep3
15313320  input_ESC_dTAG_24h_rep1
15313320  input_ESC_dTAG_24h_rep2
44597123  input_ESC_dTAG_24h_rep3
35279218  Pol2_ESC_DMSO_rep1
39964121  Pol2_ESC_DMSO_rep2
35342996  Pol2_ESC_dTAG_24h_rep1
14297274  Pol2_ESC_dTAG_24h_rep2
49903588  Med23_ESC_DMSO_rep1
107669156  Med23_ESC_DMSO_rep2
73871774  Med23_ESC_dTAG_24h_rep1
65622785  Med23_ESC_dTAG_24h_rep2
5610583  input_ESC_DMSO_Pol2
2358339  input_ESC_dTAG_24h_Pol2
19483039  input_ESC_DMSO_Med23
5433101  input_ESC_dTAG_24h_Med23

READS UNIQUELY ALIGNED DROSOPHILA (dm6)
44597123  H3K27Ac_ESC_DMSO_rep1
11659503  H3K27Ac_ESC_DMSO_rep2
11051087  H3K27Ac_ESC_DMSO_rep3
9505642  H3K27Ac_ESC_dTAG_24h_rep1
10509582  H3K27Ac_ESC_dTAG_24h_rep2
22429630  H3K27Ac_ESC_dTAG_24h_rep3
10520245  H3K9me3_ESC_DMSO_rep1
17902550  H3K9me3_ESC_DMSO_rep2
13217409  H3K9me3_ESC_DMSO_rep3
20398860  H3K9me3_ESC_dTAG_24h_rep1
20398860  H3K9me3_ESC_dTAG_24h_rep2
21341601  H3K9me3_ESC_dTAG_24h_rep3
11451871  input_ESC_DMSO_rep1
11024536  input_ESC_DMSO_rep2
9499176  input_ESC_DMSO_rep3
4831231  input_ESC_dTAG_24h_rep1
9254723  input_ESC_dTAG_24h_rep2
13725562  input_ESC_dTAG_24h_rep3
16837010  Pol2_ESC_DMSO_rep1
20002855  Pol2_ESC_DMSO_rep2
18386451  Pol2_ESC_dTAG_24h_rep1
7731020  Pol2_ESC_dTAG_24h_rep2
1197237  input_ESC_DMSO_Pol2
1197237  input_ESC_dTAG_24h_Pol2

Antibodies

RNAPII (8WG16; Biolegend)

Spike-in antibody (61686, Active Motif)

MED23 antibody (A300-425A; Bethyl Laboratories)

H3K27Ac (ab4729; Abcam)

H3K9me3 (ab8898; Abcam)

| Peak calling parameters | MED23 |
|---|---|

Raw reads of treatment and input samples were subjected to adapter and quality trimming with cutadapt (version 2.4; parameters: --nextseq-trim 20 --overlap 5 --minimum-length 25 --adapter AGATCGGAAGAGC -A AGATCGGAAGAGC). Reads were aligned to the mouse genome (mm10) using bwa with the 'mem' command (version 0.7.17, default parameters). A sorted BAM file was obtained and indexed using samtools with the 'sort' and 'index' commands (version 1.10). Duplicate reads were identified and removed using gatk (version 4.1.4.1) with the 'MarkDuplicates' command and default parameters. Technical replicates of treatment and input samples were merged respectively using samtools 'merge'.

Peaks were called with reads aligning to the mouse genome using MACS2 'callpeak' (version 2.1.2; parameters --bdg --SPMR) using the input samples as control samples. Genome-wide coverage tracks for single and merged replicates normalized by library size were computed using bamCoverage (version: 3.4.3; parameters: --normalizeUsing CPM --extendReads).

ALL OTHER

Raw reads of treatment and input samples were subjected to adapter and quality trimming with cutadapt (version 2.4; parameters: --nextseq-trim 20 --overlap 5 --minimum-length 25 --adapter AGATCGGAAGAGC -A AGATCGGAAGAGC). Reads were aligned separately to the mouse genome (mm10) and to the fly genome (D. Melanogaster, dm6) using bwa with the 'mem' command (version 0.7.17, default parameters). A sorted BAM file was obtained and indexed using samtools with the 'sort' and 'index' commands (version 1.10). Duplicate reads were identified and removed using gatk (version 4.1.4.1) with the 'MarkDuplicates' command and default parameters. Technical replicates of treatment and input samples were merged respectively using samtools 'merge'.

Peaks were called with reads aligning to the mouse genome only using MACS2 'callpeak' (version 2.1.2; parameters --bdg --SPMR) using the input samples as control samples. For H3K9me3 only, the '--broad' option was used. Genome-wide coverage tracks for single and merged replicates normalized by library size were computed using bamCoverage (version: 3.4.3; parameters: --normalizeUsing CPM --extendReads) and in addition normalized by the spike-in factor obtained from the reads aligning to the drosophila genome.

| Data quality | Quality of raw reads was assessed using FastQC. Reads were trimmed using cutadapt in order to remove low-quality bases and adapter content. |
|---|---|

FRACTION ALIGNED TO MOUSE (mm10) - The fraction might deviate from the numbers noted above due to reads being excluded after the trimming process.

0.93  H3K27Ac_ESC_DMSO_rep1
0.93  H3K27Ac_ESC_DMSO_rep2
0.93  H3K27Ac_ESC_DMSO_rep3
0.93  H3K27Ac_ESC_dTAG_24h_rep1
0.93  H3K27Ac_ESC_dTAG_24h_rep2
0.94  H3K27Ac_ESC_dTAG_24h_rep3
0.90  H3K9me3_ESC_DMSO_rep1
0.91  H3K9me3_ESC_DMSO_rep2
0.91  H3K9me3_ESC_DMSO_rep3
0.89  H3K9me3_ESC_dTAG_24h_rep1
0.89  H3K9me3_ESC_dTAG_24h_rep2
0.90  H3K9me3_ESC_dTAG_24h_rep3
0.89  input_ESC_DMSO_rep1
0.89  input_ESC_DMSO_rep2
0.89  input_ESC_DMSO_rep3
0.91  input_ESC_dTAG_24h_rep1
0.90  input_ESC_dTAG_24h_rep2
0.84  input_ESC_dTAG_24h_rep3
0.75  Pol2_ESC_DMSO_rep1
0.73  Pol2_ESC_DMSO_rep2
0.73  Pol2_ESC_dTAG_24h_rep1
0.71  Pol2_ESC_dTAG_24h_rep2
0.99  Med23_ESC_DMSO_rep1
0.99  Med23_ESC_DMSO_rep2
0.99  Med23_ESC_dTAG_24h_rep1
0.99  Med23_ESC_dTAG_24h_rep2
0.91  input_ESC_DMSO_Pol2
0.89  input_ESC_dTAG_24h_Pol2
0.99  input_ESC_DMSO_Med23
0.98  input_ESC_dTAG_24h_Med23

| Software | cutadapt |
|---|---|

cutadapt
bwa mem
samtools
gatk
MACS2
bamCoverage

# Flow Cytometry

## Plots

Confirm that:

☒ The axis labels state the marker and fluorochrome used (e.g. CD4-FITC).

☒ The axis scales are clearly visible. Include numbers along axes only for bottom left plot of group (a 'group' is an analysis of identical markers).

☐ All plots are contour plots with outliers or pseudocolor plots.

☒ A numerical value for number of cells or percentage (with statistics) is provided.

## Methodology

| | |
|---|---|
| Sample preparation | Cells were blocked with 10% FCS for 15 minutes followed by incubation with primary antibody for one hour at 37 degree C. This was followed by two washes at room temperature for 10 minutes each and measured with the flow cytometer. |
| Instrument | BD FACS Celesta, BD FACS Aria II |
| Software | FACS Diva (BD Biosciences) for collection and FlowJo (v1.07) for analysis |
| Cell population abundance | For each sample, a population of at least 10,000-30,000 cells was obtained, and the overall cell population was calculated using forward and side-scatter patterns. The abundance of cells in a population is represented as the normalized mode. |
| Gating strategy | Gating for negative and positive population was determined with untreated or isotype controls. |

☐ Tick this box to confirm that a figure exemplifying the gating strategy is provided in the Supplementary Information.

