## [Peer Review File · Nature Genetics]

Peer Review Information

Manuscript Title: Hijacking of transcriptional condensates by endogenous retroviruses

Corresponding author name(s): Dr Denes Hnisz

Reviewer Comments & Decisions:

Decision Letter, initial version:
--

21st Sep 2021

Dear Denes,

Your Article, "Hijacking of transcriptional condensates by endogenous retroviruses" has now been seen by 4 referees. You will see from their comments copied below that while they find your work of potential interest, they have raised quite substantial concerns that must be thoroughly addressed. In light of these comments, we cannot accept the manuscript for publication, but would be interested in considering a revised version that addresses these serious concerns.

Reviewer #1 says that, if robust, the findings would be important. However, they think that the data do not fully support the conclusions drawn at this stage. They have suggested a substantial number of experiments, which would entail a major revision.

Reviewer #2 is enthusiastic about this paper overall and only has relatively minor requests.

Reviewer #3 thinks that the proposed model is intriguing. However, they raise some serious issues with the conclusions drawn based on the data shown (limited number of loci examined, insufficient clarity and/or quantification, etc.). Their comments are thoughtful and extensive.

Reviewer #4 feels that there are some interesting findings here but highlights several problems with the data/analysis, particularly with the imaging/phase separation parts.

We hope you will find the referees' comments useful as you decide how to proceed. If you wish to submit a substantially revised manuscript, please bear in mind that we will be reluctant to approach the referees again in the absence of major revisions.

If you choose to revise your manuscript taking into account all reviewer comments, please highlight all changes in the manuscript text file. At this stage we will need you to upload a copy of the manuscript

in MS Word .docx or similar editable format.

We are committed to providing a fair and constructive peer-review process. Do not hesitate to contact me if there are specific requests from the reviewers that you believe are technically impossible or unlikely to yield a meaningful outcome.

*2) If you have not done so already please begin to revise your manuscript so that it conforms to our Article format instructions, available [here](http://www.nature.com/ng/authors/article_types/index.html). Refer also to any guidelines provided in this letter.

[REDACTED]

If you wish to submit a suitably revised manuscript we would hope to receive it within ~6 months. If you cannot send it within this time, please let us know. We will be happy to consider your revision so long as nothing similar has been accepted for publication at Nature Genetics or published elsewhere. Should your manuscript be substantially delayed without notifying us in advance and your article is eventually published, the received date would be that of the revised, not the original, version.

Nature Genetics is committed to improving transparency in authorship. As part of our efforts in this direction, we are now requesting that all authors identified as 'corresponding author' on published papers create and link their Open Researcher and Contributor Identifier (ORCID) with their account on the Manuscript Tracking System (MTS), prior to acceptance. ORCID helps the scientific community

achieve unambiguous attribution of all scholarly contributions. You can create and link your ORCID from the home page of the MTS by clicking on 'Modify my Springer Nature account'. For more information please visit www.springernature.com/orcid.

Thank you for the opportunity to review your work.

Sincerely,

Tiago

Tiago Faial, PhD
Senior Editor
Nature Genetics
<https://orcid.org/0000-0003-0864-1200>

Reviewers' Comments:

Reviewer #1:

Remarks to the Author:

In this work, Asimi and colleagues investigated the mechanism by which the transcriptional derepression of endogenous ERVs mediated by TRIM28 inactivation drives early embryonic lethality. To address this question, they developed advanced methodologies to overcome the intrinsic limitations related to gene knockout or knockdown approaches, reaching the required spatial and temporal resolution to dissect the molecular mechanisms involved in this relevant biological process. Specifically, they generated zygotic deletion of TRIM28, which permitted to show that derepression of IAPs occurred at E3.5 blastocysts, concomitantly with the downregulation of the pluripotency factors, leading to perturbation of the epiblast specification and an increase of extraembryonic lineages. To investigate the relationship between derepression of ERVs and downregulation of pluripotency factors they generated knock-in ESCs (and iPSCs) that encode degradation-sensitive TRIM28-FKBP alleles, allowing the stimulation of TRIM28 proteolysis upon exposure to the dTAG-13 ligand. By analyzing nascent transcript levels combined with chromatin profiling upon TRIM28 degradation they measured a concomitant derepression of ERVs and the downregulation of pluripotency genes, which are transcriptionally regulated by super-enhancers (SE). They proposed that the reduced transcription occurring at the SE was caused by the dissociation of the transcriptional condensates from the SE loci, as the ERV transcripts outcompete for RNAP and Mediator assembly in newly transcriptional condensates localized at the ERV loci. In sum they concluded that the derepression of ERVs causes a spatially restricted increment of ncRNAs acting as a nucleation event for the assembly of transcriptional condensates at the expense of the maintenance of the transcriptional hubs at the SE loci, which would cause the downregulation of the associated pluripotency genes. Although the suggested regulatory mechanism is of high interest and could potentially explain the observed transcriptional regulatory pattern, the provided data do not fully support the drawn

conclusions. Specifically, the authors did not demonstrate neither in vitro or within cells that the accumulation of ERV transcript competes for transcriptional condensates at the expense of SE loci. The Piggybac transposon system is an indirect measurement of the proposed mechanism and is not sufficient to support this conclusion. The in vitro assays showed that RNA molecules could enhance phase separation of MED1 and RNAP-CTD, but this phenomenon seems to occur on the basis of the electrostatic interactions with little specificity towards ERV transcripts. The knock-down assay showed that the reduced transcript levels of ERVs was concomitant with the rescue of SE transcript and the increase of the associated pluripotency gene expression. However, this assay suffers of some technical issues that need to be addressed to fully support the main conclusion. In sum the current form of the manuscript is not suitable for publication and it would need a major revision to correctly address the main points raised by the authors.

Major Issues:

1. If the main conclusion of this work consists in a regulatory mechanism based on the competition between ERV and SE transcripts to nucleate the assembly of transcriptional condensates at specific loci, then the authors should provide direct evidence for this mechanism, possibly in vitro and in vivo. For example, they could perform the droplet formation assay by incubating the recombinant proteins (MED1 and RNAP-CTD) with labelled eRNAs to a defined molarity, to then add the labelled IAP RNAs at increasing concentrations and measuring the competition between the ncRNA molecules to assemble the droplets.

In the presented data (Fig. 3d-h) the authors did not indicate the molarity of the recombinant proteins, so it is difficult to determine the relative efficacy in the droplet formation. This information was provided in ED Fig. 12, in which they measured the effects of increasing amount of IAP RNAs on the droplet formation of RNAP-CTD and MED1. However, the retrieved data showed no linear correlation between the concomitant increase of protein and RNA concentration and the number (or size?) of the assembled droplets (ED Fig. 12a-b). This finding is not clear and is potentially in contrast with the recent work from Henninger et al. Cell 2021 (PMID: 33333019), which showed that the ratio between negative and positive net charges governs the assembly and disassembly of in vitro droplets. However, in the same set of experiments (ED Fig. 13), the authors presented data retrieved from the same assay that indeed showed a concentration-dependent effects of multiple RNA molecules (IAP, major satellite and eRNAs) on the assembly of MED1 droplets, showing that at the highest concentration the assembly is disfavored. How could these results be in agreement with the proposed model of RNA molecule competition for the assembly of the transcriptional condensates? This point needs to be addressed. In addition, they used a very high concentration of the crowding agent PEG (20%) without any explanation: could the authors justify this procedure?

2. On the same line, the TT-SLAM-seq results showed by meta-analysis representation (ED Fig. 3c) an accumulation of nascent transcripts at SE, which seems to be way higher than what is measured along gene bodies. Although the same analysis was not performed for the ERV transcripts, it would be expected that the relative accumulation of nascent transcripts at these loci will not exceed the amount measured at the SE or within gene bodies, as highlighted in the screenshot referring to Cthrc1 locus (Fig. 2e). Can the authors provide this information? If so, how can they explain the possible competition of these relative low abundant transcripts for the assembly of transcriptional condensates with respect to SE and the associated genes? The explanation of few ERVs being localized in proximity of activated genes is not sufficient to justify the proposed mechanism. Although the ERV TKO experiment shows the influence of these elements in the activation of Cthrc1 it is not possible to exclude that the derepression is necessary for their functioning as de novo enhancers, as previously

shown (PMID: 22722858 and 27259207). In addition, in the same experimental setting it has not been clarified whether the other proximal genes are equally sensitive to ERV KO or TRIM28 degradation: how can these divergent results be reconciled in a general mechanism governing transcriptional condensates assembly?

3. The Hi-C assays showed that the majority of ERV taxa switched from B (inactive) to A (active) compartments, showing an increase of contacts with active genes. Although this finding may support the notion that the derepression of ERVs determines their assembly in transcriptionally active compartments, the provided data do not fully support the conclusion. Indeed, the PC1 ratio shown in ED Fig. 10c showed a very modest increment and without any other reference, it is difficult properly evaluate the biological significance of this measurement. The authors should provide further analyses showing the same analyses with respect to activated genes and or enhancers. In addition, the interactions occurring at the highlighted locus corresponding to chr12, which is characterized by a large stretch of ERV elements in a gene-poor region (Fig. 2h) were not properly measured by the Hi-C experiment, as shown in ED Fig. 10b, possibly due to poor read coverage. On the basis of this, the described switch from B to A compartments should be re-interpreted or demonstrated using orthogonal approaches.

4. To prove the competition between ERV and SE transcripts for the assembly of the transcriptional condensates within cells, the authors knocked-down the ERV transcripts before inducing TRIM-28 degradation. In this condition, they could retrieve the rescue of the downregulation of both SE and the linked gene expression (Fig. 3a-c and ED Fig. 11). However, the authors did not explain why it is necessary to knock down the ERV transcripts even before they are derepressed. In addition, they did not verify whether the analyzed cells still represent pluripotent stem cells, as the transcript levels of key pluripotency factors are particularly low in this setting. For example, the levels of Klf4, Fgf4 and miR290-295 are much lower with respect to what have been measured by qRT-PCR in the same set of experiments in which the knockdown co-occurred with the TRIM28 degradation (ED Fig. 11a-b) or the time-course analyses shown in ED Fig. 5c. These inconsistencies suggest that possibly TRIM28 degradation drives the exit from pluripotency, as supported by the gene expression analyses performed at later time points. These data bring toward a possible alternative interpretation of the data by which the measured changes in the SE transcript levels in consequence of TRIM28 degradation are not, at least uniquely, depending on the derepression of ERVs and the competition between transcripts may play a marginal role, if any.

5. To address this critical point, the authors should directly measure the contribution of ERV transcripts to the newly assembly of transcriptional condensates at the expenses of the SE-assembled condensates, causing their downregulation. For example, they could monitor in living cells the transient assembly/disassembly of the endogenous MED1 (or RNAP) into condensates at the ERV and SE loci, in response to TRIM28 degradation. By combining live FISH approach (PMID: 31488703) with PALM imaging (PMID: 27138339) they could determine the potential changes in the protein lifetime at the specified loci and the consequence of perturbing ERV or SE transcript levels. Although these are challenging experiments that may require extensive setting, we believe that is necessary to measure the dynamic response of the soluble factors that participate in the assembly of the transcriptional condensates upon the seeding of ERV transcripts.

6. Some critical controls are missing: considering that the majority of the experiments rely on a single knock-in clone of ESCs, it is fundamental to provide further data indicating that these cells behave similarly to the parental one. There are no sufficient data showing that this clone maintained the self-

renewal capacity and the cell lineage potential compared to the parental cell lines, in both DMSO and dTAG-13 treated (24h) conditions. The authors should provide this information, focusing on the capacity of the cells to give rise to the EpiLCs and also to extraembryonic lineages, by performing the dedicated differentiation assays and measuring the related phenotypes. In addition, the WB showing the efficacy of the degron system should include the TRIM28 protein level of the parental cell line (Fig. 1d), to verify whether the added tag did not alter the protein stability. Similarly, all the gene expression analyses measuring the nascent transcripts were performed by comparing the knock-in cell line treated with DMSO and dTAG-13. However, to ensure that the analyses changes are not in part due to a different cellular state of the analyzed clone, the authors should provide a comparative gene expression analyses between the DMSO-treated knock-in cell line and the parental ESCs.

7. It is not clear how the number of the differentially expressed genes have been obtained: following what is written in the manuscript the authors used the following parameters: 2-fold change, $FDR < 0,05$. However, on the basis of the available data (Table S4), the 328 downregulated genes are retrieved only by not considering the adjusted p-value cutoff. Is this correct? If so, the number of the differentially expressed genes at 2h and 6h shown in Fig. 1g do not correspond the numbers retrieved from the same table. Please clarify this point and correct it accordingly.

8. The authors measured a strong increment of nascent transcripts of different ERV taxa with a clear increment of some classes, such as IAP and MERVK (ED Fig. 4), while other such as MMETn are not shown. However, the ChIP-seq experiments (ED Fig. 9b) show no enrichment for MED23 or RNAP and depositing of H3K27ac at IAP loci: could the authors explain these discrepancies?

9. The authors adopted a combination of IF-RNA FISH to measure the colocalization of transcription sites with RNAP and MED1. The designed method needs further control to support the quantitative analyses as the authors measured the coefficient of correlation between the two signals at the center of the FISH foci and as a control they choose a random position. Considering that the nascent RNA-FISH will detect at maximum two loci/cell, it is quite expected that a chosen random region will show very poor colocalization, whose signal may only depend to the technical artefacts/background. We recommend to compare the appropriate level of colocalization between SE loci and RNAP with respect to another transcriptionally active region (not SE-enriched) by RNA-FISH. Only this comparison may indicate whether the analysed loci are enriched at the experimentally defined transcriptional condensates, and the relative changes upon TRIM28 degradation.

10. The authors measured by IF-RNA FISH the colocalization between RNAP, MED1 and NFY with the nascent transcripts of IAPs upon TRIM28 degradation. They claim that the obtained results indicated an increase of colocalization at some IAP loci. However, the obtained measurements by Mean menders colocalizations (MCC) is 0.1, which would indicate that the two signals poorly colocalize. They claim that a similar pattern is obtained by IF for MED1 and MED23; however the quality of these images are so poor (especially for MED23) that it is impossible to rely on these data. Of note, in the same manuscript the authors described a reduction of colocalization between SE and the same proteins when they measured a change of the MCC from 0.86 to 0.73, upon TRIM28 degradation. The interpretation of these data lack consistency. Considering these results, we strongly suggest to reconsidering the interpretations of the data or to perform different measurements of the colocalization. For example, the authors could determine the frequency of the colocalization events/cell, considering the distributed pattern of IAP loci. Alternatively, they could quantify and plot the relative distance between the IF cluster and the FISH foci (see also PIMD: 31494034). Beside that, they should also measure the relative colocalization for other classes of ERVs, considering that IAP loci

showed a very modest (if any) enrichment for H3K27ac, RNAP and Mediator retrieved by ChIP-seq.

11. To define the possible relationship between SE activation and the assembly of condensates at ERV loci, the authors used Dox-inducible expression of pluripotency factors in iPSCs knock-in for TRIM28 degen system. The obtained results indicate that upon 24h induction of TRIM28 degradation there is a strong decrease of OCT4 and SOX2 protein levels (Fig. 4b), in contrast to what has been achieved in the ESC clone (Fig. 1d). How do they explain this discrepancy? How can they exclude that these cells are actually exiting from pluripotency independently from the ERV derepression?

12. In most cases it is not clear/specified whether the retrieved quantification of colocalizations were performed on independent biological and/or technical replicates. The same holds true for the qRT-PCR measurements and for the in vitro droplet quantifications.

Reviewer #2:

Remarks to the Author:

A. Summary of the key results

The work from Asimi et al. set out to answer a fundamental question in TE/ERV biology: how does ERV derepression cause such dramatic effects on early embryos?

To this end, they established a system in which they can degrade Trim28 protein post-translationally in order to perform loss-of-function experiments. Since other systems have used Cre-lox or shRNA which only fully deplete Trim28 at later time points (Rowe et al. Nature 2010), this work from Asimi et al. is more elegant and conclusive.

Their major findings:

- 1) Depletion of Trim28 preferentially decreases super-enhancer driven transcription in mESC (Fig 1)
- 2) In the context of Trim28 depletion, ERV RNAs are spatially co-localized with RNA pol-II condensates (Fig 2)
- 3) Surprisingly, this effect is likely mediated through the ability of ERV RNA to direct or stabilize RNA pol-II to other sites and away from super-enhancer driven transcripts. (Fig 3)
- 4) Although the authors first focus on the effect of ERV RNA to confer this effect (Fig 3), they also show that the abundant ERV genomic loci also compete with other OSKM bound loci (canonically super-enhancers) for transcriptional machinery, thus elaborating on another mechanism through which ERV de-repression can affect transcription of other genes. (Fig 4)
- 5) Finally, they show that depletion of Trim28 in embryos leads to the loss of the pluripotent compartment in embryos, presumably through a similar interference with OSKM super-enhancer regulation of pluripotency genes. (Fig 5)

B. Originality and significance: if not novel, please include reference

This paper addresses several important unanswered that have dogged the TE/ERV field for many years. It was assumed that TE gene products—specifically retrotranspositionally competent LINE-1s or ERVs (such as MERV-L) were re-integrating into the genome during derepression that the cellular toxicity was mainly mediated through a damage response or dsRNA toxicity. Based on this work, in early embryos and embryonic stem cells, it appears that the RNA produced can re-target the RNA pol-II machinery and potentially that the binding of OSKM into new sites (and away from canonical super-enhancers) or titration of the RNA pol-II away from SE can also contribute to the loss of pluripotency. Both of these distinct mechanisms are extremely important for the TE field to embrace, and the work

from Asimi et al. finally provide a compelling molecular mechanism for earlier important work (Jachowitz et al. *Nature Genetics*, 2017) that unfortunately remained in the realm of phenomenology. I cannot stress how crucial this paper will be for the TE/ERV field, it will reorient many people's view on TE reactivation during early development.

C. Data & methodology: validity of approach, quality of data, quality of presentation

In general, the manuscript is extremely compelling, easy to read, and the data are clearly presented. As mentioned above in my comments, the strategy of using a Trim28-degron approach is clearly an improvement from earlier Trim28 depletion strategies (Rowe et al. *Nature* 2010). Although the authors use Crispr/Cas9 to deplete Trim28 in embryos, this will not affect the enormous maternally-deposited stockpile of the protein (described in Messerschmidt, et al. *Science* 2012) but they instead focus on the effect of Trim28 depletion on the pluripotent compartment (Fig 5)—which is totally appropriate given the pluripotency heavy emphasis for the rest of their manuscript.

There are minor recurring issues with the presentation of some fluorescence data that should be rectified: I would suggest changing the color scheme of dark purple on black (Figs. 1i, 2a-c,f,g, 3c,d,h, 4f, 5b,g) to grayscale in order for the signal to actually be seen.

D. Appropriate use of statistics and treatment of uncertainties

In an effort to conform to the *Nature Genetics* format, I would suggest for each figure legend to briefly mention the statistical test performed. For instance, although Fig 1E, 1G is likely DESeq2 analysis, this is only described in the methods but not the figure legend itself. Additionally, if DESeq2 is used for Fig 1G, would it be more standard to label the y-axis as log₂-FDR not p-value?

For Fig 1H: based on the shape of the boxplot, I can almost guarantee these are not normal distributions, and therefore t-test are inappropriate. Why not use a non-parametric test (Wilcoxon-Mann Whitney), etc? For Fig 1H, it would be appropriate to include the # of each category "n=blank" either in the figure legend or the figure panel itself.

Fig 2: It would be a "best practice" to quantify the co-localization of IAP ERV RNA with indicated features (RNA pol-II, MED1, NFYA) in Fig 2a,b,c rather than only show a single z-slice, and include the # of images quantified (to match the sample size reporting in Fig 2f, g).

Maybe the sample size is reported in the methods, but for Fig 3b and 4e,g,i the sample size is not indicated in the figure legend or directly on the figure panel.

The sample size in 5c,g is not indicated.

E. Conclusions: robustness, validity, reliability

The conclusions are robust, and the depth of the mechanistic rigor (particularly for Figs. 3a,b, 4i) is excellent.

I was sad to see that the data from the ERV TKO experiment (Extended data Fig 7g) was tucked away in the supplement—as this is really a definitive genetic proof of the effect of the nearby ERVs on *Cthrc1* transcription levels.

A point of confusion: the in vitro condensate data from Fig 3d-i: these data would be more impactful if the specificity controls (satellite or eRNAs) were included in the main figure so we can compare them.

In Extended data Fig 13b, the authors perform experiments comparing different RNAs and their ability to promote droplet formation. Do the authors claim that IAP RNAs preferentially allow for better droplet formation compared to other RNAs? Is the claim that IAP ERV RNA has special sequence or structural features that mechanistically explain the difference in droplet formation in vitro? X-axis in Fig 3e-g and Extended Data Fig 13b is nM of RNA. Is this nM of individual bases or nM of RNA (polymer) molecules? Are the length of the IAP, satellite, and eRNA the same? Is the eRNA mostly

dsRNA or are these non-complementary RNAs?

F. Suggested improvements: experiments, data for possible revision

This manuscript is in an extremely advanced form, and in large part, the claims the authors make are well supported by their data. Based on my routine reading of Nature Genetics and the excellent TE/ERV research it publishes, this manuscript meets this very high bar without additional, largely superficial, reviewer-requested experiments that often do not significantly change the major conclusions of the work.

G. References: appropriate credit to previous work?

Yes.

H. Clarity and context: lucidity of abstract/summary, appropriateness of abstract, introduction and conclusions

The manuscript is well-written, the abstract is structured well to focus on claims that they investigate in the manuscript, and the intro and conclusion are appropriate.

I would like for the authors to comment on why the Trim28 KO embryo phenotype has such a specific depletion of the pluripotent cell compartment. Are the other cell types (ExEnd and ExE derivatives) somehow less sensitive to ERV reactivation after Trim28 depletion? Is the reason the authors observe such a striking depletion of the epiblast (in Trim28 KO embryos see Fig 5e-g) because ExEnd and ExE rely less on super-enhancers for their cell identity or is it because ERVs are preferentially expressed by TFs present in the ICM/epiblast and therefore loss of Trim28 has an outsized effect on epiblast survival? If it is the latter, TE/ERV expression analysis of the existing Trim28 scRNAseq dataset should reveal if ERVs are derepressed in the ExEnd and ExE and these cell types are just more tolerant of this derepression. This is a relatively minor point, but a cursory clarification on this point in the text would help to frame their ESC data in the context of the *in vivo* embryo data.

Reviewer #3:

Remarks to the Author:

Endogenous retroviruses (ERVs) occupy about 10% of mammalian genomes and are potentially mutagenic if uncontrolled. Although numerous mouse mutants that cause ERV derepression (including Trim28 KOs) are embryonic lethal, whether ERV de-repression *per se* disturbs early embryonic development is still unclear. In this manuscript, Asimi et al. used the dTAG system to rapidly degrade TRIM28, a factor required by KRABZinc Finger Proteins (KZFPs) to repress many ERVs in mouse ES cells. They report that TRIM28 degradation causes the dissociation of transcriptional condensates at super-enhancers, with a relocalization of condensates at de-repressed/transcribed ERV loci. Further knockdown of ERVs in TRIM28 depleted cells rescues the dissociation of transcriptional condensates at the MiR290-295 locus. *In vitro* droplet assays demonstrate that IAP RNA can facilitate condensates with RNAPolIII CTD, HP1, and MED1 IDR. The authors then show that forced expression of OSKM factors (Oct4, Sox2, Klf4, c-Myc) during TRIM28 degradation in iPSC suppresses IAP transcripts and foci, restoring expression and transcriptional condensate for MiR290-295. Using a zygotic perturbation platform, the authors KO TRIM28 in embryos and report a correlation between ERV de-repression and the onset of lethality and the loss of specific cell lineages in mouse embryos. Based on these results, the authors propose a hijacking model that ERV derepression can hijack transcriptional condensates

from super-enhancers, providing a mechanism to explain “why ERV de-repression may be lethal in early embryos”.

This paper is of interest to readers in the field of epigenetics and endogenous retroviruses (ERVs). It aims to address the acute consequences of ectopic ERV expression upon loss of the adaptor protein TRIM28 (long known to be involved in ERV repression). The experimental systems used in this study are timely and elegant (e.g. overcoming the lethality of Trim28 in ES cells using the dTAG technique, assessing nascent transcripts using TT-SLAM-seq and characterizing mouse embryos using single-cell seq) and appropriate controls are used. Furthermore the “enhancer hijacking model” of ERVs is attractive and relatively novel in this field. Overall, however there are a number of major issues with the manuscript that prevent me from recommending publication. On the positive side, the authors provide some compelling data consistent with their model of redistribution of condensates upon TRIM28 depletion, however it is only shown for one locus (MIR290-295), and some of the imaging data supplied was of too low quality or lacking statistics to allow a more quantitative assessment (specifically the new formation of condensates at IAP loci). Furthermore, there is a major disconnect between the experiments performed in cell culture models and those performed *in vivo*, and at least one of the experiments performed in cell culture lacks a clearly explained rationale (the induced expression of a GFP from a DNA transposon integrated transgene). Perhaps most importantly however, the authors fail to address the key question set out at the start of the manuscript: whether ERV transcription cause the lethal phenotype. I will detail these concerns below.

Major points:

1. “TRIM28 degradation leads to the loss of transcriptional condensates at super-enhancers in mESCs” is a key point in this manuscript. However, the loss of transcriptional condensates at super-enhancers is only assessed at the MiR290-295 locus. (A more minor issue is why the authors used the nascent RNA of MiR290-295 to check the dissociation of transcriptional condensates on the super-enhancer rather than assessing the super enhancer directly (Figure 1i). Moreover, whether loss of transcriptional condensates at super-enhancers after TRIM28 degradation is a generic pattern or just a special case for MiR290-295 is not clear. The ChIP-seq data on RNAPII seems to neither support that transcriptional condensates are generically assembled at super-enhancers, nor supports the generic loss of transcriptional condensates on super-enhancers after TRIM28 degradation (Extended data Fig. 9b).
2. “De-repressed IAPs form nuclear foci that associate with RNAPII condensates” is another key point for the hijacking model. Although some IAP foci appear to co-localized with RNAPII condensates, there are many more IAP foci that are not co-localized with RNAPII condensates (Figure 2a). It should be tested whether IAP foci are significantly overlapped with RNAPII condensates or not compared with other transcribed regions (Figure 2a). In addition, the images for the co-localization of IAP with other parts of transcriptional condensates (i.e., MED1 and MED23) seem to have high background, I cannot even assess whether MED1 and MED23 signals are enriched in the nucleus (Figure 2b and Extended data Fig. 7b).
3. To test whether de-repression of ERVs can form and hijack transcriptional condensates, the authors mainly use IAPs as a model through the whole manuscript. However, the ChIP-seq data do not support that components forming nuclear condensates, namely RNAPII and Med23, are largely moved to IAP regions after TRIM28 degradation (Extended data Fig. 9b). And further knocking down IAPs in TRIM28 depleted cells did not rescue the expression of miR290-295 and Fgf4 or their super-enhancers (Extended Data Fig. 11d), indicating IAPs may not hijack transcriptional condensates in the way

described.

4. The authors failed to activate IAPs with CRISPR approaches (as have many labs!) and they instead used a dox-inducible GFP integrated as part of a DNA transposon-based vector to mimic the expression of an interspersed ERV (Figure 4h-I and Extended Data Fig. 15). The rationale for this choice is not clear or well explained. Neither the sequence expressed (GFP), the regulatory sequences, nor the copy number strongly resembles the ERV families in question (IAPs). It is my guess they are arguing that expressing any repeat locus to high levels could hijack the transcriptional machinery, and I am not sure what value such a statement makes with regards to the function of TRIM28 and ERVs. It is also hard to imagine why clones with "high" GFP expression have such altered expression of some super-enhancers and their target genes but clones with "middle" GFP expression have totally no influence (Figure 4i and Extended Data Fig. 15d), despite only a two-fold reduction in expression (extended data Fig. 15c). Is there really such a sweet spot for transcription of a repeat that would hijack these condensates? Would a single locus work if the expression levels were high enough?

5. The molecular characterization of the mouse embryos furthers our understanding of the Trim28 phenotype, by demonstrating a loss of pluripotent lineages and expression of ERV-derived proteins. This analysis *in vivo* seems however, rather disconnected from the experiments *in vitro*: functional links between ERV expression and the loss of pluripotent lineages are not assessed/observed in ESCs and the pluripotency gene and other markers assessed are not linked to the loci assessed in the first part of the paper. This needs to be rectified.

6. A major question posed in the manuscript is whether/why ERV de-repression may be lethal in early embryo and whether a redistribution of condensates away from super-enhancers can explain the lethality. However, the evidence for whether IAP de-repression causes lethality is still very weak. TRIM28 deletion or depletion causes many molecular phenotypes in ESCs and *in vivo* in addition to those phenotypes described in this manuscript relating to nuclear condensates. For example, Trim28 depletion causes ERV and potentially toxic gene de-repression, loss of imprinting, loss of heterochromatin, accumulation of DNA damage, activated retrotransposition of ERVs and LINEs, formation of viral particles, amongst others, that could all contribute to lethality. None of these have been ruled out nor extensively discussed. There is also data from KZFP cluster KO ESCs and embryos that demonstrate that ERV de-repression is compatible with embryogenesis. How do the authors reconcile this with their model? Is it the extent of derepression that matters?

7. In this work, the authors use shIAP, shMMEVK10c, shMMERVK9c and shMMETn to knock down these ERVs during TRIM28 degradation. However, qPCR primers used to measure MMETn and MMERK9c elements failed as stated by the authors (Lines 619-621 and extended data Fig. 11b). In this situation, how do the authors know whether the shMMETn/MMERK9c assays succeeded or not?

8. For the scRNA-seq in embryos, the authors performed scRNA-seq for the wild-type embryos at E5.5 and Trim28 KO at E6.5 and downloaded those data for wild-type E6.5 and E7.0 from GEO. The proportion of cell populations from wild-type E5.5 and Trim28 KO E6.5 present obviously larger variance than these from GEO (wild-type E6.5 and E7.0) (Extended Data Fig. 17a), indicating there are significant issues during embryo isolation and/or cell sampling for scRNA-seq. In addition, how TRIM28 KO embryos are identified and how the authors deal with the mosaic samples (if Trim28 is partially KO in the embryos) is unclear. These should be addressed.

Minor Points:

1. Parts of the abstract may be difficult to understand for non-specialists, e.g.
 - a. Line 39: NFY may not be a widely known factor
 - b. Line 45/46: "nascent RNAs may facilitate "hijacking" of transcriptional condensates in various developmental and disease contexts". It may not be obvious to the reader what hijacking of transcriptional condensates means.
2. Line 32 "ERV transcription or RNA transcripts may underlie essentiality of ERV repression". To our knowledge the essentiality of ERV repression for the Trim28 KO phenotype has not been formally proven experimentally. We thus suggest rewording this sentence e.g. by adding 'potential/suggested essentiality of ERV repression'. The main text sentence line 60/61 is more accurate "why ERV de-repression may be lethal in early embryos and pluripotent cells is a mystery".
3. Figure 1a and Extended Figure 1b and Extended Figure 1a and 4a are identical. Is the repetition of this figure in the extended data figure necessary?
4. Line 144/145: "These results demonstrate that transcriptional condensates may incorporate genes and de-repressed ERVs". While the data in this paragraph does demonstrate that combined KO of nearby ERVs rescue the phenotype at this locus (i.e. no upregulation of the nearby transcript is observed) it does not show that this rescue relies on the formation of new transcriptional condensates. As the ERVs are also proximal to 2 other genes, Slc25a32 and Dcaf13, it would also be of interest to know if these genes are also upregulated upon Trim28 KO. Do the authors think these ERVs act as enhancers, e.g. do they contain histone marks indicative of putative enhancers?
5. KD of the four most-induced ERV families partially rescued the downregulation of super-enhancers particularly when shRNAs were administered 24h prior to Trim28 degradation. What about other genes (genome-wide changes) or embryonic lethality? Are these rescued?
6. How do the authors explain that in the absence of Trim28 pluripotent cells in the embryo are gradually depleted while ESCs seem to not alter pluripotency marker expression? Please discuss.
7. Line 87-88: were the down-regulated genes enriched specifically for super-enhancer associated pluripotency genes or pluripotency genes overall? How are enhancers less transcribed upon acute loss of Trim28 when Trim28 is not bound to these enhancers or is it?
8. Ext Figure 4b+c: Does TT-SLAM-seq not detect LINE and SINE elements or are these simply not nascently transcribed in ESCs? These have previously been shown to be derepressed in Trim28 KOs.
9. Can the authors speculate what could cause the reduction of pluripotent cells in TRIM28 mutant blastocysts and gastrulation-stage mouse embryos? Next to de-repressing ERVs, Trim28 seems to regulate critical developmental TFs such as e.g. Fgf4 (Fig 1g) that impair proliferation of the inner cell mass when lacking (Feldmann 1995) are responsible? Please comment/discuss.
10. Line 228: "To probe which cell types are affected by the presence of IAP foci in blastocysts..." > this experiment does not probe effects of the IAP foci. Please change the wording accordingly or test functionally.

review signed by Todd Macfarlan

Reviewer #4:

Remarks to the Author:

The manuscript by Asimi and colleagues presents a very thorough molecular investigation of mouse ES cells upon acute depletion of Trim28/KAP1. The authors characterise changes in H3K9me3, H3K27ac, HP1a and binding of OCT4, SOX2 and Nanog (OSN). Their main conclusions are that some TEs are derepressed and transcription (and most likely associated activity) of eRNAs from pluripotency-associated enhancers bound by OSN are downregulated. The authors propose that TE transcription competes with enhancer transcription, presumably indirectly by 'hijacking' RNA polII. To support these conclusions, they present analysis of RNA PolII ChIP and microscopy-based analyses of RNA PolII and RNA-FISH for TEs and a chosen pluripotency-enhancer driven gene upon depletion of Trim28 as well as an 'overexpression' approach with GFP.

In general, this is a very interesting paper and I find it convincing and novel but I do not think that the phase separation (PS) part adds much to the manuscript, nor to the mechanistic investigation of the observations therein, and the quality of this data is in general less good. All the section on PS is very general and it does not really explain the underlying e.g. competition hypothesis and only general conclusions can be drawn from this part (e.g. that i) RNA is important- which we know- but not really specific for ERVs and ii) that hexanediol affects transcription in general, which we know). While the role of Trim28 in regulating TEs is well known in the literature, the effects on pluripotency/enhancers are completely novel and well documented by the authors in this manuscript.

Overall, my comments regard the addition of some missing controls and quantifications throughout and strengthening the PS section, as follows:

1. Authors relate RNA PolII staining (and ChIP) to RNA PolII activity, but they used an antibody recognising the non-phosphorylated (CTD) RNA polII, which is not the best proxy for RNA polII activity – can the authors validate (at least some of their key findings) using a CTD-phosphorylated, active RNA pol II antibody?
2. In Fig. 1j, the colocalization is hard to evaluate: there is also quite some signal 'adjacent' to the mir290-295 FISH. The shape of the FISH spot in the merge and in the inset looks very different, it would seem that image contrast has been adjusted, perhaps unequally. The authors should show raw data. Along the same lines, the data would be much more robust if the authors could provide better quantifications throughout their co-localisation datasets: they used Mander's coefficient for some of their data, but not for all. Also, it would be useful if they would provide an estimate of the variability across cells.
3. Authors conclude on Cthrc1 FISH colocalization/lack of colocalisation with NFYA based on images shown in Fig. 2f. However, NFYA is all over the place and so the probability of 'any' FISH to colocalise with NFYA is probably very high and the relevance of this potential colocalization is therefore a little overstated. Perhaps including quantifications of number of FISH spots that co-localise (or not) with NFYA and the inverse analysis may help to strengthen this conclusion. The Mander coefficient is a good proxy for global colocalisation, but it does mask additional important information of the number of events when this is observed.

4. Is droplet formation (Fig. 3c) affected by the phospho status of the CTD ?
5. Figure 2b. Why is there so much Mediator signal in the cytoplasm? The Mediator immunostainings are very poor (also those in Fig. S7), and hardly conclusive if at all, considering broad cytoplasmic localisation.
6. The interpretation of the authors that ERV transcription 'competes' with transcription of OSN super-enhancers is interesting, but alternative interpretations may be plausible. It assumes that RNA PolII is limiting, but it is unclear if the phenotype is due to a shortage of RNAPII or of Mediator? Can they rescue the effect on super-enhancers by overexpressing RNAPII or MED23?
7. The in vitro droplet work is not entirely convincing: it seems that any RNA will have an effect on droplet formation, as the authors do point out. But there seems then not to be a negative control – what about using an of an ERV that does not increase their expression after Trim28 KD (fig 3d-I and fig S13)?
8. Along the same lines, the figure for the NFYC-IDR incorporation in MED1 droplets (Fig 3h-i) has no negative control.
9. The conclusions on 'condensates' (and PS) in cells are relatively weak, the only real experiment which aims to (partly) address whether the puncta are actually condensates is based on 1,6-HD treatment. While this is routinely used in the field, in isolation it is not a strong indication that PS or condensates form in vivo, and in addition, 1,6-HD can just be inhibiting transcription in general. The authors should perform additional experiments in cells to strengthen their manuscript. Otherwise, the term 'condensates' should not be used. For example, they could test the role of NFYC IDR in cells by deleting the IDR from NFYC and investigate if Trim28 depletion leads to upregulation of IAP or Cthrc1 transcripts.
10. The pattern of GAG expression in blastocyst in Fig 5b is inconsistent with what is known in the literature: GAG is known to localise throughout the cytoplasmic of embryonic cells, which has been shown both by EM but also by similar immunostaining experiments (see for example Fig. 6d in PMID 25457166 work by A. Surani). This raises some doubts on the validity of the quantifications and conclusions drawn from these experiments. As with the Fig. 5d (see below), this data do not add much to the manuscript, as it is only correlation.
11. I find Fig. 5d rather misleading. The authors associate temporal phenotype to time of derepression of ERVs in chromatin mutants. However, there are many causes why embryos may die that are unrelated with ERV expression. Without experiments that firmly link onset of expression of ERVs DIRECTLY to a lethality phenotype, this data is misleading and should be removed.
12. The results in Fig. 5g are surprising, since Trim28 KO does not have a penetrant phenotype in terms of Nanog/Oct4 expression, which persists in the KO (Genes Dev 2017, Kumar et al). How do the authors explain these differences? Please discuss.
13. In their model (lines 250-256), the authors suggest that derepression of 'super-enhancers' may explain the lethality observed in early embryos. However, the authors really have no data to make such a claim both, because i) the mutants they analyse have additional, multiple target genes which

could all explain the embryonic phenotype and ii) we do not really even know if superenhancers are actually working/existing/functional in embryos.

Minor comments:

Fig. 1h – please add n number for each box plot.

Fig. 1d – are the TRIM28 levels in the TRIM28-dTAG cell line comparable to the parental, wt cell line?

Many ERVs do not change their expression after Trin28 KD (Fig. S4b and e). Please tone down/adapt conclusions accordingly to reflect specificity of ERVs in a more accurate way.

Fig. 2i – the contact analysis is pretty nice. However, I think it would be most appropriate to do the same analysis but using only DE genes (as opposed to all genes), and compare those with non-DE genes.

The RNA FISH signals do not seem to be consistent across experiments. For example, Fig 2b shows quite a high signal of IAP FISH in the cytoplasm. Also, the pattern of the signal is very different between

The differences in OCT4 levels, as measured by western after overexpression (Fig S14.b) do not really correlate with the expected changes of IAP (by RNA FISH) levels (Fig. 4c-d). Why is so?

HP1 and MED1 graphs in Fig. S13b are repeated/duplicated from Fig. 3g. This is understandable, however the authors should make this explicit in the figure legend to avoid potential future issues of duplication.

The phenotype of Trim28 KO has already been described, and is largely in line with that presented in Fig. S17a. While the current manuscript clearly presents a higher level of characterisation, it would be fair to incorporate at least a citation to previous work by Messerschmidt et al.

The conclusion in lines 213-214 is overstated: the analysis of TEs that the authors done is rather limited to IAP and thus indicating that this is a general phenomenon, specially considering that the authors did not study at all additional/any specific subfamily members, is overstated. Also, the competition experiments that are the basis for this conclusion are based on a very artificial system (GFP overexpression).

Why did the Piggyback experiment uses GFP and not IAP sequence? (Fig. 4h-i)?

Does GFP RNA copy number correlate with droplet formation in in-vitro assays?

Figs S5a and b are missing labels.

N numbers are missing throughout all the (cellular) microscopy. Please amend.

Reviewer #5:
None

Author Rebuttal to Initial comments

General response to reviewers

We thank the referees for their valuable comments and suggestions, which have helped us produce a substantially improved manuscript. The reviewers described findings in the original submission as “highly interesting”, “convincing” and “novel”, and data as “compelling”. The reviewers also noted that the manuscript would benefit from more direct experimental tests of the competition between ERVs and super-enhancers, more rigor in measuring condensate association with genomic loci, better linking in vivo findings with the cell-based results, and more controls and quantifications. We have revised the manuscript to address these and additional concerns.

- In the revised manuscript, we directly tested the competition between ERVs and super-enhancers. We generated mESC lines harboring multiple copies of inducible IAPEz or MMERVK10C ERV fragments, and show that forced transcription of these elements leads to a reduction of super-enhancer transcription, while forced transcription of GFP transgenes with the same copy number, insertion sites and same RNA-transcript length does not.
- To consolidate the condensate localization experiments, we performed live cell super-resolution imaging experiments (PALM) using the genetically labelled *Sox2* super-enhancer locus. We also measured RNAPII condensate localization at an additional super-enhancer locus (*Fgf4*) using fixed cell immunofluorescence. The results reinforce that TRIM28 degradation leads to loss of RNAPII-condensates at multiple super-enhancer loci.
- We quantified transcription of various ERV subfamilies in mouse embryos using scRNA-Seq data, and found that some ERV subfamilies are de-repressed in specific lineages in TRIM28 knockout embryos, and that the most ERV subfamilies are de-repressed in the pluripotent lineages. These data provide new insights into the tissue-specificity of ERV de-repression in heterochromatin-deficient embryos in vivo. The results are consistent with the in vitro and cell-based data, and further support our model that the amount of ERV transcripts correlates with their phenotypic effects.
- We have substantially improved the controls and quantifications of existing and new data. These include: i) a detailed characterization of the cell lines used, including the demonstration that the TRIM28-FKBP tagged mESC line is pluripotent, expresses pluripotency markers, and complements tetraploid embryos, ii) extended Hi-C analyses that further demonstrate contacts between de-repressed ERVs and transcribed genes in TRIM28-degraded cells, iii) multiple approaches of quantifying RNA FISH-IF data.

Taken together the results indicate that ERVs compete with super-enhancers for transcriptional condensates in part through the ability of their RNA transcripts to facilitate condensation of transcriptional activators.

The RNA-mediated condensate hijacking model goes significantly beyond the current consensus view that the importance of repressing ERVs in mammalian cells is because of regulatory elements encoded in their LTRs. The results explain mysteries why ERV activation is associated with embryonic lethality even though virtually all ERVs are inactive in mammals, or why ERV RNAs are heavily modified by m6A methylation. The model further creates a framework to dissect the impact of high amounts of locally produced RNA on the control of cell states during differentiation and various disease contexts.

A detailed response (blue font) to the referees' comments (black font) is below.

Reviewers' Comments:

Reviewer #1:

Remarks to the Author:

In this work, Asimi and colleagues investigated the mechanism by which the transcriptional derepression of endogenous ERVs mediated by TRIM28 inactivation drives early embryonic lethality. To address this question, they developed advanced methodologies to overcome the intrinsic limitations related to gene knockout or knockdown approaches, reaching the required spatial and temporal resolution to dissect the molecular mechanisms involved in this relevant biological process. Specifically, they generated zygotic deletion of TRIM28, which permitted to show that derepression of IAPs occurred at E3.5 blastocysts, concomitantly with the downregulation of the pluripotency factors, leading to perturbation of the epiblast specification and an increase of extraembryonic lineages. To investigate the relationship between derepression of ERVs and downregulation of pluripotency factors they generated knock-in ESCs (and iPSCs) that encode degradation-sensitive TRIM28-FKBP alleles, allowing the stimulation of TRIM28 proteolysis upon exposure to the dTAG-13 ligand. By analyzing nascent transcript levels combined with chromatin profiling upon TRIM28 degradation they measured a concomitant derepression of ERVs and the downregulation of pluripotency genes, which are transcriptionally regulated by super-enhancers (SE). They proposed that the reduced transcription occurring at the SE was caused by the dissociation of the transcriptional condensates from the SE loci, as the ERV transcripts outcompete for RNAP and Mediator assembly in newly transcriptional condensates localized at the ERV loci. In sum they concluded that the derepression of ERVs causes a spatially restricted increment of ncRNAs acting as a nucleation event for the assembly of transcriptional condensates at the expense of the maintenance of the transcriptional hubs at the SE loci, which would cause the downregulation of the associated pluripotency genes.

Although the suggested regulatory mechanism is of high interest and could potentially explain the observed transcriptional regulatory pattern, the provided data do not fully support the drawn conclusions. Specifically, the authors did not demonstrate neither in vitro or within cells that the accumulation of ERV transcript competes for transcriptional condensates at the expense of SE loci. The Piggybac transposon system is an indirect measurement of the proposed mechanism and is not sufficient to support this conclusion. The in vitro assays showed that RNA molecules could enhance phase separation of MED1 and RNAP-CTD, but this phenomenon seems to occur on the basis of the electrostatic interactions with little specificity towards ERV transcripts. The knock-down assay showed that the reduced transcript levels of ERVs was concomitant with the rescue of SE transcript and the increase of the associated pluripotency gene expression. However, this assay suffers of some technical issues that need to be addressed to fully support the main conclusion. In sum the current form of the manuscript is not suitable for publication and it would need a major revision to correctly address the main points raised by the authors.

We thank the reviewer for the useful comments, and challenging us to directly test the competition between ERVs and super-enhancers. We generated mESC lines harboring multiple copies of inducible IAPez or MMERVK10C ERV fragments, and show that forced transcription of these elements leads to a reduction of super-enhancer transcription, while forced transcription of GFP transgenes with the same copy number, insertion sites and same RNA-transcript length does not. In addition, we provide better context for the in vitro experiments, and address the technical issues of the knock-down assays. The new data are described in detail at the specific points below.

Major Issues:

1. If the main conclusion of this work consists in a regulatory mechanism based on the competition between ERV and SE transcripts to nucleate the assembly of transcriptional condensates at specific loci,

then the authors should provide direct evidence for this mechanism, possibly in vitro and in vivo. For example, they could perform the droplet formation assay by incubating the recombinant proteins (MED1 and RNAP-CTD) with labelled eRNAs to a defined molarity, to then add the labelled IAP RNAs at increasing concentrations and measuring the competition between the ncRNA molecules to assemble the droplets. In the presented data (Fig. 3d-h) the authors did not indicate the molarity of the recombinant proteins, so it is difficult to determine the relative efficacy in the droplet formation. This information was provided in ED Fig. 12, in which they measured the effects of increasing amount of IAP RNAs on the droplet formation of RNAP-CTD and MED1. However, the retrieved data showed no linear correlation between the concomitant increase of protein and RNA concentration and the number (or size?) of the assembled droplets (ED Fig. 12a-b). This finding is not clear and is potentially in contrast with the recent work from Henninger et al. Cell 2021 (PMID: 33333019), which showed that the ratio between negative and positive net charges governs the assembly and disassembly of in vitro droplets. However, in the same set of experiments (ED Fig. 13), the authors presented data retrieved from the same assay that indeed showed a concentration-dependent effects of multiple RNA molecules (IAP, major satellite and eRNAs) on the assembly of MED1 droplets, showing that at the highest concentration the assembly is disfavored. How could these results be in agreement with the proposed model of RNA molecule competition for the assembly of the transcriptional condensates? This point needs to be addressed. In addition, they used a very high concentration of the crowding agent PEG (20%) without any explanation: could the authors justify this procedure?

We thank the reviewer for suggesting these important experiments. In the revised manuscript, we directly tested the competition between ERVs and super-enhancers. We generated mESC lines harboring multiple copies of inducible IAPEz or MMERVK10C ERV fragments, and show that forced transcription of these elements leads to a reduction of super-enhancer transcription, while forced transcription of GFP transgenes with the same copy number, insertion sites and same RNA-transcript length does not (new Fig. 4k-m, Extended Data Fig. 17). These data provide important evidence for the specificity of IAPEz and MMERVK10C ERV transcripts on super-enhancer transcription in cells.

We also attempted testing the competition in vitro using the setup suggested by the Reviewer. We pre-equilibrated seRNA and MED1 IDR, and titrated in various concentrations of IAP RNA. We found that RNA-protein droplets quickly age in the in vitro assay, which is concomitant with an arrest of RNA turnover in the droplets. This unfortunately prevented a meaningful use of the in vitro assay for testing competition.

We now include the information of the molarity of the proteins in the in vitro experiments where it was previously missing (new Fig. 4h-j).

Our in vitro data on RNA-MED1 IDR are consistent with data in the Henninger et al (Henninger et al., 2021), and we better explain our results in the context of Henninger et al's findings in the discussion. As the reviewer noted, the Henninger paper shows concentration-dependent effects of RNA on MED1 IDR droplets, whereby low concentration of RNA facilitates, high concentration dissolves condensates. The IAP concentrations used in new Fig. 4e-g, 4i-j, are in the lower part of the concentration spectrum, where RNA facilitates condensation. The IAP concentration in these assays were selected to mimic early activation of IAPs, where the amount of transcript they produce is still moderate. When we increase the concentration (above 100 μ M), we clearly observe condensate dissolution (Extended Data Fig. 16), consistent with Henninger et al's findings. An additional insight that emerged from our in vitro data is that the concentration at which RNA facilitates condensate dissolution is lower for heterochromatin protein 1, than for the MED1 IDR. This finding could explain why heterochromatic regions (e.g. ERVs) tend to be transcribed at a basal level.

We used 10% PEG as crowding agent in the in vitro assays. An equal volume of the 20% PEG stock solution was added to the proteins before the imaging, resulting in a final concentration of 10% PEG. We made this clear in the methods.

2. On the same line, the TT-SLAM-seq results showed by meta-analysis representation (ED Fig. 3c) an accumulation of nascent transcripts at SE, which seems to be way higher than what is measured along gene bodies. Although the same analysis was not performed for the ERV transcripts, it would be expected that the relative accumulation of nascent transcripts at these loci will not exceed the amount measured at the SE or within gene bodies, as highlighted in the screenshot referring to *Cthrc1* locus (Fig. 2e). Can the authors provide this information? If so, how can they explain the possible competition of these relative low abundant transcripts for the assembly of transcriptional condensates with respect to SE and the associated genes? The explanation of few ERVs being localized in proximity of activated genes is not sufficient to justify the proposed mechanism. Although the ERV TKO experiment shows the influence of these elements in the activation of *Cthrc1* it is not possible to exclude that the derepression is necessary for their functioning as de novo enhancers, as previously shown (PMID: 22722858 and 27259207). In addition, in the same experimental setting it has not been clarified whether the other proximal genes are equally sensitive to ERV KO or TRIM28 degradation: how can these divergent results be reconciled in a general mechanism governing transcriptional condensates assembly?

We thank the reviewer for pointing to a discrepancy in our analysis in Extended Data Fig. 3c. In the submitted version, the y-axis for genes and enhancers were not directly comparable, since genes were quantified in strand-specific manner, but y-axis for enhancers represented fragment coverage from both strands. We now include a revised version of the plot where the values for all displayed genomic loci are tags per bp per peak per 10^7 mapped reads, and now the TT-SLAM-Seq values across gene bodies are indeed higher than at super-enhancers (Extended Data Fig. 3c).

We now include repeat-level analysis of the TT-SLAM-Seq data. We plotted the TT-SLAM-Seq reads as a heatmap and metaplot for the ERVs that contain differentially expressed LTRs identified in our data. These analyses revealed that the most highly transcribed ERVs produce nascent transcripts on par with nascent transcripts produced at super-enhancers (new Extended Data Fig. 5d). These results argue that ERV-transcripts could compete with SEs.

We note that the previous models of ERVs functioning as enhancers are compatible and complementary to our proposed model of ERV RNA transcripts facilitating condensation of transcriptional activators. Indeed, ERV LTRs are known to contain transcription factor binding sites and act as enhancers. Our proposal is that the ERV RNA transcripts contribute to impact of ERV de-repression through facilitating condensation of transcriptional activators.

In the revised manuscript we include new data on the expression of proximal genes at the *Cthrc1* locus, as requested by the reviewer (new Extended Data Fig. 10a-e). In brief, we found that most genes within the same topologically associating domain (TAD) with *Cthrc1* experience detectable upregulation in TRIM28-degraded cells.

Last, we also note that condensate assembly and dissolution at genomic sites is a non-equilibrium process. Rather, transcriptional condensates dynamically assemble, dissociate and interact with genomic loci (Cho et al., 2018; Henninger et al., 2021). The consequence of ERV de-repression is that substantially more genomic loci become available to compete for transcriptional activators in TRIM28-deficient cells.

3. The Hi-C assays showed that the majority of ERV taxa switched from B (inactive) to A (active) compartments, showing an increase of contacts with active genes. Although this finding may support the notion that the derepression of ERVs determines their assembly in transcriptionally active compartments,

the provided data do not fully support the conclusion. Indeed, the PC1 ratio shown in ED Fig. 10c showed a very modest increment and without any other reference, it is difficult properly evaluate the biological significance of this measurement. The authors should provide further analyses showing the same analyses with respect to activated genes and or enhancers. In addition, the interactions occurring at the highlighted locus corresponding to chr12, which is characterized by a large stretch of ERV elements in a gene-poor region (Fig. 2h) were not properly measured by the Hi-C experiment, as shown in ED Fig. 10b, possibly due to poor read coverage. On the basis of this, the described switch from B to A compartments should be re-interpreted or demonstrated using orthogonal approaches.

We expanded the Hi-C analyses, including the compartment analyses in Extended Data Fig 11c, to include several genomic features as references. We included upregulated genes, downregulated genes, non-“differentially expressed” genes, as well as the Top 1000 H3K9me3 peaks (based on size of the peaks), and the LTRs defined as ‘differentially upregulated’ in our data. The latter provide a ‘positive’ control in terms of the largest change expected in the TRIM28-degraded cells. The analyses revealed that extend of compartment switch of some ERV taxa are on par with the Top 1000 H3K9me3 peaks measured in our data (new Extended data Fig 11c).

The reviewer is correct that the example ERV-rich locus we included in new Fig 2i (old Fig 2h) indeed is impacted by low read coverage. We note that the poor read coverage affects the sequence of the ERVs themselves, and that the surrounding sequences have substantial read coverage for the analysis to be meaningful. Regardless, we included additional loci with lower ERV density and better read coverage, where the shift in compartment score, and the shift in the corresponding chromatin signatures are readily apparent (new Extended Data Fig. 12a-b).

4. To prove the competition between ERV and SE transcripts for the assembly of the transcriptional condensates within cells, the authors knocked-down the ERV transcripts before inducing TRIM-28 degradation. In this condition, they could retrieve the rescue of the downregulation of both SE and the linked gene expression (Fig. 3a-c and ED Fig. 11). However, the authors did not explain why it is necessary to knock down the ERV transcripts even before they are derepressed. In addition, they did not verify whether the analyzed cells still represent pluripotent stem cells, as the transcript levels of key pluripotency factors are particularly low in this setting. For example, the levels of Klf4, Fgf4 and miR290-295 are much lower with respect to what have been measured by qRT-PCR in the same set of experiments in which the knockdown co-occurred with the TRIM28 degradation (ED Fig. 11a-b) or the time-course analyses shown in ED Fig. 5c. These inconsistencies suggest that possibly TRIM28 degradation drives the exit from pluripotency, as supported by the gene expression analyses performed at later time points. These data bring toward a possible alternative interpretation of the data by which the measured changes in the SE transcript levels in consequence of TRIM28 degradation are not, at least uniquely, depending on the derepression of ERVs and the competition between transcripts may play a marginal role, if any.

In the revised manuscript we performed total RNA-Seq analyses in the knockdown system, and provide evidence that the transcriptome profile of the ERV knockdown cells that lack TRIM28 is similar to the transcriptome profile of the pluripotent cells that do not experience perturbation. In the context of super-enhancers, we found that super-enhancer transcription is globally rescued in ERV knockdown cells that lack TRIM28, to levels observed in the control conditions (new Fig. 4c).

Regarding the necessity to induce shRNA expression before ERVs are de-repressed, we believe there are two major contributing aspects. First, while ERVs are repressed in mESC, some ERVs do experience minimal but detectable transcription. For example, ~1% of TT-SLAM-Seq reads map to ERVs in control cells. This could in part be explained by slightly lower TRIM28 protein levels in the TRIM28-FKBP cell line (new Extended Data Fig 2m), but some level of ERV transcription is also detectable in mESCs

publicly available RNA-Seq data [see e.g. (He et al., 2019)]. Furthermore, the assembly of RNP-complexes that mediate the knockdown likely requires some time. We also note that the knockdown without pre-treatment also has a moderate but detectable rescue of super-enhancer transcription (Extended Data Fig. 14a-b).

Last, we agree with the reviewer that TRIM28 degradation indeed could have downstream effects not related to the de-repression of ERVs. However, the global rescue of the transcriptome profile upon the ERV knockdown in TRIM28-degraded cells indicates that the contribution of the ERV RNA species at least on the short term is important.

5. To address this critical point, the authors should directly measure the contribution of ERV transcripts to the newly assembly of transcriptional condensates at the expenses of the SE-assembled condensates, causing their downregulation. For example, they could monitor in living cells the transient assembly/disassembly of the endogenous MED1 (or RNAP) into condensates at the ERV and SE loci, in response to TRIM28 degradation. By combining live FISH approach (PMID: 31488703) with PALM imaging (PMID: 27138339) they could determine the potential changes in the protein lifetime at the specified loci and the consequence of perturbing ERV or SE transcript levels. Although these are challenging experiments that may require extensive setting, we believe that is necessary to measure the dynamic response of the soluble factors that participate in the assembly of the transcriptional condensates upon the seeding of ERV transcripts.

In the revised manuscript we include live cell super-resolution imaging with PALM (in collaboration with the Cisse lab who are leading experts on this technology) to address condensate-RNA properties in TRIM28 degraded cells in live cells. In brief, we generated an mESC cell line in which nascent transcripts are visualized at the *Sox2* super-enhancer locus, RNAPII molecules are visualized with the photoconvertible dendra2 tag, and TRIM28 alleles are tagged with FKBP. 24h dTAG-treatment in these cells led to a significant reduction in size of the RNAPII cluster nearest to the *Sox2* locus (new Fig. 1k), and an increase in the distance between the *Sox2* locus and the nearest RNAPII cluster (new Fig. 1k), while the size of RNAPII clusters in the cells and average number of RNAPII clusters per cell did not change (new Fig. 1k). These data indicate reduced association of RNAPII condensates at the *Sox2* super-enhancer locus upon acute TRIM28-degradation in live cells, detected at super-resolution. These results provide new insights into condensate features in live TRIM28-degraded cells, and further support that ERV de-repression alters condensate dynamics at super-enhancer loci.

6. Some critical controls are missing: considering that the majority of the experiments rely on a single knock-in clone of ESCs, it is fundamental to provide further data indicating that these cells behave similarly to the parental one. There are no sufficient data showing that this clone maintained the self-renewal capacity and the cell lineage potential compared to the parental cell lines, in both DMSO and dTAG-13 treated (24h) conditions. The authors should provide this information, focusing on the capacity of the cells to give rise to the EpiLCs and also to extraembryonic lineages, by performing the dedicated differentiation assays and measuring the related phenotypes. In addition, the WB showing the efficacy of the degron system should include the TRIM28 protein level of the parental cell line (Fig. 1d), to verify whether the added tag did not alter the protein stability. Similarly, all the gene expression analyses measuring the nascent transcripts were performed by comparing the knock-in cell line treated with DMSO and dTAG-13. However, to ensure that the analyses changes are not in part due to a different cellular state of the analyzed clone, the authors should provide a comparative gene expression analyses between the DMSO-treated knock-in cell line and the parental ESCs.

In the revised manuscript we include a series of controls that demonstrate similarity of the TRIM28-FKBP tagged mESC line to the parental V6.5 line, including analyses of multiple clones.

We demonstrate that the TRIM28-FKBP tagged mESC line is pluripotent using multiple directed differentiation assays. We show that the TRIM28-FKBP tagged mESC line maintains the ability to differentiate into EpiSCs (new Extended Data Fig. 2d-e), into extraembryonic endoderm cells (iXEN) (new Extended Data Fig. 2d, 2f), and into neural precursor cells (NPCs) (new Extended Data Fig. 2d, 2g). We also show that the cell line complements tetraploid embryos, and forms an ICM, while the TRIM28-degraded cells do not (new Extended Data Fig. 2h-i).

Furthermore, we tested the similarity of the global expression profiles between the tagged and parental lines. We performed global RNA-Seq analyses of the parental V6.5 cell line, the TRIM28-FKBP tagged mESC line, and NPCs differentiated from the TRIM28-FKBP tagged mESC line. The results show that the gene expression profile of the TRIM28-FKBP tagged mESC line is similar to that of the parental V6.5 cell line, and both are distinct to the expression profile of NPCs (new Extended Data Fig. 2n).

Moreover, we tested expression of the pluripotency marker SSEA-1 and OCT4 in multiple TRIM28-FKBP tagged mESC clones, and found the levels to be similar to the levels detected in the parental V6.5 mESC line (new Extended Data Fig. 2l-m).

Last, we include TRIM28 Western blots to compare TRIM28 protein levels between the parental V6.5 and TRIM28-FKBP tagged mESC lines. The TRIM28 protein level is slightly lower in the tagged cell line (new Extended Data Fig. 2m). This slight reduction did not seem to substantially alter the pluripotent state of the cells (as detailed above).

7. It is not clear how the number of the differentially expressed genes have been obtained: following what is written in the manuscript the authors used the following parameters: 2-fold change, $FDR < 0.05$. However, on the basis of the available data (Table S4), the 328 downregulated genes are retrieved only by not considering the adjusted p-value cutoff. Is this correct? If so, the number of the differentially expressed genes at 2h and 6h shown in Fig. 1g do not correspond the numbers retrieved from the same table. Please clarify this point and correct it accordingly.

We thank the reviewer for pointing us to this discrepancy. Indeed, the previous version of Table S4 was printed incorrectly, so that p-value was $-\log$ transformed while the header suggested values to be adjusted p-values. In the revised manuscript, Table S4 is corrected, and include adjusted p-values (FDR). The reported numbers of differentially expressed genes were correct in the original Fig. 1g, and are now consistent with the data in Table S4.

8. The authors measured a strong increment of nascent transcripts of different ERV taxa with a clear increment of some classes, such as IAP and MERVK (ED Fig. 4), while other such as MMETn are not shown. However, the ChIP-seq experiments (ED Fig. 9b) show no enrichment for MED23 or RNAP and depositing of H3K27ac at IAP loci: could the authors explain these discrepancies?

The moderate increase in ChIP-Seq signal at IAP loci is in large part explained by not every IAP element being transcribed in TRIM28-degraded cells. Indeed, the MED23, RNAPII and H3K27Ac ChIP-Seq experiments show a moderate but detectable increase in read densities at IAP loci in TRIM28-degraded cells, though at the level of *all* full length IAP loci the increase is moderate (see new Fig 2d. and compare the heatmaps and metaplots in Extended Data Fig. 8b). Also, some IAPs have detectable RNAPII and MED23 signal even in control cells, and these appear not to gain substantial additional signal in TRIM28 degraded cells (see the IAP element at the *Cthrc1* locus shown in Fig. 2e). Overall, the moderate increase in ChIP-Seq signal is consistent with previous reports that show a heterogenous activation of various ERVs elements (including IAP subfamilies) in heterochromatin-perturbed cells (He et al., 2019).

9. The authors adopted a combination of IF-RNA FISH to measure the colocalization of transcription sites with RNAP and MED1. The designed method needs further control to support the quantitative analyses as the authors measured the coefficient of correlation between the two signals at the center of the FISH foci and as a control they choose a random position. Considering that the nascent RNA-FISH will detect at maximum two loci/cell, it is quite expected that a chosen random region will show very poor colocalization, whose signal may only depend to the technical artefacts/background. We recommend to compare the appropriate level of colocalization between SE loci and RNAP with respect to another transcriptionally active region (not SE-enriched) by RNA-FISH. Only this comparison may indicate whether the analysed loci are enriched at the experimentally defined transcriptional condensates, and the relative changes upon TRIM28 degradation.

We further clarify previous controls, and provide several additional controls for the FISH-IF experiments.

First, we use correlation analysis of the FISH and IF signals to compare localization of FISH foci with protein clusters between the DMSO and dTAG-treated cells. In these analyses the correlation of two signals at random positions is not calculated – as the reviewer notes there is no FISH signal at random nuclear positions. Rather, this control shows that the likelihood of observing a protein puncta at a random nuclear positions is lower than observing a protein puncta at FISH foci. Furthermore, we also include analyses of the mean IF intensity at the FISH foci, which shows that the IF signal intensity at the FISH foci is higher than the mean nuclear signal (see e.g. Extended Data Fig. 6c, 6e, 13c). The images displayed as “average IF at random” include a metaplot of the same number of random positions selected as the number of FISH foci included in the analyses. The metaplots show that one is less likely to find a protein cluster at the same number of randomly selected nuclear location than at FISH foci (Fig 1j, 2f-g, 3f, 4d, Extended Data Fig. 6a, 6d, 9d). Moreover, we show that at the *Cthrc1* locus, RNAPII puncta tend to co-localize with the locus, but NRF1 puncta do not (Extended Data Fig. 9d). Importantly, previous reports have already demonstrated that MED1 puncta co-localize with super-enhancer loci but not with highly transcribed genes not associated with super-enhancers. For example, Sabari, 2018 showed that co-localization between the MIR290-295 super-enhancers and MED1 puncta using RNA FISH-IF, and also showed that other transcriptionally active loci not associated with super-enhancers do co-localize with MED1 puncta [see e.g. Fig. S4G in (Sabari et al., 2018)].

10. The authors measured by IF-RNA FISH the colocalization between RNAP, MED1 and NFY with the nascent transcripts of IAPs upon TRIM28 degradation. They claim that the obtained results indicated an increase of colocalization at some IAP loci. However, the obtained measurements by Mean menders colocalizations (MCC) is 0.1, which would indicate that the two signals poorly colocalize. They claim that a similar pattern is obtained by IF for MED1 and MED23; however the quality of these images are so poor (especially for MED23) that it is impossible to rely on these data. Of note, in the same manuscript the authors described a reduction of colocalization between SE and the same proteins when they measured a change of the MCC from 0.86 to 0.73, upon TRIM28 degradation. The interpretation of these data lack consistency. Considering these results, we strongly suggest to reconsidering the interpretations of the data or to perform different measurements of the colocalization. For example, the authors could determine the frequency of the colocalization events/cell, considering the distributed pattern of IAP loci. Alternatively, they could quantify and plot the relative distance between the IF cluster and the FISH foci (see also PIMD: 31494034). Beside that, they should also measure the relative colocalization for other classes of ERVs, considering that IAP loci showed a very modest (if any) enrichment for H3K27ac, RNAP and Mediator retrieved by ChIP-seq.

We thank the reviewer for pointing out the important issue of measuring co-localization between IAPs and IF signal. We attempted to perform unbiased quantitation of the overlap between IAP FISH foci and protein factors. We note that this is a computationally-intense, as-yet-unsolved problem for repetitive

genomic loci using 3D imaging data, and provide several lines of analyses that support association of protein puncta with a single copy ERV-dependent locus.

Indeed, not every IAP RNA FISH focus co-localizes with a protein puncta, and this is expected and consistent with previous work. For example, data from a recent live cell imaging study revealed that condensates are localized within 300nm of a super-enhancer locus visualized by labeling nascent transcripts ~30% of the time over the course of 1 hour imaged in embryonic stem cells [Fig. 4d-e in (Cho et al., 2018)]. When using populations of fixed cells for imaging, this results in protein puncta not ‘perfectly’ co-localizing with genomic loci labeled with RNA FISH in many cells, and not every locus for repetitive loci. In previous papers, the percentage of cells where protein puncta (Mediator, RNAPII, splicing factors) co-localized with super-enhancer loci was between 10-40% within a population of mESCs (Boija et al., 2018; Guo et al., 2019; Sabari et al., 2018; Zamudio et al., 2019). These numbers are consistent with current models that nascent RNA is not a constant structural component of transcriptional condensates, but dynamically associates with them in a concentration-dependent manner (Cho et al., 2018; Henninger et al., 2021). Furthermore, the IAP RNA is a single exon, non-processed transcript. Therefore, the RNA FISH labels every IAP RNA product, not only the ones at the sites of their production. This effect further contributes to not every IAP FISH focus overlapping with protein puncta.

We performed unbiased quantification of the overlap between IAP foci and RNAPII using a 2D projection of the 3D Z-stacks. We calculated a mean Mander’s co-localization for another, larger set of images, resulting in an MCC of 0.193 for RNAPII and MCC of 0.135 for MED1 (new Fig 2a-b). We argue that such analyses are not particularly meaningful as there is no appropriate negative or positive control for such analysis using the repetitive IAP foci. We also quantified the distance between IAP foci and the nearest IF puncta, and found that ~20% of FISH foci are within 200nm of RNAPII, consistent with a distance of regulatory interactions (Cho et al., 2018). We also note that that we performed several additional analyses on a single copy locus (*Cthrc1*) - for which we used intronic RNA probes that label the locus - and for which statistical analyses are more straightforward (Fig. 2f-g), and demonstrate that the transcriptional response at the locus depends on the presence of three ERVs (Fig. 2h). These data collectively indicate the ERV foci can co-localize with Mediator or RNAPII puncta.

The noted discrepancy in colocalization coefficients between SE-IF and IAP-IF colocalization is due to the difference in the type of reported correlation analysis. SE-IF correlation score of 0.86 is not MCC but a Spearman’s correlation coefficient that compares overlap of RNA FISH and IF channels in the average plots. MCC, on the other hand, measures the portion of the intensity in IAP FISH channel that coincide with some intensity in the IF channel across all visualized IAP foci. The two scores are therefore not directly comparable. We added the information on the correlation coefficient to the Figure legends (Fig 1j, 2f-g, 3f, 4d).

Finally, focusing on IAPs for the imaging (FISH) experiments also had technical reasons. IAPs are the most abundant ERVs in the mouse genome, and they are among the youngest family (Thompson et al., 2016; Wells and Feschotte, 2020). Since the sequence of IAPs have not diverged as much as the sequence of other, evolutionarily older ERVs, it is possible to visualize multiple IAP foci with the same set of RNA FISH probes. To illustrate, there are ~3,000 full length IAP loci annotated in the mouse genome, and our FISH probe set routinely detects ~100 nuclear foci. We have performed RNA FISH using probes against the MMETn elements, and found that of the ~300 annotated full length MMETn elements, we could detect one single nuclear focus with the RNA FISH probe set (not shown).

11. To define the possible relationship between SE activation and the assembly of condensates at ERV loci, the authors used Dox-inducible expression of pluripotency factors in iPSCs knock-in for TRIM28 degron system. The obtained results indicate that upon 24h induction of TRIM28 degradation there is a strong decrease of OCT4 and SOX2 protein levels (Fig. 4b), in contrast to what has been achieved in the

ESC clone (Fig. 1d). How do they explain this discrepancy? How can they exclude that these cells are actually exiting from pluripotency independently from the ERV derepression?

We have repeated the Western blot analyses several times, and conclude that OCT4 levels are only moderately lower upon 24h dTAG treatment in the iPSCs (see Fig 3b, and Extended Data Fig 13b), whereas they are not significantly altered upon 24h dTAG treatment in the mESCs. The difference could be explained by differences between cell lines, as the iPSCs are derived from a secondary reprogramming MEF system.

Most importantly – as explained in detail above at point 4 - we performed total RNA-Seq analyses in the knockdown system, and provide evidence that the transcriptome profile of the ERV knockdown cells that lack TRIM28 is similar to the transcriptome profile of the control pluripotent cells that do not experience TRIM28-degradation (new Fig. 4c). We agree with the reviewer that TRIM28 degradation indeed could have downstream effects not related to the de-repression of ERVs. However, the global rescue of the transcriptome profile upon the ERV knockdown in TRIM28-degraded cells indicates that the contribution of the ERV RNA species at least on the short term is important.

12. In most cases it is not clear/specified whether the retrieved quantification of colocalizations were performed on independent biological and/or technical replicates. The same holds true for the qRT-PCR measurements and for the in vitro droplet quantifications.

In the revised manuscript we added the information on the number and nature of replicates in the figure legends (Fig. 1e, 2h, 3e, 3g, 4b, 4c, 4f, 4g, 4h, 4j, 4m, 5c, 5e, 5f, Extended Data Fig. 4b, 4c, 4e, 4f, 7e, 10d, 10e, 14b, 14d, 14f, 14g, 15c, 15e, 15i, 16b, 17d, 17e, 20a, 20b, 21a, 21b, 21c).

Reviewer #2:

Remarks to the Author:

A. Summary of the key results

The work from Asimi et al. set out to answer a fundamental question in TE/ERV biology: how does ERV derepression cause such dramatic effects on early embryos? To this end, they established a system in which they can degrade Trim28 protein post-translationally in order to perform loss-of-function experiments. Since other systems have used Cre-lox or shRNA which only fully deplete Trim28 at later time points (Rowe et al. Nature 2010), this work from Asimi et al. is more elegant and conclusive.

Their major findings:

- 1) Depletion of Trim28 preferentially decreases super-enhancer driven transcription in mESC (Fig 1)
- 2) In the context of Trim28 depletion, ERV RNAs are spatially co-localized with RNA pol-II condensates (Fig 2)
- 3) Surprisingly, this effect is likely mediated through the ability of ERV RNA to direct or stabilize RNA pol-II to other sites and away from super-enhancer driven transcripts. (Fig 3)
- 4) Although the authors first focus on the effect of ERV RNA to confer this effect (Fig 3), they also show that the abundant ERV genomic loci also compete with other OSKM bound loci (canonically super-enhancers) for transcriptional machinery, thus elaborating on another mechanism through which ERV derepression can affect transcription of other genes. (Fig 4)
- 5) Finally, they show that depletion of Trim28 in embryos leads to the loss of the pluripotent compartment in embryos, presumably through a similar interference with OSKM super-enhancer regulation of pluripotency genes. (Fig 5)

B. Originality and significance: if not novel, please include reference

This paper addresses several important unanswered that have dogged the TE/ERV field for many years. It was assumed that TE gene products—specifically retrotranspositionally competent LINE-1s or ERVs (such as MERV-L) were re-integrating into the genome during derepression that the cellular toxicity was mainly mediated through a damage response or dsRNA toxicity. Based on this work, in early embryos and embryonic stem cells, it appears that the RNA produced can re-target the RNA pol-II machinery and potentially that the binding of OSKM into new sites (and away from canonical super-enhancers) or titration of the RNA pol-II away from SE can also contribute to the loss of pluripotency. Both of these distinct mechanisms are extremely important for the TE field to embrace, and the work from Asimi et al. finally provide a compelling molecular mechanism for earlier important work (Jachowitz et al. Nature Genetics, 2017) that unfortunately remained in the realm of phenomenology. I cannot stress how crucial this paper will be for the TE/ERV field, it will reorient many people's view on TE reactivation during early development.

We thank the reviewer for the words of encouragement. It is always inspiring for young trainees when senior colleagues show appreciation for their work!

C. Data & methodology: validity of approach, quality of data, quality of presentation

In general, the manuscript is extremely compelling, easy to read, and the data are clearly presented. As mentioned above in my comments, the strategy of using a Trim28-degron approach is clearly an improvement from earlier Trim28 depletion strategies (Rowe et al. Nature 2010). Although the authors use Crispr/Cas9 to deplete Trim28 in embryos, this will not affect the enormous maternally-deposited stockpile of the protein (described in Messerschmidt, et al. Science 2012) but they instead focus on the effect of Trim28 depletion on the pluripotent compartment (Fig 5)—which is totally appropriate given the pluripotency heavy emphasis for the rest of their manuscript.

There are minor recurring issues with the presentation of some fluorescence data that should be rectified: I would suggest changing the color scheme of dark purple on black (Figs. 1i, 2a-c,f,g, 3c,d,h, 4f, 5b,g) to grayscale in order for the signal to actually be seen.

We replaced the fluorescence images with high resolution images that help to see the fluorescence data better. Furthermore, we use RGB color scale which provide better contrast.

D. Appropriate use of statistics and treatment of uncertainties

In an effort to conform to the Nature Genetics format, I would suggest for each figure legend to briefly mention the statistical test performed. For instance, although Fig 1E, 1G is likely DESeq2 analysis, this is only described in the methods but not the figure legend itself. Additionally, if DESeq2 is used for Fig 1G, would it be more standard to label the y-axis as \log_2 -FDR not p-value?

In the revised manuscript we include the statistical tests used in the legends for each figure panel (Fig. 1e, 1i, 1k, 2h, 3e, 3g, 4b, 4c, 4f, 4g, 4j, 4m, Extended Data Fig. 5c, 6b, 6c, 6e, 7e, 9a, 9c, 10e, 13c, 13e-h, 14b, 14d, 14f, 15c, 15e, 15i, 17e, 17h).

We include now the information that DESeq2 was used for the analysis in Fig. 1g in the figure legends. We also corrected to y-axis to $-\log_2$ FDR (the y-axis was indeed mislabeled before).

For Fig 1H: based on the shape of the boxplot, I can almost guarantee these are not normal distributions, and therefore t-test are inappropriate. Why not use a non-parametric test (Wilcoxon-Mann Whitney), etc? For Fig 1H, it would be appropriate to include the # of each category “n=blank” either in the figure legend or the figure panel itself.

In the revised manuscript we use Wilcoxon-Mann-Whitney test in Fig. 1h. Also, we now include the number in the figure legends.

Fig 2: It would be a “best practice” to quantify the co-localization of IAP ERV RNA with indicated features (RNA pol-II, MED1, NFYA) in Fig 2a,b,c rather than only show a single z-slice, and include the # of images quantified (to match the sample size reporting in Fig 2f, g).

We performed additional imaging series of the IAP FISH-IF experiments at an increased resolution, to facilitate better quantification of the co-localization signal. We replaced all images in Fig. 2a-b with the high-resolution data, and provide additional controls and ways of quantification. In brief, 24 nuclei were imaged and analyzed for both RNAPII and MED1 (We removed the NFYA images from the revised manuscript). Images were acquired over at least 2 fields of view (acquisitions) in high resolution using the Airyscan mode with z-stacks, profiling at least 2 micrometers in the z-dimension. Representative images show single z-slices on the left and maximum intensity projections (of 2.25 micrometer) to the right. Colocalization image is based on the maximum intensity projections. M_{oc} (Mander’s overlap coefficient) score is reported on the image and the text and is based on all the analyzed nuclei.

Maybe the sample size is reported in the methods, but for Fig 3b and 4e,g,i the sample size is not indicated in the figure legend or directly on the figure panel.

In the revised manuscript we include the sample size in the figure legends (new Fig. 3e, 3g, 4b, 4m).

The sample size in 5c,g is not indicated.

We now include the sample size in the figure legends for Fig. 5c, and Fig. 5g.

E. Conclusions: robustness, validity, reliability

The conclusions are robust, and the depth of the mechanistic rigor (particularly for Figs. 3a,b, 4i) is excellent. I was sad to see that the data from the ERV TKO experiment (Extended data Fig 7g) was tucked away in the supplement—as this is really a definitive genetic proof of the effect of the nearby ERVs on *Cthrc1* transcription levels.

We again thank the reviewer for appreciating our data! We now include the ERV TKO qRT-PCR data as a main figure panel (new Fig. 2h), and include additional supporting data for the ERV TKO experiments as new Extended Data Fig. 10a-e.

A point of confusion: the in vitro condensate data from Fig 3d-i: these data would be more impactful if the specificity controls (satellite or eRNAs) were included in the main figure so we can compare them. In Extended data Fig 13b, the authors perform experiments comparing different RNAs and their ability to promote droplet formation. Do the authors claim that IAP RNAs preferentially allow for better droplet formation compared to other RNAs? Is the claim that IAP ERV RNA has special sequence or structural features that mechanistically explain the difference in droplet formation in vitro? X-axis in Fig 3e-g and Extended Data Fig 13b is nM of RNA. Is this nM of individual bases or nM of RNA (polymer) molecules? Are the length of the IAP, satellite, and eRNA the same? Is the eRNA mostly dsRNA or are these non-complementary RNAs?

We believe that any RNA species are likely to facilitate phase separation of transcriptional activators in vitro (Henninger et al., 2021), though this has not been shown for ERV RNA before. We do not claim that the effect is specific to IAP RNA in vitro.

The nM dimension refers to the molarity of the RNA species and not individual bases in the figure legends. The lengths of the RNAs are similar. The sizes are: IAPEz GAG RNA: 820-880 bp, Major Satellite Repeat: 450-550 bp, Mir290-295 seRNA: 930 bp. The Mir290-295 super-enhancer RNA, transcribed RNA of positive and negative strand was mixed 1:1 at equimolar concentrations as previously described (Henninger et al., 2021). We included these information in the methods.

The in vitro nevertheless lend support to the important function of ERV RNAs. One new insight in our in vitro data is demonstrating that the concentration optimum of ERV RNA is different for MED1 and HP1 α . This finding could explain why heterochromatin is transcribed at low levels.

F. Suggested improvements: experiments, data for possible revision

This manuscript is in an extremely advanced form, and in large part, the claims the authors make are well supported by their data. Based on my routine reading of Nature Genetics and the excellent TE/ERV research it publishes, this manuscript meets this very high bar without additional, largely superficial, reviewer-requested experiments that often do not significantly change the major conclusions of the work.

We thank the reviewer for these comments!

G. References: appropriate credit to previous work?

Yes.

H. Clarity and context: lucidity of abstract/summary, appropriateness of abstract, introduction and conclusions

The manuscript is well-written, the abstract is structured well to focus on claims that they investigate in the manuscript, and the intro and conclusion are appropriate.

I would like for the authors to comment on why the Trim28 KO embryo phenotype has such a specific depletion of the pluripotent cell compartment. Are the other cell types (ExEnd and ExE derivatives) somehow less sensitive to ERV reactivation after Trim28 depletion? Is the reason the authors observe such a striking depletion of the epiblast (in Trim28 KO embryos see Fig 5e-g) because ExEnd and ExE rely less on super-enhancers for their cell identity or is it because ERVs are preferentially expressed by TFs present in the ICM/epiblast and therefore loss of Trim28 has an outsized effect on epiblast survival? If it is the latter, TE/ERV expression analysis of the existing Trim28 scRNAseq dataset should reveal if ERVs are derepressed in the ExEnd and ExE and these cell types are just more tolerant of this derepression. This is a relatively minor point, but a cursory clarification on this point in the text would help to frame their ESC data in the context of the in vivo embryo data.

We found that some ERV subfamilies are de-repressed in specific lineages in TRIM28 knockout embryos, and that the most ERV subfamilies are de-repressed in the pluripotent lineages (new Fig. 5e). The results support a model that the amount of ERV transcripts correlates with their phenotypic effects. Together with the knockdown experiments described at Major Comment 6 of Reviewer 3, these results suggest that pluripotent cells are especially vulnerable to high level of ERV transcription. This notion is consistent with findings that the size and distribution of RNAPII and Mediator condensates are different between mESCs and differentiated cells (Cho et al., 2018). We cannot rule out cell-specific functions of TRIM28, but it does appear that RNA production of de-repressed ERVs has a significant contribution to the phenotypic effects of TRIM28 perturbation. We discuss these points in the revised text.

Reviewer #3:

Remarks to the Author:

Endogenous retroviruses (ERVs) occupy about 10% of mammalian genomes and are potentially mutagenic if uncontrolled. Although numerous mouse mutants that cause ERV derepression (including Trim28 KO) are embryonic lethal, whether ERV de-repression per se disturbs early embryonic development is still unclear. In this manuscript, Asimi et al. used the dTAG system to rapidly degrade TRIM28, a factor required by KRABZinc Finger Proteins (KZFPs) to repress many ERVs in mouse ES cells. They report that TRIM28 degradation causes the dissociation of transcriptional condensates at super-enhancers, with a relocalization of condensates at de-repressed/transcribed ERV loci. Further knockdown of ERVs in TRIM28 depleted cells rescues the dissociation of transcriptional condensates at the MiR290-295 locus. In vitro droplet assays demonstrate that IAP RNA can facilitate condensates with RNAPolIII CTD, HP1, and MED1 IDR. The authors then show that forced expression of OSKM factors (Oct4, Sox2, Klf4, c-Myc) during TRIM28 degradation in iPSC suppresses IAP transcripts and foci, restoring expression and transcriptional condensate for MiR290-295. Using a zygotic perturbation platform, the authors KO TRIM28 in embryos and report a correlation between ERV de-repression and the onset of lethality and the loss of specific cell lineages in mouse embryos. Based on these results, the authors propose a hijacking model that ERV derepression can hijack transcriptional condensates from super-enhancers, providing a mechanism to explain “why ERV de-repression may be lethal in early embryos”.

This paper is of interest to readers in the field of epigenetics and endogenous retroviruses (ERVs). It aims to address the acute consequences of ectopic ERV expression upon loss of the adaptor protein TRIM28 (long known to be involved in ERV repression). The experimental systems used in this study are timely and elegant (e.g. overcoming the lethality of Trim28 in ES cells using the dTAG technique, assessing nascent transcripts using TT-SLAM-seq and characterizing mouse embryos using single-cell seq) and appropriate controls are used. Furthermore the “enhancer hijacking model” of ERVs is attractive and relatively novel in this field. Overall, however there are a number of major issues with the manuscript that prevent me from recommending publication. On the positive side, the authors provide some compelling data consistent with their model of redistribution of condensates upon TRIM28 depletion, however it is only shown for one locus (MIR290-295), and some of the imaging data supplied was of too low quality or lacking statistics to allow a more quantitative assessment (specifically the new formation of condensates at IAP loci). Furthermore, there is a major disconnect between the experiments performed in cell culture models and those performed in vivo, and at least one of the experiments performed in cell culture lacks a clearly explained rationale (the induced expression of a GFP from a DNA transposon integrated transgene). Perhaps most importantly however, the authors fail to address the key question set out at the start of the manuscript: whether ERV transcription cause the lethal phenotype. I will detail these concerns below.

We thank the reviewer for the constructive criticism and the useful comments that helped us substantially improve the manuscript. In the revised manuscript i) we include two additional super-enhancer loci where we investigate condensate colocalization, ii) replaced the imaging data with high resolution images and improved quantification, iii) provide new evidence connecting the in vitro and in vivo findings, and iv) include additional experiments that link ERV transcription to the lethal phenotype.

Major points:

1. “TRIM28 degradation leads to the loss of transcriptional condensates at super-enhancers in mESCs” is a key point in this manuscript. However, the loss of transcriptional condensates at super-enhancers is only assessed at the MiR290-295 locus. (A more minor issue is why the authors used the nascent RNA of

MiR290-295 to check the dissociation of transcriptional condensates on the super-enhancer rather than assessing the super enhancer directly (Figure 1i). Moreover, whether loss of transcriptional condensates at super-enhancers after TRIM28 degradation is a generic pattern or just a special case for MiR290-295 is not clear. The ChIP-seq data on RNAPII seems to neither support that transcriptional condensates are generically assembled at super-enhancers, nor supports the generic loss of transcriptional condensates on super-enhancers after TRIM28 degradation (Extended data Fig. 9b).

In the revised manuscript we include analyses of two additional super-enhancer loci, and demonstrate reduced association of RNAPII clusters with the super-enhancer loci in TRIM28-degraded cells. First, we include live cell super-resolution imaging with PALM (in collaboration with the Cisse lab who are leading experts on this technology) to address condensate-RNA properties in TRIM28 degraded cells in live cells. In brief, we generated an mESC cell line in which nascent transcripts are visualized at the *Sox2* super-enhancer locus, RNAPII molecules are visualized with the photoconvertible dendra2 tag, and TRIM28 alleles are tagged with FKBP. 24h dTAG-treatment in these cells led to a significant reduction in size of the RNAPII cluster nearest to the *Sox2* locus (new Fig. 1k), and an increase in the distance between the *Sox2* locus and the nearest RNAPII cluster (new Fig. 1k), while the size of RNAPII clusters in the cells and average number of RNAPII clusters per cell did not change (new Fig. 1k). These data indicate reduced association of RNAPII condensates at the *Sox2* super-enhancer locus upon acute TRIM28-degradation in live cells. The results provide new insights into condensate features in live TRIM28-degraded cells, and further support that ERV de-repression alters condensate dynamics at super-enhancer loci. Furthermore, we analyzed RNAPII cluster co-localization at the *Fgf4* super-enhancer locus. We found that that RNAPII puncta show reduced co-localization with the *Fgf4* locus in TRIM28-degraded cells (new Extended Data Fig. 6d-e).

For the *miR290-295* locus, we used RNA FISH probes that were previously validated (Sabari et al., 2018).

We also note that co-localization of RNAPII and Mediator condensates with super-enhancers have been described in prior literature at several additional SE loci, and that our ChIP-Seq data is on par with the data in those papers (Guo et al., 2019; Sabari et al., 2018).

2. “De-repressed IAPs form nuclear foci that associate with RNAPII condensates” is another key point for the hijacking model. Although some IAP foci appear to co-localized with RNAPII condensates, there are many more IAP foci that are not co-localized with RNAPII condensates (Figure 2a). It should be tested whether IAP foci are significantly overlapped with RNAPII condensates or not compared with other transcribed regions (Figure 2a). In addition, the images for the co-localization of IAP with other parts of transcriptional condensates (i.e., MED1 and MED23) seem to have high background, I cannot even assess whether MED1 and MED23 signals are enriched in the nucleus (Figure 2b and Extended data Fig. 7b).

We thank the reviewer for pressing on the important issue of measuring the co-localization between IAPs and IF signal. We attempted to perform unbiased quantitation of the overlap between IAP FISH foci and protein factors. We note that this is a computationally-intense, as-yet-unsolved problem for repetitive genomic loci using 3D imaging data, and provide several lines of analyses that support association of protein puncta with a single copy ERV-dependent locus.

Indeed, not every IAP RNA FISH focus co-localizes with a protein puncta, and this is expected and consistent with previous work. For example, data from a recent live cell imaging study revealed that condensates are localized within 300nm of a super-enhancer locus visualized by labeling nascent transcripts ~30% of the time over the course of 1 hour imaged in embryonic stem cells [Fig. 4d-e in (Cho et al., 2018)]. When using populations of fixed cells for imaging, this results in protein puncta not ‘perfectly’ co-localizing with genomic loci labeled with RNA FISH in many cells, and not every locus for

repetitive loci. In previous papers, the percentage of cells where protein puncta (Mediator, RNAPII, splicing factors) co-localized with super-enhancer loci was between 10-40% within a population of mESCs (Boija et al., 2018; Guo et al., 2019; Sabari et al., 2018; Zamudio et al., 2019). These numbers are consistent with current models that nascent RNA is not a constant structural component of transcriptional condensates, but dynamically associates with them in a concentration-dependent manner (Cho et al., 2018; Henninger et al., 2021). Furthermore, the IAP RNA is a single exon, non-processed transcript. Therefore, the RNA FISH labels every IAP RNA product, not only the ones at the sites of their production. This effect further contributes to not every IAP FISH focus overlapping with Mediator or RNAPII puncta.

We performed unbiased quantification of the overlap between IAP foci and RNAPII using a 2D projection of the 3D Z-stacks. We calculated a mean Mander's co-localization for another, larger set of images, resulting in an MCC of 0.193 for RNAPII and MCC of 0.135 for MED1 (new Fig 2a-b). We argue that such analyses are not particularly meaningful as there is no appropriate negative or positive control for such analysis using the repetitive IAP foci. We also quantified the distance between IAP foci and the nearest IF puncta, and found that ~20% of FISH foci are within 200nm of RNAPII, consistent with a distance of regulatory interactions (Cho et al., 2018). We also note that that we performed several additional analyses on a single copy locus (*Cthrc1*) - for which we used intronic RNA probes that label the locus - and for which statistical analyses are more straightforward (Fig. 2f-g), and demonstrate that the transcriptional response at the locus depends on the presence of three ERVs (Fig. 2h). These data collectively indicate the ERV foci can co-localize with protein clusters.

Focusing on IAPs for the imaging (FISH) experiments also had technical reasons. IAPs are the most abundant ERVs in the mouse genome, and they are among the youngest family (Thompson et al., 2016; Wells and Feschotte, 2020). Since the sequence of IAPs have not diverged as much as the sequence of other, evolutionarily older ERVs, it is possible to visualize multiple IAP foci with the same set of RNA FISH probes. To illustrate, there are ~3,000 full length IAP loci annotated in the mouse genome, and our FISH probe set routinely detects ~100 nuclear foci. We have performed RNA FISH using probes against the MMETn elements, and found that of the ~300 annotated full length MMETn elements, we could detect one single nuclear focus with the RNA FISH probe set (not shown).

Last, the MED1 antibody is known to cross-react with a cytoplasmic protein, and the cytoplasmic artefactual staining for Mediator antibodies is known in the literature (Boija et al., 2018; Guo et al., 2019; Sabari et al., 2018). We validated the nuclear signal of the MED1 antibody being MED1 using a degradation sensitive MED1-FKBP KBM7 cell line (see reviewer Figure 1 below). Also, some of the IF signal outside highlighted nuclear regions comes from neighboring ESCs, as mESCs are known to stick together and forming colonies under the serum+LIF conditions we culture them.

Reviewer Figure 1.
Testing specificity of the MED1 IF staining using a degron-tagged KBM7 cell line. Yellow arrows highlight cytoplasmic signal in dTAG-treated cells.

3. To test whether de-repression of ERVs can form and hijack transcriptional condensates, the authors mainly use IAPs as a model through the whole manuscript. However, the ChIP-seq data do not support that components forming nuclear condensates, namely RNAPII and Med23, are largely moved to IAP regions after TRIM28 degradation (Extended data Fig. 9b). And further knocking down IAPs in TRIM28 depleted cells did not rescue the expression of miR290-295 and Fgf4 or their super-enhancers (Extended Data Fig. 11d), indicating IAPs may not hijack transcriptional condensates in the way described.

We thank the reviewer for raising this important point. Our model is that ERVs have the capacity to contribute to condensates at ERV-proximal loci. We do not propose that this capacity is specific to IAPs, nor that it is only the ERVs that contribute to the hijacking effect at genomic loci. Rather, it is the amount of the locally produced RNA which ‘traps’ the transcription machinery through the RNA’s ability to facilitate phase separation of transcriptional activators. We clarified this in the Discussion.

Several lines of evidence in the data cited by the reviewer are consistent with our proposed model. For example, super-enhancers are known to be sites where transcriptional condensates associate with the genome (Boijja et al., 2018; Cho et al., 2018; Guo et al., 2019; Sabari et al., 2018; Zamudio et al., 2019). The *Cthrc1* locus (as shown in Fig. 2e) is not classified as a super-enhancer in DMSO-treated mESCs, but it is classified as a super-enhancer in the dTAG-treated cells using the H3K27ac ChIP-Seq data and the ROSE algorithm (Whyte et al., 2013). So the seemingly moderate changes in the chromatin signatures that occur at the three ERVs at the locus do appear to have important contributions to the overall signal density observed at the locus. Moreover, not every IAP, MMERVK10C, MMERVK9C, MMETn in the genome get induced to the same level in TRIM28-degraded cells. Rather, it is a relatively small population of ERVs that get highly transcribed, and the signal (e.g. TT-SLAM-Seq, or ChIP-Seq) at those elements are comparable to the density of signal at super-enhancers (see e.g. new Extended Data Fig. 5d).

Overall, we wish to stress that our model is not specific to IAPs. Rather, it is the amount of the locally produced RNA which ‘traps’ the transcription machinery through the RNA’s ability to facilitate phase separation of transcriptional activators. We made clarifications throughout the manuscript text.

4. The authors failed to activate IAPs with CRISPR approaches (as have many labs!) and they instead used a dox-inducible GFP integrated as part of a DNA transposon-based vector to mimic the expression of an interspersed ERV (Figure 4h-I and Extended Data Fig. 15). The rationale for this choice is not clear or well explained. Neither the sequence expressed (GFP), the regulatory sequences, nor the copy number strongly resembles the ERV families in question (IAPs). It is my guess they are arguing that expressing any repeat locus to high levels could hijack the transcriptional machinery, and I am not sure what value such a statement makes with regards to the function of TRIM28 and ERVs. It is also hard to imagine why clones with “high” GFP expression have such altered expression of some super-enhancers and their target genes but clones with “middle” GFP expression have totally no influence (Figure 4i and Extended Data Fig. 15d), despite only a two-fold reduction in expression (extended data Fig. 15c). Is there really such a sweet spot for transcription of a repeat that would hijack these condensates? Would a single locus work if the expression levels were high enough?

In the revised manuscript, we directly tested the competition between ERVs and super-enhancers. We generated mESC lines harboring multiple copies of inducible IAPEz or MMERVK10C ERV fragments, and show that forced transcription of these elements leads to a reduction of super-enhancer transcription, while forced transcription of GFP transgenes with the same copy number, insertion sites and same RNA-transcript length does not (new Fig. 4k-m, Extended Data Fig. 17). These data provide important evidence for the specificity of IAPEz and MMERVK10C ERV transcripts on super-enhancer transcription in cells.

We agree with the reviewer that the transposon system expressing only GFP did not adequately test competition between ERVs and removed those data from the revised manuscript.

We also believe that a sweet spot of repeat transcription where it becomes inhibitory to super-enhancers – as suggested by the reviewer - likely exists, and is in part a combination of the copy number of the activated loci and strength of recruitment of the transcription machinery. For example, in Figure 6C of a previous study, sustained induction of an integrated single-copy transgene driven by a strong promoter appeared to alter Mediator condensates in nuclei (Henninger et al., 2021). We clarified the role of RNA in the revised Discussion.

5. The molecular characterization of the mouse embryos furthers our understanding of the Trim28 phenotype, by demonstrating a loss of pluripotent lineages and expression of ERV-derived proteins. This analysis in vivo seems however, rather disconnected from the experiments in vitro: functional links between ERV expression and the loss of pluripotent lineages are not assessed/observed in ESCs and the pluripotency gene and other markers assessed are not linked to the loci assessed in the first part of the paper. This needs to be rectified.

In the revised manuscript we include new analyses that establish a link between the in vitro and in vivo findings. We quantified transcription of various ERV subfamilies in mouse embryos using the scRNA-Seq data. We found that some ERV subfamilies are de-repressed in specific lineages in TRIM28 knockout embryos, and that the most ERV subfamilies are de-repressed in the pluripotent lineages (new Fig. 5e). These data provide new insights into the tissue-specificity of ERV de-repression in heterochromatin-deficient embryos in vivo. The results are consistent with the in vitro and cell-based data, and further support our model that the amount of ERV transcripts correlates with their phenotypic effects.

In the revised manuscript we performed total RNA-Seq analyses in the knockdown system, and provide evidence that the transcriptome profile of the ERV-knockdown cells that lack TRIM28 is similar to the transcriptome profile of the pluripotent cells that do not experience perturbation. In the context of super-enhancers, we found that super-enhancer transcription is globally rescued in ERV knockdown cells that lack TRIM28 to levels observed in the control conditions (new Fig. 4c). These data further reinforce the link between ERV expression and loss of pluripotency in mESCs.

6. A major question posed in the manuscript is whether/why ERV de-repression may be lethal in early embryo and whether a redistribution of condensates away from super-enhancers can explain the lethality. However, the evidence for whether IAP de-repression causes lethality is still very weak. TRIM28 deletion or depletion causes many molecular phenotypes in ESCs and in vivo in addition to those phenotypes described in this manuscript relating to nuclear condensates. For example, Trim28 depletion causes ERV and potentially toxic gene de-repression, loss of imprinting, loss of heterochromatin, accumulation of DNA damage, activated retrotransposition of ERVs and LINEs, formation of viral particles, amongst others, that could all contribute to lethality. None of these have been ruled out nor extensively discussed. There is also data from KZFP cluster KO ESCs and embryos that demonstrate that ERV de-repression is compatible with embryogenesis. How do the authors reconcile this with their model? Is it the extent of derepression that matters?

In the revised manuscript, we provide additional evidence supporting the link between ERV-derepression and lethality. (We stress that we do not think the effect is specific to IAPs, see above at Comment 3).

In the revised manuscript we performed total RNA-Seq analyses in the knockdown system, and provide evidence that the transcriptome profile of the ERV-knockdown cells that lack TRIM28 is similar to the transcriptome profile of the control pluripotent cells that do not experience TRIM28 degradation. In the context of super-enhancers, we found that super-enhancer transcription is globally rescued in ERV knockdown cells that lack TRIM28 to levels observed in the control conditions (new Fig. 4c). These

results indicate that ERV knockdown can at least on the short term rescue the effect of TRIM28-degradation in mESCs.

As described in detail at Comment 5, we quantified transcription of various ERV subfamilies in mouse embryos using the scRNA-Seq data. We found that some ERV subfamilies are de-repressed in specific lineages in TRIM28 knockout embryos, and that the most ERV subfamilies are de-repressed in the pluripotent lineages (new Fig. 5e). The results are consistent with the in vitro and cell-based data, and further support our model that the amount of ERV transcripts correlates with their phenotypic effects.

We agree with the reviewer that our analyses do not rule out effects of TRIM28-degradation other than ERV de-repression contributes to cellular effects. Regardless, the knockdown data suggest that ERV knockdown can at least on the short term rescue the effect of TRIM28-degradation, supporting the model that indeed the extent of de-repression matters.

7. In this work, the authors use shIAP, shMMEVK10c, shMMERVK9c and shMMETn to knock down these ERVs during TRIM28 degradation. However, qPCR primers used to measure MMETn and MMERK9c elements failed as stated by the authors (Lines 619-621 and extended data Fig. 11b). In this situation, how do the authors know whether the shMMETn/MMERK9c assays succeeded or not?

In the revised manuscript we performed total RNA-Seq analyses in the knockdown system. Using the sequence reads, we could confirm knockdown of the IAPs, MMERVK10s, and MMERVK9c elements (new Extended Data Fig. 14g).

8. For the scRNA-seq in embryos, the authors performed scRNA-seq for the wild-type embryos at E5.5 and Trim28 KO at E6.5 and downloaded those data for wild-type E6.5 and E7.0 from GEO. The proportion of cell populations from wild-type E5.5 and Trim28 KO E6.5 present obviously larger variance than these from GEO (wild-type E6.5 and E7.0) (Extended Data Fig. 17a), indicating there are significant issues during embryo isolation and/or cell sampling for scRNA-seq. In addition, how TRIM28 KO embryos are identified and how the authors deal with the mosaic samples (if Trim28 is partially KO in the embryos) is unclear. These should be addressed.

For wild type embryos at E5.5, the observed variation between individual embryos may be accounted for by the technical difficulty of dissociating these early-stage embryos. Due to the presence of Reichert's membrane, extra-embryonic lineages are generally more challenging to obtain intact. Nonetheless, it may occasionally result in the loss of non-embryonic lineages such as ExE/trophoblast, Parietal, and Visceral endoderm. In comparison to E6.5 and E7.0 embryo isolation, E5.5 embryos require nearly double the time to dissociate, resulting in a greater loss/drop-out of cells with low viability and a high mitochondrial count. This also applies to TRIM28 KO embryos, which are less susceptible to enzymatic dissociation and require mechanical disaggregation (new Extended Data Fig. 18i). Reichert's membrane was carefully peeled off before dissociation. We took care to avoid processing maternal tissue for the scRNA-seq experiment. The phenotype of TRIM28 KO embryos was consistent with our data, which indicated a deficiency of epiblast (no discernible separation of embryonic/extra-embryonic compartments) and an excess of parietal endoderm (cobble-stone-like structure encircling the KO embryos) (new Extended Data Fig. 18i). For this purpose, E6.5 TRIM28 KO was compared only to E6.5 wild type embryos, performed under the same conditions.

As previously demonstrated (Andergassen et al., 2021; Grosswendt et al., 2020; Smith et al., 2017) our platform for zygotic perturbation generates highly efficient KO embryos with no detectable mosaicism. In the revised manuscript, we show that TRIM28 sgRNAs have a high KO efficiency by target site analysis (new Extended Figure 18b) and immunofluorescence (Extended Data Fig. 18c). Additionally, all embryos

profiled exhibit de-repression of repeat elements (new Fig. 5e), not observed in heterozygous/mosaic embryos (Rowe et al., 2010).

Minor Points:

1. Parts of the abstract may be difficult to understand for non-specialists, e.g.
 - a. Line 39: NFY may not be a widely known factor
 - b. Line 45/46: “nascent RNAs may facilitate “hijacking” of transcriptional condensates in various developmental and disease contexts”. It may not be obvious to the reader what hijacking of transcriptional condensates means.

We revised the abstract, and attempted to improve its clarity.

2. Line 32 “ERV transcription or RNA transcripts may underlie essentiality of ERV repression”. To our knowledge the essentiality of ERV repression for the Trim28 KO phenotype has not been formally proven experimentally. We thus suggest rewording this sentence e.g. by adding ‘potential/suggested essentiality of ERV repression’. The main text sentence line 60/61 is more accurate “why ERV de-repression may be lethal in early embryos and pluripotent cells is a mystery”.

We reworded the sentence, which now reads “RNA transcripts produced by ERVs may contribute to developmental phenotypes associated with ERV de-repression.” (lines 54-55)

3. Figure 1a and Extended Figure 1b and Extended Figure 1a and 4a are identical. Is the repetition of this figure in the extended data figure necessary?

We believe the repetition helps non-specialist readers navigate the relationship between the ERV families and subfamilies when the schemes are present on the figures with the most relevant data.

4. Line 144/145: “These results demonstrate that transcriptional condensates may incorporate genes and de-repressed ERVs”. While the data in this paragraph does demonstrate that combined KO of nearby ERVs rescue the phenotype at this locus (i.e. no upregulation of the nearby transcript is observed) it does not show that this rescue relies on the formation of new transcriptional condensates. As the ERVs are also proximal to 2 other genes, Slc25a32 and Dcaf13, it would also be of interest to know if these genes are also upregulated upon Trim28 KO. Do the authors think these ERVs act as enhancers, e.g. do they contain histone marks indicative of putative enhancers?

In the revised manuscript we include new data on the expression of proximal genes at the *Cthrc1* locus, as requested by the Reviewer (and Reviewer 1), (new Extended Data Fig. 10a-e). In brief, we found that most genes within the same topologically associating domain (TAD) with *Cthrc1* experience detectable upregulation in TRIM28-degraded cells. We note that some ERVs indeed are occupied by enhancer-associated H3K27Ac histone marks (Extended Data Fig. 8b).

5. KD of the four most-induced ERV families partially rescued the downregulation of super-enhancers particularly when shRNAs were administered 24h prior to Trim28 degradation. What about other genes (genome-wide changes) or embryonic lethality? Are these rescued?

In the revised manuscript we performed total RNA-Seq analyses in the knockdown system, and provide evidence that the transcriptome profile of the ERV knockdown cells that lack TRIM28 is similar to the transcriptome profile of the pluripotent cells that do not experience perturbation. In the context of super-

enhancers, we found that super-enhancer transcription is globally rescued in ERV knockdown cells that lack TRIM28 to levels observed in the control conditions (new Fig. 4c).

We also provide experimental evidence for the importance of condensates for activating the *Cthrc1* gene in TRIM28-degraded cells. We performed experiments using 1-6 hexanediol (1-6 HD) treatment of control and TRIM28-degraded cells. Short term 1-6 HD treatment dissolved RNAPII puncta in cells, and compromised *Cthrc1* induction after TRIM28-degradation (Extended Data Fig. 7c-e).

6. How do the authors explain that in the absence of Trim28 pluripotent cells in the embryo are gradually depleted while ESCs seem to not alter pluripotency marker expression? Please discuss.

Our data indicate that long term TRIM28 degradation does lead to a gradual loss of pluripotency marker expression both in embryos (Fig. 5) and mESCs (Extended Data Fig. 21). Specifically, we show that 2-6h of TRIM28-degradation in mESCs does not substantially alter the expression profile of the cells, including the expression level of pluripotency markers. Some reduction of the expression levels of pluripotency genes is detectable after 24h TRIM28-degradation. After 94h of TRIM28 degradation, the expression of pluripotency genes is markedly downregulated, and differentiation markers are induced (Extended Data Fig. 21b). The induced genes include GATA6, which is a gene markedly upregulated in the 'ICM' of TRIM28 KO embryos (Fig. 5f). These data suggest similarities between the impact of sustained TRIM28 degradation in mESCs and embryos.

7. Line 87-88: were the down-regulated genes enriched specifically for super-enhancer associated pluripotency genes or pluripotency genes overall? How are enhancers less transcribed upon acute loss of Trim28 when Trim28 is not bound to these enhancers or is it?

The down-regulated genes are enriched for super-enhancer associated genes (new Fig. 1h). Super-enhancer -associated genes were previously shown to be highly enriched in pluripotency genes in mESCs (Whyte et al., 2013). We moved these data to a main figure panel to improve clarity (new Fig. 1h).

Our model is that enhancers are less transcribed upon acute loss TRIM28 (even though TRIM28 is indeed not bound at enhancers), because sporadic transcription at de-repressed ERV loci attract ('hijack') transcriptional activators away from enhancers in part through the effect of their RNA transcripts mediating condensation of transcriptional activators.

8. Ext Figure 4b+c: Does TT-SLAM-seq not detect LINE and SINE elements or are these simply not nascently transcribed in ESCs? These have previously been shown to be derepressed in Trim28 KOs.

We see little to no evidence of LINE and SINE de-repression in our data. The TT-STAM-Seq data is sparse in reads mapping uniquely to LINE and SINE elements, as the reads have low complexity because of the T-C conversion. We observed a marginal increase in L2 LINE elements in the RNA-Seq data after 24h of dTAG-13 -treatment (Extended Data Fig. 4c). We note that the original papers on TRIM28 perturbation also found minimal increase of LINE expression and no detectable increase in SINE expression following genetic deletion of *Trim28*, consistent with our data (Rowe et al., 2010; Rowe et al., 2013).

9. Can the authors speculate what could cause the reduction of pluripotent cells in TRIM28 mutant blastocysts and gastrulation-stage mouse embryos? Next to de-repressing ERVs, Trim28 seems to regulate critical developmental TFs such as e.g. *Fgf4* (Fig 1g) that impair proliferation of the inner cell mass when lacking (Feldmann 1995) are responsible? Please comment/discuss.

We found that some ERV subfamilies are de-repressed in specific lineages in TRIM28 knockout embryos, and that the most ERV subfamilies are de-repressed in the pluripotent lineages (new Fig. 5e). The results support a model that the amount of ERV transcripts correlates with their phenotypic effects. Together with the knockdown experiments described at Major Comment 6, these results suggest that pluripotent cells are especially vulnerable to high level of ERV transcription. This notion is consistent with findings that the size and distribution of RNAPII and Mediator condensates are different between mESCs and differentiated cells (Cho et al., 2018). We cannot rule out cell-specific functions of TRIM28, but it does appear that RNA production of de-repressed ERVs has a significant contribution to the phenotypic effects of TRIM28 perturbation.

10. Line 228: “To probe which cell types are affected by the presence of IAP foci in blastocysts...” > this experiment does not probe effects of the IAP foci. Please change the wording accordingly or test functionally.

We reworded the sentence: “To probe which cell types are affected by de-repression of ERVs...”

review signed by Todd Macfarlan

Reviewer #4:

Remarks to the Author:

The manuscript by Asimi and colleagues presents a very thorough molecular investigation of mouse ES cells upon acute depletion of Trim28/KAP1. The authors characterise changes in H3K9me3, H3K27ac, HP1a and binding of OCT4, SOX2 and Nanog (OSN). Their main conclusions are that some TEs are derepressed and transcription (and most likely associated activity) of eRNAs from pluripotency-associated enhancers bound by OSN are downregulated. The authors propose that TE transcription competes with enhancer transcription, presumably indirectly by ‘hijacking’ RNA polIII. To support these conclusions, they present analysis of RNA PolIII ChIP and microscopy-based analyses of RNA PolIII and RNA-FISH for TEs and a chosen pluripotency-enhancer driven gene upon depletion of Trim28 as well as an ‘overexpression’ approach with GFP.

In general, this is a very interesting paper and I find it convincing and novel but I do not think that the phase separation (PS) part adds much to the manuscript, nor to the mechanistic investigation of the observations therein, and the quality of this data is in general less good. All the section on PS is very general and it does not really explain the underlying e.g. competition hypothesis and only general conclusions can be drawn from this part (e.g. that i) RNA is important– which we know- but not really specific for ERVs and ii) that hexanediol affects transcription in general, which we know). While the role of Trim28 in regulating TEs is well known in the literature, the effects on pluripotency/enhancers are completely novel and well documented by the authors in this manuscript.

We thank the reviewer for the constructive criticism and guidance on improving the manuscript. In brief, we added new data that address phase separation in live cells using super-resolution imaging (PALM), and provide evidence for specificity of ERV RNAs in cells. The data, as well as several additional experiments and analyses that further strengthen the paper are described in detail below. Furthermore, we provide new evidence for the specificity of ERV RNAs in competing with super-enhancers in cells.

Overall, my comments regard the addition of some missing controls and quantifications throughout and strengthening the PS section, as follows:

1. Authors relate RNA PolIII staining (and ChIP) to RNA PolIII activity, but they used an antibody recognising the non-phosphorylated (CTD) RNA polIII, which is not the best proxy for RNA polIII activity – can the authors validate (at least some of their key findings) using a CTD-phosphorylated, active RNA pol II antibody?

In the revised manuscript we include live cell super-resolution imaging with PALM (in collaboration with the Cisse lab who are leading experts on this technology) to address condensate-RNA properties in TRIM28 degraded cells in live cells. In brief, we generated an mESC cell line in which nascent transcripts are visualized at the *Sox2* super-enhancer locus, RNAPII molecules are visualized with the photoconvertible dendra2 tag, and TRIM28 alleles are tagged with FKBP. 24h dTAG-treatment in these cells led to a significant reduction in size of the RNAPII cluster nearest to the *Sox2* locus (new Fig. 1k), and an increase in the distance between the *Sox2* locus and the nearest RNAPII cluster (new Fig. 1k), while the size of RNAPII clusters in the cells and average number of RNAPII clusters per cell did not change (new Fig. 1k). These data indicate reduced association of RNAPII condensates at the *Sox2* super-enhancer locus upon acute TRIM28-degradation in live cells, detected at super-resolution. These results provide new insights into condensate features in live TRIM28-degraded cells, and further support that

ERV de-repression alters condensate dynamics at super-enhancer loci. Of important note, RNAPII condensates in this system are visualized using an endogenous tag, and not antibodies.

Furthermore, we note that the 8WG16 RNAPII antibody which was originally raised against the hypophosphorylated CTD, seems to act as a pan-RNA Polymerase II antibody. Signal with this antibody is detected across the entire body of genes, and this antibody has been used to visualize RNAPII-condensates associating with super-enhancer loci (Guo et al., 2019).

2. In Fig. 1j, the colocalization is hard to evaluate: there is also quite some signal ‘adjacent’ to the mir290-295 FISH. The shape of the FISH spot in the merge and in the inset looks very different, it would seem that image contrast has been adjusted, perhaps unequally. The authors should show raw data. Along the same lines, the data would be much more robust if the authors could provide better quantifications throughout their co-localisation datasets: they used Mander’s coefficient for some of their data, but not for all. Also, it would be useful if they would provide an estimate of the variability across cells.

The inset in the image shows the zoom-in of the FISH focus in the one cell, whose images are displayed in the same panel. The average image is a meta-plot of the FISH foci of all nuclei that were used in the global analysis (Fig 1j, 2f-g, 3f, 4d, Extended Data Fig. 6a, 6d, 9d, 14j). This explains why there could be differences in the shape of the 2D projection of single FISH focus vs. a meta representation of dozens of foci.

The contrast was adjusted the exact same way for all images found on the same figure panel. Regardless, we are in the process of depositing all raw imaging (and all other raw data) in public repositories – which we do routinely for all our publications.

We further clarify previous controls, and provide several additional controls for the FISH-IF experiments. First, use the correlation analysis of the FISH and IF signals to compare localization of FISH foci with protein clusters between the DMSO and dTAG-treated cells. Furthermore, we also include analyses of the mean IF intensity at the FISH foci, which shows that the IF signal intensity at the FISH foci is higher than the mean nuclear signal (see e.g. Extended Data Fig. 6b, 6c, 6e, 13c). The images displayed as “average IF at random” include a metaplot of the same number of random nuclear positions selected as the number of FISH foci included in the analyses. The metaplots show that one is less likely to find a protein cluster at the same number of randomly selected nuclear location than at FISH foci (Fig 1j, 2f-g, 3f, 4d, Extended Data Fig. 6a, 6d, 9d, 14j).

To address variability between cells, we also show that in the IF experiments the total level of RNAPII and number of RNAPII do not change (Extended Data Fig. 6b, 6c, 6e, 13c). Also, as described in detail at Comment 1 above, we corroborated the fixed cell FISH-IF imaging with live cell super-resolution imaging (new Fig. 1k), which include additional controls (described above).

3. Authors conclude on *Cthrc1* FISH colocalization/lack of colocalisation with NFYA based on images shown in Fig. 2f. However, NFYA is all over the place and so the probability of ‘any’ FISH to colocalise with NFYA is probably very high and the relevance of this potential colocalization is therefore a little overstated. Perhaps including quantifications of number of FISH spots that co-localise (or not) with NFYA and the inverse analysis may help to strengthen this conclusion. The Mander coefficient is a good proxy for global colocalisation, but it does mask additional important information of the number of events when this is observed.

We agree that the NFY signal is somewhat diffuse, therefore, we removed the IAP-NFYA analyses from the revised paper, and simplified the statements we make based on the *Cthrc1*-NFYA data. Moreover, as a control for these experiments we show that at the *Cthrc1* locus, the RNAPII IF signal is higher than the

mean nuclear NFYA signal in the same nuclei (Extended Data Fig. 9c). As an additional control, we also show that puncta of a control transcription factor NRF1, do not co-localize with *Cthrc1* FISH foci (Extended Data Fig. 9d).

4. Is droplet formation (Fig. 3c) affected by the phospho status of the CTD ?

The droplet formation by purified RNAPII CTD is known to be affected by phosphorylation (Boehning et al., 2018). As described above at Comment 1, we provide live-cell super-resolution imaging data on RNAPII condensates using an endogenous fluorescent tag, therefore, we did not pursue this angle of the in vitro data further (new Fig. 1k).

5. Figure 2b. Why is there so much Mediator signal in the cytoplasm? The Mediator immunostainings are very poor (also those in Fig. S7), and hardly conclusive if at all, considering broad cytoplasmic localisation.

The MED1 antibody is known to cross-react with a cytoplasmic protein, and the cytoplasmic artefactual staining for Mediator antibodies is known in the literature (Boija et al., 2018; Guo et al., 2019; Sabari et al., 2018). We validated the nuclear signal of the MED1 antibody being MED1 using a degradation sensitive MED1-FKBP KBM7 cell line (see reviewer Figure 1 below). Also, some of the IF signal outside highlighted nuclear regions comes from neighboring ESCs, as mESCs are known to stick together and forming colonies under the serum+LIF conditions we culture them.

6. The interpretation of the authors that ERV transcription ‘competes’ with transcription of OSN super-enhancers is interesting, but alternative interpretations may be plausible. It assumes that RNA PolIII is limiting, but it is unclear if the phenotype is due to a shortage of RNAPII or of Mediator? Can they rescue the effect on super-enhancers by overexpressing RNAPII or MED23?

In the revised manuscript, we directly tested the competition between ERVs and super-enhancers. We generated mESC lines harboring multiple copies of inducible IAPEz or MMERVK10C ERV fragments, and show that forced transcription of these elements leads to a reduction of super-enhancer transcription, while forced transcription of GFP transgenes with the same copy number, insertion sites and same RNA-transcript length does not (new Fig. 4k-m, Extended Data Fig. 17). These data provide important evidence for the specificity of IAPEz and MMERVK10C ERV transcripts on super-enhancer transcription in cells.

The suggested rescue experiment by overexpressing RNAPII or MED23 are intriguing; however, RNAPII and Mediator are large multiprotein complexes consisting of >20 subunits, so it’s unlikely that overexpressing individual subunits would have a large effect.

7. The in vitro droplet work is not entirely convincing; it seems that any RNA will have an effect on droplet formation, as the authors do point out. But there seems then not to be a negative control – what about using an of an ERV that does not increase their expression after Trim28 KD (fig 3d-I and fig S13)?

We believe that the reviewer is indeed correct that any RNA species are likely to facilitate phase separation of transcriptional activators in vitro (Henninger et al., 2021), though this has not been shown for ERV RNA before. Our in vitro data nevertheless lend support to the important function of ERV RNAs. One new insight in our in vitro data is demonstrating that the concentration optimum of ERV RNA is different for MED1 and HP1 α (Fig 4h). This finding could explain why heterochromatin is transcribed at low levels.

The negative control in the experiments noted by the reviewer are conditions with no RNA added to the proteins, and these data are included in the figures (new Fig 4e-j, Extended Data Fig. 16).

8. Along the same lines, the figure for the NFYC-IDR incorporation in MED1 droplets (Fig 3h-i) has no negative control.

The negative control and additional control experiments for the NFYC in vitro experiments are in Extended Data Fig. 15f-i. We show that NFYC-mEGFP is incorporated into MED1 IDR droplets, whereas equimolar amount of mEGFP is not (Extended Data Fig. 15h-i).

9. The conclusions on ‘condensates’ (and PS) in cells are relatively weak, the only real experiment which aims to (partly) address whether the puncta are actually condensates is based on 1,6-HD treatment. While this is routinely used in the field, in isolation it is not a strong indication that PS or condensates form in vivo, and in addition, 1,6-HD can just be inhibiting transcription in general. The authors should perform additional experiments in cells to strengthen their manuscript. Otherwise, the term ‘condensates’ should not be used. For example, they could test the role of NFYC IDR in cells by deleting the IDR from NFYC and investigate if Trim28 depletion leads to upregulation of IAP or Cthrc1 transcripts.

In the revised manuscript we include live cell super-resolution imaging with PALM (in collaboration with the Cisse lab who are leading experts on this technology) to address condensate-RNA properties in TRIM28 degraded cells in live cells. In brief, we generated an mESC cell line in which nascent transcripts are visualized at the *Sox2* super-enhancer locus, RNAPII molecules are visualized with the photoconvertible dendra2 tag, and TRIM28 alleles are tagged with FKBP. 24h dTAG-treatment in these cells led to a significant reduction in size of the RNAPII cluster nearest to the *Sox2* locus (new Fig. 1k), and an increase in the distance between the *Sox2* locus and the nearest RNAPII cluster (new Fig. 1k), while the size of RNAPII clusters in the cells and average number of RNAPII clusters per cell did not change (new Fig. 1k). These data indicate reduced association of RNAPII condensates at the *Sox2* super-enhancer locus upon acute TRIM28-degradation in live cells, detected at super-resolution. These results provide new insights into condensate features in live TRIM28-degraded cells, and further support that ERV de-repression alters condensate dynamics at super-enhancer loci.

10. The pattern of GAG expression in blastocyst in Fig 5b is inconsistent with what is known in the literature: GAG is known to localise throughout the cytoplasmic of embryonic cells, which has been shown both by EM but also by similar immunostaining experiments (see for example Fig. 6d in PMID 25457166 work by A. Surani). This raises some doubts on the validity of the quantifications and conclusions drawn from these experiments. As with the Fig. 5d (see below), this data do not add much to the manuscript, as it is only correlation.

The images of IAP-GAG staining in embryos are higher resolution than those in PMID: 25457166 (Kim et al., 2014). The staining was validated in 2C-4C stage wild type embryos where IAP GAG is expressed

(See Reviewer Figure 2 below). Fig. 3C from another study demonstrates IAP-GAG staining with large distinct puncta and much smaller puncta (Fasching et al., 2015), indicating a range of de-repression, and this staining pattern is similar to staining pattern our data. IAP-GAG levels may be modulated in wild type 2C-4C embryos, knockout embryos, somatic or pluripotent cell types, thereby reflecting the number of foci or the intensity of expression.

Because the quantification was performed using consistent thresholds across all KOs, we believe the IAP GAG staining results to be robust.

Reviewer Figure 2. Additional validation of IAP GAG staining in wild type early mouse embryos.

IAP GAG protein is stained green. Nuclei are in magenta. Shown are three embryos, one consisting of 3 cells (3C), one of four cells (4C) and one of 16 cells (16C). IAP GAG is known to be present around the 4C stage, and IAPs are silenced at around the 16C stage.

11. I find Fig. 5d rather misleading. The authors associate temporal phenotype to time of derepression of ERVs in chromatin mutants. However, there are many causes why embryos may die that are unrelated with ERV expression. Without experiments that firmly link onset of expression of ERVs DIRECTLY to a lethality phenotype, this data is misleading and should be removed.

We removed Fig. 5d from the revised manuscript.

12. The results in Fig. 5g are surprising, since *Trim28* KO does not have a penetrant phenotype in terms of *Nanog*/*Oct4* expression, which persists in the KO (Genes Dev 2017, Kumar et al). How do the authors explain these differences? Please discuss.

While zygotic *TRIM28* perturbation results in arrest at the gastrulation stage, maternal *TRIM28* perturbation results in early male-specific lethality (E4.5). Female embryos exhibit a range of phenotypes with a broad lethality window extending up to the P0 stage (Cammass et al., 2000; Messerschmidt et al., 2012; Sampath Kumar et al., 2017).

Trim28 knockdown in (Sampath Kumar et al., 2017) has no effect on *Oct4*/*Nanog* expression because only maternal *Trim28* is depleted (oocyte specific KO), while zygotic *Trim28* expression is unaffected (Zygotic *Trim28* gets expressed at the 4C stage).

13. In their model (lines 250-256), the authors suggest that derepression of ‘super-enhancers’ may explain the lethality observed in early embryos. However, the authors really have no data to make such a claim both, because i) the mutants they analyse have additional, multiple target genes which could all explain the embryonic phenotype and ii) we do not really even know if super-enhancers are actually working/existing/functional in embryos.

We agree with the reviewer that our analyses do not rule out that effects of *TRIM28*-degradation other than ERV de-repression contributes to cellular phenotypes. Regardless, the knockdown data suggest that ERV knockdown can at least on the short term rescue the effect of *TRIM28*-degradation, supporting the model that indeed the extent of de-repression matters (Fig 4a-d, Extended Data Fig. 14). Therefore, we wrote that ERV de-repression *may* explain lethality observed in embryos.

Minor comments:

Fig. 1h – please add n number for each box plot.

We added the numbers in the legends for Fig. 1h.

Fig. 1d – are the TRIM28 levels in the TRIM28-dTAG cell line comparable to the parental, wt cell line?

The level of TRIM28 is slightly lower in the TRIM28-dTAG cell line compared to the parental, wt cell line (new Extended Data Fig. 2m). As detailed above at Major Comment 6 of Reviewer 1, we include a series of controls that demonstrate similarity of the TRIM28-FKBP tagged mESC line to the parental V6.5 line in terms of protein levels of pluripotency marker genes (new Extended Data Fig. 2j-m), differentiation potential (new Extended Data Fig. 2d-i) and transcriptome profile (new Extended Data Fig. 2n).

Many ERVs do not change their expression after Trim28 KD (Fig. S4b and e). Please tone down/adapt conclusions accordingly to reflect specificity of ERVs in a more accurate way.

We adjusted the conclusions at the results (line 111), and Discussion.

Fig. 2i – the contact analysis is pretty nice. However, I think it would be most appropriate to do the same analysis but using only DE genes (as opposed to all genes), and compare those with non-DE genes.

We included additional contact and compartment analyses in Extended Data Fig. 11c-d. DE and non-DE genes are included in the compartment analysis in revised Extended Data Fig. 11c. For contact frequency analysis, we found that the contact frequencies were too sparse to make meaningful comparisons, as there are very few DE genes in our data.

The RNA FISH signals do not seem to be consistent across experiments. For example, Fig 2b shows quite a high signal of IAP FISH in the cytoplasm. Also, the pattern of the signal is very different between

There is indeed some heterogeneity in the IAP RNA FISH signal, and several factors contribute to heterogeneity. For example, the IAP transcript is unprocessed (does not contain introns); therefore, the FISH probes detect every IAP transcript, as opposed to e.g. the *Cthrc1* intronic RNA FISH probes that label the nascent pre-processed transcript enriched at the site of transcription. The IAP RNA gets exported from the nucleus, so some cytoplasmic staining is expected. This is consistent with the IAP FISH signal being more enriched in the nucleus after 24h of dTAG treatment than after 48h dTAG treatment (Extended Data Fig. 7a). Furthermore, mESCs stick together in colonies, and tend to contain relatively little cytoplasm, so some ‘extranuclear’ FISH signal is coming from nuclei of adjacent cells.

The differences in OCT4 levels, as measured by western after overexpression (Fig S14.b) do not really correlate with the expected changes of IAP (by RNA FISH) levels (Fig. 4c-d). Why is so?

It appears that the rescue of IAP de-repression by overexpression of the OCT4, SOX2, KLF4, MYC (OSKM) transcription factors requires some time, and may be transient. The correlation seems apparent at 48h of OSKM induction. After 48h, TRIM28-degraded cells have reduced OCT4 expression, and about 90% cells have visible IAP foci (Fig. 4c-d). At the same timepoint, TRIM28-degraded cells that overexpress OSKM factors have ‘normal’ level of OCT4, and only 50% of cells have apparent IAP foci (Fig. 4c-d).

HP1 and MED1 graphs in Fig. S13b are repeated/duplicated from Fig. 3g. This is understandable, however the authors should make this explicit in the figure legend to avoid potential future issues of duplication.

We now include the information that the panels are duplicated in the corresponding figure legends.

The phenotype of Trim28 KO has already been described, and is largely in line with that presented in Fig. S17a. While the current manuscript clearly presents a higher level of characterisation, it would be fair to incorporate at least a citation to previous work by Messerschmidt et al.

We now cite Messerschmidt et al in the context of describing TRIM28 KO phenotype (line 300).

The conclusion in lines 213-214 is overstated: the analysis of TEs that the authors done is rather limited to IAP and thus indicating that this is a general phenomenon, especially considering that the authors did not study at all additional/any specific subfamily members, is overstated. Also, the competition experiments that are the basis for this conclusion are based on a very artificial system (GFP overexpression).

We included several analyses of additional ERV taxa, including IAPs. For example, we included analysis of IAPs, MMERVK10Cs, MMERVK9Cs, MMETNs in embryos (new Fig. 5e), we tested MMERVK10C and IAPEz fragments in the competition experiment (new Extended Data Fig. 17), and included IAPEz, MMERVK10C, MMERVK9C and MMETNs in the knockdown experiments (Fig. 4a-d, Extended Data Fig. 14e-g).

Why did the Piggybac experiment use GFP and not IAP sequence? (Fig. 4h-i)?

In the revised manuscript we generated mESC lines harboring multiple copies of inducible IAPEz or MMERVK10C ERV fragments, and show that forced transcription of these elements leads to a reduction of super-enhancer transcription, while forced transcription of GFP transgenes with the same copy number, insertion sites and same RNA-transcript length does not. (new Fig. 4k-m, Extended Data Fig. 17). These data provide important evidence for the specificity of IAPEz and MMERVK10C ERV transcripts on super-enhancer transcription in cells. We removed the the Piggybac experiments that only used GFP.

Does GFP RNA copy number correlate with droplet formation in in-vitro assays?

The copy number of ERV fragments force-transcribed in cells correlates with the downregulation of super-enhancer transcription (Fig. 4k-m, Extended Data Fig. 17). In the in vitro system, the concentration of RNA correlates with the partitioning of RNAPII CTD, and MED1 IDR into droplets up to ~100nM (Fig. 4e-f, Extended Data Fig. 15a-e, 16).

Figs S5a and b are missing labels.

We added the missing labels to Extended Data Fig. 5a-b.

N numbers are missing throughout all the (cellular) microscopy. Please amend.

We included the numbers for the cellular microcopy experiments in the figures (Fig. 1j, 2a-b, 2f-g, 3f, 4d, Extended Data Fig. 2j, 6a, 6d, 9d, 14j, 18c). For the embryo microscopy data, the number of embryos is included in the figure legends (Fig. 5c, 5f, Extended Data Fig. 20a-b).

References

- Andergassen, D., Smith, Z.D., Kretzmer, H., Rinn, J.L., and Meissner, A. (2021). Diverse epigenetic mechanisms maintain parental imprints within the embryonic and extraembryonic lineages. *Developmental cell* *56*, 2995-3005 e2994.
- Boehning, M., Dugast-Darzacq, C., Rankovic, M., Hansen, A.S., Yu, T., Marie-Nelly, H., McSwiggen, D.T., Kokic, G., Dailey, G.M., Cramer, P., *et al.* (2018). RNA polymerase II clustering through carboxy-terminal domain phase separation. *Nature structural & molecular biology* *25*, 833-840.
- Boija, A., Klein, I.A., Sabari, B.R., Dall'Agnese, A., Coffey, E.L., Zamudio, A.V., Li, C.H., Shrinivas, K., Manteiga, J.C., Hannett, N.M., *et al.* (2018). Transcription Factors Activate Genes through the Phase-Separation Capacity of Their Activation Domains. *Cell* *175*, 1842-1855 e1816.
- Cammas, F., Mark, M., Dolle, P., Dierich, A., Chambon, P., and Losson, R. (2000). Mice lacking the transcriptional corepressor TIF1beta are defective in early postimplantation development. *Development* *127*, 2955-2963.
- Cho, W.K., Spille, J.H., Hecht, M., Lee, C., Li, C., Grube, V., and Cisse, II (2018). Mediator and RNA polymerase II clusters associate in transcription-dependent condensates. *Science*.
- Fasching, L., Kapopoulou, A., Sachdeva, R., Petri, R., Jonsson, M.E., Manne, C., Turelli, P., Jern, P., Cammas, F., Trono, D., *et al.* (2015). TRIM28 represses transcription of endogenous retroviruses in neural progenitor cells. *Cell reports* *10*, 20-28.
- Grosswendt, S., Kretzmer, H., Smith, Z.D., Kumar, A.S., Hetzel, S., Wittler, L., Klages, S., Timmermann, B., Mukherji, S., and Meissner, A. (2020). Epigenetic regulator function through mouse gastrulation. *Nature* *584*, 102-108.
- Guo, Y.E., Manteiga, J.C., Henninger, J.E., Sabari, B.R., Dall'Agnese, A., Hannett, N.M., Spille, J.H., Afeyan, L.K., Zamudio, A.V., Shrinivas, K., *et al.* (2019). Pol II phosphorylation regulates a switch between transcriptional and splicing condensates. *Nature* *572*, 543-548.
- He, J., Fu, X., Zhang, M., He, F., Li, W., Abdul, M.M., Zhou, J., Sun, L., Chang, C., Li, Y., *et al.* (2019). Transposable elements are regulated by context-specific patterns of chromatin marks in mouse embryonic stem cells. *Nature communications* *10*, 34.
- Henninger, J.E., Oksuz, O., Shrinivas, K., Sagi, I., LeRoy, G., Zheng, M.M., Andrews, J.O., Zamudio, A.V., Lazaris, C., Hannett, N.M., *et al.* (2021). RNA-Mediated Feedback Control of Transcriptional Condensates. *Cell* *184*, 207-225 e224.
- Kim, S., Gunesdogan, U., Zyllich, J.J., Hackett, J.A., Cougot, D., Bao, S., Lee, C., Dietmann, S., Allen, G.E., Sengupta, R., *et al.* (2014). PRMT5 protects genomic integrity during global DNA demethylation in primordial germ cells and preimplantation embryos. *Molecular cell* *56*, 564-579.
- Messerschmidt, D.M., de Vries, W., Ito, M., Solter, D., Ferguson-Smith, A., and Knowles, B.B. (2012). Trim28 is required for epigenetic stability during mouse oocyte to embryo transition. *Science* *335*, 1499-1502.
- Rowe, H.M., Jakobsson, J., Mesnard, D., Rougemont, J., Reynard, S., Aktas, T., Maillard, P.V., Layard-Liesching, H., Verp, S., Marquis, J., *et al.* (2010). KAP1 controls endogenous retroviruses in embryonic stem cells. *Nature* *463*, 237-240.
- Rowe, H.M., Kapopoulou, A., Corsinotti, A., Fasching, L., Macfarlan, T.S., Tarabay, Y., Viville, S., Jakobsson, J., Pfaff, S.L., and Trono, D. (2013). TRIM28 repression of retrotransposon-based enhancers is necessary to preserve transcriptional dynamics in embryonic stem cells. *Genome research* *23*, 452-461.

Sabari, B.R., Dall'Agnesse, A., Boija, A., Klein, I.A., Coffey, E.L., Shrinivas, K., Abraham, B.J., Hannett, N.M., Zamudio, A.V., Manteiga, J.C., *et al.* (2018). Coactivator condensation at super-enhancers links phase separation and gene control. *Science*.

Sampath Kumar, A., Seah, M.K., Ling, K.Y., Wang, Y., Tan, J.H., Nitsch, S., Lim, S.L., Lorthongpanich, C., Wollmann, H., Low, D.H., *et al.* (2017). Loss of maternal Trim28 causes male-predominant early embryonic lethality. *Genes & development* *31*, 12-17.

Smith, Z.D., Shi, J., Gu, H., Donaghey, J., Clement, K., Cacchiarelli, D., Gnirke, A., Michor, F., and Meissner, A. (2017). Epigenetic restriction of extraembryonic lineages mirrors the somatic transition to cancer. *Nature* *549*, 543-547.

Thompson, P.J., Macfarlan, T.S., and Lorincz, M.C. (2016). Long Terminal Repeats: From Parasitic Elements to Building Blocks of the Transcriptional Regulatory Repertoire. *Molecular cell* *62*, 766-776.

Wells, J.N., and Feschotte, C. (2020). A Field Guide to Eukaryotic Transposable Elements. *Annual review of genetics* *54*, null.

Whyte, W.A., Orlando, D.A., Hnisz, D., Abraham, B.J., Lin, C.Y., Kagey, M.H., Rahl, P.B., Lee, T.I., and Young, R.A. (2013). Master transcription factors and mediator establish super-enhancers at key cell identity genes. *Cell* *153*, 307-319.

Zamudio, A.V., Dall'Agnesse, A., Henninger, J.E., Manteiga, J.C., Afeyan, L.K., Hannett, N.M., Coffey, E.L., Li, C.H., Oksuz, O., Sabari, B.R., *et al.* (2019). Mediator Condensates Localize Signaling Factors to Key Cell Identity Genes. *Molecular cell*.

Decision Letter, first revision:

25th Jan 2022

Dear Denes,

Your Article, entitled "Hijacking of transcriptional condensates by endogenous retroviruses", has now been seen by 3 of the 4 original referees. Unfortunately, reviewer #3 was unable to comment on the revised manuscript. You will see from their comments below that while they find your work improved overall, some important points are raised that must be thoroughly addressed. We are interested in the possibility of publishing your study in Nature Genetics, but would like to consider your response to these concerns in the form of a revised manuscript before we make a final decision on publication.

Reviewer #1 thinks that the paper has improved substantially but is asking you to provide some missing controls.

Reviewer #2 also says that this is a major improvement but thinks some additional clarification(s) should be added to the text. They have suggested some experiments that they believe would provide key mechanistic insight.

Reviewer #4 acknowledges that the paper is much better but they are not completely satisfied. They highlight some points that were not well addressed and, like Reviewer #2, notes that the concern about RNA specificity is still unresolved, which significantly limits the potential impact of the paper.

We invite you to revise your manuscript taking into account all reviewer comments. Please highlight all changes in the manuscript text file. At this stage we will need you to upload a copy of the manuscript in MS Word .docx or similar editable format.

We are committed to providing a fair and constructive peer-review process. Do not hesitate to contact me if there are specific requests from the reviewers that you believe are technically impossible or unlikely to yield a meaningful outcome.

*2) If you have not done so already please begin to revise your manuscript so that it conforms to our Article format instructions, available

http://www.nature.com/ng/authors/article_types/index.html>here.

*3) Include a revised version of any required Reporting Summary:

It will be available to referees (and, potentially, statisticians) to aid in their evaluation if the

manuscript goes back for peer review.
A revised checklist is essential for re-review of the paper.

[REDACTED]

We hope to receive your revised manuscript within 3-6 months. If you cannot send it within this time, please let us know.

Sincerely,

Tiago

Tiago Faial, PhD
Senior Editor
Nature Genetics
<https://orcid.org/0000-0003-0864-1200>

Reviewers' Comments:

Reviewer #1:

Remarks to the Author:

The revised manuscript addressed most of the raised criticisms by adopting advanced approaches that permitted to substantially improve the manuscript. Indeed, the new set of data are supportive of the drawn conclusions.

Still, some minor controls should be included to further improve the robustness of the presented data. For example, the authors should include the transcriptome profile of the ERV knockdown cells lacking TRIM28 and the unperturbed cells, not referring only to the transcript level of SE (Fig. 4c) but providing also the same comparison for coding genes.

Similarly, they should include some controls regarding the IAPEz (and GFP) lines, by analyzing the potential changes in the expression level of other genes that are regulated by canonical enhancers. A similar control was requested for the IF-RNA FISH experiments upon TRIM28 degradation, but the authors did not provide it in the revised manuscript. They claim that such control was already shown in a previous publication (Sabari et al. 2018); although relevant, this does not support the SE-specific effect in response to TRIM28 degradation, that was not the topic of the mentioned work.

For what concerns the PALM experiments (Fig. 1k), the authors demonstrated that upon TRIM28 degradation there is a concomitant increase of distance between SOX2 SE and the nearest RNAP clusters, whose size was decreased, in agreement with the finding that enhancers increase the frequency of transcription bursting that correlates with RNAP cluster levels. It would be an added value if the authors would include representative images of the proximity between the labelled locus and the RNAP.

In line with this observation it would be convenient to proceed with similar measurements (cluster distances) on the fixed samples (Fig. 1j and ED Fig. 6a-d). Indeed, in this set of experiments the authors presented quantitative measurements of RNAP intensities in the vicinity of the investigated loci, as a proxy of RNAP cluster size (yet not commented/mentioned in the manuscript). Including these quantifications may strengthen the obtained results.

Reviewer #2:

Remarks to the Author:

The authors have sufficiently addressed my comments on their initial manuscript and have provided a much-improved version. I will limit my comments here to new data included in this revised manuscript:

1) Major improvement: extended the analysis of the effects of Trim28 depletion on more than one super enhancer. As prompted by reviewer #3, they have now included analyses of Klf4, Fgf4, and the Sox2 super-enhancers in addition to their favorite Mir290-295 locus. The addition of these new loci alleviate concerns that the effects they observed after Trim28 depletion were restricted to the Mir290-295 locus.

2) Major improvement: they have included an orthogonal assay with live-cell super-resolution (!) imaging with the Sox2 super-enhancer that complements the other assays (RNA-seq, RT-qPCR, fixed cell microscopy) included in the original submission. The authors should be commended for adding such an impressive new technology into the manuscript during the revision stage.

3) Major improvement: The authors have improved upon the experimental design of the GFP transgene expression effect on super-enhancers. In the initial submission, they used a GFP transgene to mimic the titration of the transcriptional machinery away from SE loci. Reviewer #3 criticized this experimental design, and although I thought it was poorly explained and contextualized in the original submission, in my opinion it was sufficient as a proof-of-principle experiment. This experiment has been improved upon by a technical trick to compare isogenic GFP transgene cell lines and IAPez line and their SE activity after doxycycline induction (Fig 4k) and the authors report a surprising finding.

The experiment described in Fig 4k, l, m is an incredibly important addition to the revised manuscript, but it still needs some explanation. In the response to reviewers, the authors state: "...we wish to stress that our model is not specific to IAPs. Rather, it is the amount of the locally produced RNA which 'traps' the transcriptional machinery through the RNA's ability to facilitate phase separation of transcriptional activators". This is a plausible explanation, since the isogenic cells (GFP vs. floxed-GFP+IAPez) have the same copy number of transgenes, the same genomic locations, and the same number of Tet-O operator sequences that should attract a similar amount of RNA polymerase and co-activators—thus leaving the specific RNA product produced from the transgene (IAPez/MMERVK10C) as the specific molecule responsible for mediating the transcriptional condensate highjacking. An important point that may escape the non-TE expert crowd: presumably the effect seen in Fig 4k-m is not due to IAPez protein products since the effect can be generated using a 900-bp MMERVK10C fragment (Line 278-279) but clarifying this in the text would help to lead the reader to focus on the ERV RNA and not an ERV-encoded protein as the biological relevant molecule.

Furthermore, in the discussion, (line 335-338) they state "In cells, forced transcription of ERV RNA transcripts from multiple loci led to a profound decrease in super-enhancer transcription while transcription of GFP RNA had a moderate effect, suggesting sequence contribution to the impact of RNAs on condensates".

When I asked in the initial round of review "Is the claim that IAP/MMERVK10C ERV RNA has a special sequence or structural features that mechanistically explain the difference in droplet formation in vitro [biophysical assays]?" the authors replied in the response to reviewers "we believe that any RNA species are likely to facilitate phase separation of transcriptional activators in vitro".

Based on these statements, it is unclear how the authors mechanistically explain why IAP/MMERVK10C RNA preferentially compared to GFP mRNA hijacks transcriptional condensates (Fig 4k-m). It is possible to simply just further clarify this in the text. The authors do speculate in the discussion (line 340-341) that perhaps ERV RNA modifications (m6A?) may provide specificity or enhanced ability to sequester transcriptional condensates (an exciting possibility). Of course m6A may affect these biophysical properties in cells due to m6A readers, but would not be captured in the in vitro condensate assays (Fig 4e-j).

It is also possible to perform additional experiments to more completely flesh-out the mechanism through which ERV RNA preferentially highjacks transcriptional condensates. I loathe to suggest more experiments for an already outstanding and broadly rigorous manuscript, but the lack of a rock-solid molecular mechanism explaining the phenomena in which ERV RNA can preferentially highjack transcriptional condensates is, I suspect, not an ancillary issue but at the heart of the impact of this paper.

For instance, the authors can perform similar experiments to Fig 4k-m utilizing transgenes based on chimeras between IAPez/MMERVK10C and non-highjacking ERVS, deletion series of the

IAPez/MMERVK10C transgene to narrow down a critical region of the mRNA sequence, or re-coding the IAPez/MMERVK10C sequence to remove or substitute adenosine bases to test if an m6A modification potentially facilitates the effect seen in Fig 4k-m.

In summary, the authors have improved upon an already compelling manuscript. I still think it will be of great interest to the Nature Genetics readership and will reorient the way we think about TE/ERV reactivation and its impact on development.

Reviewer #4:

Remarks to the Author:

The authors have performed a significant amount of work to address some of my comments.

In general, I still think this is an interesting manuscript. However, the PALM imaging approach they took to address my PS concerns does not address them, in spite of the high quality and technical difficulty to perform the imaging experiments with the Cissé lab, the questions I raised still remain unanswered. The RNA specificity concern I had is also not fully answered, as the question still remains where does the specificity come from for the IAP RNA but I understand that this is a difficult question and the efforts that the authors have done towards addressing it (and toning their conclusions down) are important additions to the manuscript.

Overall both the PALM and the piggyBac with the IAP are good additions to the paper (especially the piggyBac, which is much stronger evidence than the GFP that they had initially) but these experiments do not really answer the major comments I had on the first round.

In short, while I in principle dislike to suggest an additional revision round to address the open PS questions, I remain of the opinion that the authors should address them before publication. As it is, the (PS) part continues not to add much to the manuscript, nor to the mechanistic investigation of the authors' observations and the quality of this data is in general less good.

Below I comment on the answers to my specific points:

Point 1. The authors did not really address the antibody issue that I raised, however, the combined detection of endogenous RNAPolII (Rpb1 ?) and the cross-reactivity to both hypo and CTD-phosphorylated RNA Pol II is sufficient for their conclusions.

Point 2. The authors calculated Manders coef only in fig 2, but not throughout the rest of the manuscript. Why were the quantifications not performed? This was a very easy request to address, and given the not so good quality of the images (also noted by the other Reviewers), I had suggested quantifications throughout to improve the robustness of the data.

Points 3, 4 and 5. Addressed.

Point 6 is a very elegant result and strong point of the revision, even if the reason behind the specificity for IAP remains unexplained.

Point 7. The authors acknowledged the point I raised in my initial review, but did not perform an

experiment with the RNA from another ERV that does not become derepressed in Trim28 KD conditions. As such, as I initially pointed, this in vitro data does not support a specific involvement of the ERV RNA to their model at all.

Point 8. Addressed.

Point 9. The authors went through a great deal to perform PALM experiments, but unfortunately these experiments do not address the point I had raised originally. I raised a very specific suggestion for NFYC, and the PALM experiment, though strong, only gives again a very general behaviour, which is not really providing support for their thesis that PS, through ERV RNA is involved in enhancer regulation upon Trim28 Knock-down.

Point 10. I still think that the images are not optimal for quantifications, but I can accept that the authors have raised good arguments to control for the stainings.

Point 11. Addressed.

Point 12. The authors should make an effort to discuss these phenotypes. They addressed my point in their rebuttal, but the previous work on Trim28 has not been incorporated in the discussion. This is a minor point now, but this gives fair credit to previous work and puts theirs in the right context.

Point 13. The authors tone down their conclusions, so it is addressed.

Comments on the new data:

a. Line 116 and 117: is it not the other way around? (250 upregulated and 300 downregulated genes upon TRIN28 degradation)

b. The authors show that the optimal concentration of IAP RNA for HP1a to form condensates is lower. This would imply that, in theory IAP should promote more HP1a hijacking than RNAPII. This in vitro data thus do not truly support their conclusions, and this is why I have been critical on the usefulness of such in vitro studies.

Author Rebuttal, first revision:

NG-A58215R2, Hijacking of transcriptional condensates by endogenous retroviruses

Detailed response (blue font) to the referees' comments (black font).

Reviewer #1:

Remarks to the Author:

The revised manuscript addressed most of the raised criticisms by adopting advanced approaches that permitted to substantially improve the manuscript. Indeed, the new set of data are supportive of the drawn conclusions.

Still, some minor controls should be included to further improve the robustness of the presented data. For example, the authors should include the transcriptome profile of the ERV knockdown cells lacking TRIM28 and the unperturbed cells, not referring only to the transcript level of SE (Fig. 4c) but providing also the same comparison for coding genes. Similarly, they should include some controls regarding the IAPEz (and GFP) lines, by analyzing the potential changes in the expression level of other genes that are regulated by canonical enhancers. A similar control was requested for the IF-RNA FISH experiments upon TRIM28 degradation, but the authors did not provide it in the revised manuscript. They claim that such control was already shown in a previous publication (Sabari et al. 2018); although relevant, this does not support the SE-specific effect in response to TRIM28 degradation, that was not the topic of the mentioned work.

We include the requested controls in the revised manuscript. We performed further analysis of the transcriptome profile of ERV knockdown cells. The analyses revealed that knockdown of ERVs globally rescues the reduction in transcription of super-enhancer -associated coding genes in TRIM28-degraded cells (Fig. 4c). Furthermore, we performed qRT-PCR analysis of a panel of seven typical enhancer -associated genes in the IAPEz (and GFP) -expressing PiggyBac lines. We found that the RNA levels of typical enhancer associated genes are largely unaffected by the induction of the IAPEz/GFP transgenes (Extended Data Fig. 17e). These results provide further evidence that transcription of super-enhancers and their associated genes are particularly sensitive to the expression level of IAPs, supporting our model that super-enhancers compete with ERVs for transcriptional condensates.

For what concerns the PALM experiments (Fig. 1k), the authors demonstrated that upon TRIM28 degradation there is a concomitant increase of distance between SOX2 SE and the nearest RNAP clusters, whose size was decreased, in agreement with the finding that enhancers increase the frequency of transcription bursting that correlates with RNAP cluster levels. It would be an added value if the authors would include representative images of the proximity between the labelled locus and the RNAP. In line with this observation it would be convenient to proceed with similar measurements (cluster distances) on the fixed samples (Fig. 1j and ED Fig. 6a-d). Indeed, in this set of experiments the authors presented quantitative measurements of RNAP intensities in the vicinity of the investigated loci, as a proxy of RNAP cluster size (yet not commented/mentioned in the manuscript). Including these quantifications may strengthen the obtained results.

We have quantified the distance between the *miR290-295* super-enhancer locus and the nearest RNAPII puncta in the fixed cell RNA FISH-IF data in control and TRIM28-degraded cells. We found that the average distance between the locus and the nearest RNAPII puncta was slightly higher in TRIM28-degraded cells, consistent with the PALM experiments (Extended Data Fig 6a, right panel). We note that such a measurement is more informative using the live cell PALM experiments, because in the PALM we track condensates for two minutes, whereas the fixed cell FISH-IF experiments provide a snapshot of the SE locus. Finally, we agree with the reviewer's assessment that further analysis (and images) of the PALM data would be useful. Precise measurements of both distance and dynamics of condensates as well as gene expression in real-time at the locus will be part of a follow up paper by our collaborator.

Reviewer #2:

Remarks to the Author:

The authors have sufficiently addressed my comments on their initial manuscript and have provided a much-improved version. I will limit my comments here to new data included in this revised manuscript:

1) Major improvement: extended the analysis of the effects of Trim28 depletion on more than one super enhancer. As prompted by reviewer #3, they have now included analyses of Klf4, Fgf4, and the Sox2 super-enhancers in addition to their favorite Mir290-295 locus. The addition of these new loci alleviate concerns that the effects they observed after Trim28 depletion were restricted to the Mir290-295 locus.

2) Major improvement: they have included an orthogonal assay with live-cell super-resolution (!) imaging with the Sox2 super-enhancer that complements the other assays (RNA-seq, RT-qPCR, fixed cell microscopy) included in the original submission. The authors should be commended for adding such an impressive new technology into the manuscript during the revision stage.

3) Major improvement: The authors have improved upon the experimental design of the GFP transgene expression affect on super-enhancers. In the initial submission, they used a GFP transgene to mimic the titration of the transcriptional machinery away from SE loci. Reviewer #3 criticized this experimental design, and although I thought it was poorly explained and contextualized in the original submission, in my opinion it was sufficient as a proof-of-principle experiment. This experiment has been improved upon by a technical trick to compare isogenic GFP transgene cell lines and IAPez line and their SE activity after doxycycline induction (Fig 4k) and the authors report a surprising finding.

The experiment described in Fig 4k, l, m is an incredibly important addition to the revised manuscript, but it still needs some explanation. In the response to reviewers, the authors state: "...we wish to stress that our model is not specific to IAPs. Rather, it is the amount of the locally produced RNA which 'traps' the transcriptional machinery through the RNA's ability to facilitate phase separation of transcriptional activators". This is a plausible explanation, since the isogenic cells (GFP vs. floxed-GFP+IAPez) have the same copy number of transgenes, the same

genomic locations, and the same number of Tet-O operator sequences that should attract a similar amount of RNA polymerase and co-activators—thus leaving the specific RNA product produced from the transgene (IAPez/MMERVK10C) as the specific molecule responsible for mediating the transcriptional condensate highjacking. An important point that may escape the non-TE expert crowd: presumably the effect seen in Fig 4k-m is not due to IAPez protein products since the effect can be generated using a 900-bp MMERVK10C fragment (Line 278-279) but clarifying this in the text would help to lead the reader to focus on the ERV RNA and not an ERV-encoded protein as the biological relevant molecule.

We provide further experimental evidence supporting specific contributions of ERV RNA in embryonic stem cells expressing *GFP* or *IAPez* RNA from integrated multi-copy PiggyBac transposons. We performed cell fractionation in these cells, and quantified the amount of nuclear and cytoplasmic RNA. The experiments revealed that a substantially larger proportion of *IAPez* RNA is retained in the nucleus compared to *GFP* RNA (Extended Data Fig 17e). The results reinforce that the contribution of ERV RNA to transcriptional phenotypes has a sequence specific component in vivo. We clarified this in the discussion (lines 336-345).

Furthermore, in the discussion, (line 335-338) they state “In cells, forced transcription of ERV RNA transcripts from multiple loci led to a profound decrease in super-enhancer transcription while transcription of GFP RNA had a moderate effect, suggesting sequence contribution to the impact of RNAs on condensates”.

When I asked in the initial round of review “Is the claim that IAP/MMERVK10C ERV RNA has a special sequence or structural features that mechanistically explain the difference in droplet formation in vitro [biophysical assays]?” the authors replied in the response to reviewers “we believe that any RNA species are likely to facilitate phase separation of transcriptional activators in vitro”. Based on these statements, it is unclear how the authors mechanistically explain why IAP/MMERVK10C RNA preferentially compared to GFP mRNA hijacks transcriptional condensates (Fig 4k-m). It is possible to simply just further clarify this in the text. The authors do speculate in the discussion (line 340-341) that perhaps ERV RNA modifications (m6A?) may provide specificity or enhanced ability to sequester transcriptional condensates (an exciting possibility). Of course m6A may affect these biophysical properties in cells due to m6A readers, but would not be captured in the in vitro condensate assays (Fig 4e-j).

We agree with the interpretation of the reviewer, and discuss the RNA results and model with further clarification in the discussion (lines 336-345). The PiggyBac experiments – including the newly added cell fractionation data in Extended Data Fig 17f – provide key evidence that there is a sequence specific contribution of ERV RNA in cells, which is not present in vitro. Sequence specificity could be caused by RNA modifications or RNA binding proteins in vivo. Discovering the mechanistic basis of sequence specificity will be an important focus of future studies.

It is also possible to perform additional experiments to more completely flesh-out the mechanism through which ERV RNA preferentially highjacks transcriptional condensates. I loathe to suggest more experiments for an already outstanding and broadly rigorous manuscript, but the lack of a rock-solid molecular mechanism explaining the phenomena in which ERV RNA can

preferentially hijack transcriptional condensates is, I suspect, not an ancillary issue but at the heart of the impact of this paper.

As explained above, we provide further experimental evidence supporting specific contributions of ERV RNA in cells. We agree with the reviewer that discovering the mechanistic basis of sequence specificity will be an important problem for future studies.

For instance, the authors can perform similar experiments to Fig 4k-m utilizing transgenes based on chimeras between IAPez/MMERVK10C and non-highjacking ERVS, deletion series of the IAPez/MMERVK10C transgene to narrow down a critical region of the mRNA sequence, or re-coding the IAPez/MMERVK10C sequence to remove or substitute adenosine bases to test if an m6A modification potentially facilitates the effect seen in Fig 4k-m.

We thank the reviewer for the suggestion. This is an interesting and useful experiment for future studies, focusing on understanding the mechanistic basis of sequence specificity of ERV RNAs.

In summary, the authors have improved upon an already compelling manuscript. I still think it will be of great interest to the Nature Genetics readership and will reorient the way we think about TE/ERV reactivation and its impact on development.

Reviewer #4:

Remarks to the Author:

The authors have performed a significant amount of work to address some of my comments.

In general, I still think this is an interesting manuscript. However, the PALM imaging approach they took to address my PS concerns does not address them, in spite of the high quality and technical difficulty to perform the imaging experiments with the Cissé lab, the questions I raised still remain unanswered. The RNA specificity concern I had is also not fully answered, as the question still remains where does the specificity come from for the IAP RNA but I understand that this is a difficult question and the efforts that the authors have done towards addressing it (and toning their conclusions down) are important additions to the manuscript.

Overall both the PALM and the piggyBac with the IAP are good additions to the paper (especially the piggyBac, which is much stronger evidence than the GFP that they had initially) but these experiments do not really answer the major comments I had on the first round.

In short, while I in principle dislike to suggest an additional revision round to address the open PS questions, I remain of the opinion that the authors should address them before publication. As it is, the (PS) part continues not to add much to the manuscript, nor to the mechanistic investigation of the authors' observations and the quality of this data is in general less good.

We thank the reviewer for further pressing on the important issue on RNA specificity. We provide further experimental evidence supporting specific contributions of ERV RNA in

embryonic stem cells expressing *GFP* or *IAPez* RNA from integrated multi-copy PiggyBac transposons. We performed cell fractionation in these cells, and quantified the amount of nuclear and cytoplasmic RNA. The experiments revealed that a substantially larger proportion of *IAPez* RNA is retained in the nucleus compared to *GFP* RNA (Extended Data Fig 17e). The results reinforce that the contribution of ERV RNA to transcriptional phenotypes has a sequence specific component in vivo.

Below I comment on the answers to my specific points:

Point 1. The authors did not really address the antibody issue that I raised, however, the combined detection of endogenous RNAPolIII (Rpb1 ?) and the cross-reactivity to both hypo and CTD-phosphorylated RNA Pol II is sufficient for their conclusions.

RPBI is the common name of the *POLR2A* gene encoding the catalytic subunit of RNA Polymerase II.

Point 2. The authors calculated Manders coef only in fig 2, but not throughout the rest of the manuscript. Why were the quantifications not performed? This was a very easy request to address, and given the not so good quality of the images (also noted by the other Reviewers), I had suggested quantifications throughout to improve the robustness of the data.

Quantifications were performed for all RNA FISH-IF experiments throughout the manuscript. In accordance with standard considerations (Dunn et al., 2011), co-localization for single-copy loci was quantified using a Spearman's correlation coefficient calculated on the RNA FISH signal and IF signal, centered on the center of RNA FISH focus. The Spearman's correlation coefficients are displayed in the Figures (Fig. 1j, 2f, 2g, 3f, 4d, Extended Data Fig. 6b, 6d, 9d, 14j). The Manders' overlap co-efficient was used to quantify co-localization of the multi-copy IAP foci as recommended (Dunn et al., 2011), and are included in Fig. 2a-b. We also calculated Manders' coefficients for the single-copy locus data, and in every instance, we found qualitative agreement between the Spearman's and Manders' coefficients, and therefore, simply kept the Spearman's coefficients in the cited figures. Also, we note that we replaced all images with high resolution images throughout the manuscript.

Points 3, 4 and 5. Addressed.

Point 6 is a very elegant result and strong point of the revision, even if the reason behind the specificity for IAP remains unexplained.

We thank the reviewer for the comment. We agree that discovering the mechanistic basis of sequence specificity will be an important problem for future studies.

Point 7. The authors acknowledged the point I raised in my initial review, but did not perform an experiment with the RNA from another ERV that does not become derepressed in Trim28 KD conditions. As such, as I initially pointed, this in vitro data does not support a specific involvement of the ERV RNA to their model at all.

The PiggyBac experiments – including the newly added cell fractionation data in Extended Data Fig 17f – provide key evidence that there is a sequence specific contribution of ERV RNA in cells, which is not present in vitro. Further understanding of the mechanistic basis of RNA specificity will facilitate development of better in vitro models.

Point 8. Addressed.

Point 9. The authors went through a great deal to perform PALM experiments, but unfortunately these experiments do not address the point I had raised originally. I raised a very specific suggestion for NFYC, and the PALM experiment, though strong, only gives again a very general behaviour, which is not really providing support for their thesis that PS, through ERV RNA is involved in enhancer regulation upon Trim28 Knock-down.

We thank the reviewer for the suggestion. This is an interesting and useful experiment for future studies dissecting the contribution of various regulatory proteins to the association of condensates at transcribed ERVs.

Point 10. I still think that the images are not optimal for quantifications, but I can accept that the authors have raised good arguments to control for the stainings.

Point 11. Addressed.

Point 12. The authors should make an effort to discuss these phenotypes. They addressed my point in their rebuttal, but the previous work on Trim28 has not been incorporated in the discussion. This is a minor point now, but this gives fair credit to previous work and puts theirs in the right context.

We cite the key reports in question (Messerschmidt et al., 2012; Sampath Kumar et al., 2017). We believe that explaining the difference between the phenotypes of maternal TRIM28 and zygotic TRIM28 is not the most central for our manuscript investigating the roles of ERV RNA; therefore kept the discussion on the details of TRIM28 to a minimum, and focused on RNA.

Point 13. The authors tone down their conclusions, so it is addressed.

Comments on the new data:

a. Line 116 and 117: is it not the other way around? (250 upregulated and 300 downregulated genes upon TRIN28 degradation)

Indeed, the numbers were switched. We fixed them in the text.

b. The authors show that the optimal concentration of IAP RNA for HP1a to form condensates is lower. This would imply that, in theory IAP should promote more HP1a hijacking than RNAPII.

This in vitro data thus do not truly support their conclusions, and this is why I have been critical on the usefulness of such in vitro studies.

An interesting insight emerging from the HP1 α in vitro data is that the concentration at which RNA facilitates condensate dissolution is lower for HP1 α than for the MED1 IDR. This finding could explain why heterochromatic regions (e.g. ERVs) tend to be transcribed at a basal level. We thus argue that the data is useful, and suggests interesting models for future studies.

References

Dunn, K.W., Kamocka, M.M., and McDonald, J.H. (2011). A practical guide to evaluating colocalization in biological microscopy. *Am J Physiol Cell Physiol* 300, C723-742.

Messerschmidt, D.M., de Vries, W., Ito, M., Solter, D., Ferguson-Smith, A., and Knowles, B.B. (2012). Trim28 is required for epigenetic stability during mouse oocyte to embryo transition. *Science* 335, 1499-1502.

Sampath Kumar, A., Seah, M.K., Ling, K.Y., Wang, Y., Tan, J.H., Nitsch, S., Lim, S.L., Lorthongpanich, C., Wollmann, H., Low, D.H., *et al.* (2017). Loss of maternal Trim28 causes male-predominant early embryonic lethality. *Genes & development* 31, 12-17.

Decision Letter, second revision:

Our ref: NG-A58214R1

16th Mar 2022

Dear Denes,

Thank you for submitting your revised manuscript "Hijacking of transcriptional condensates by endogenous retroviruses" (NG-A58214R1). It has now been seen by the original referees and their comments are below.

Reviewer #1 is fully satisfied and is supportive of publication. The reviewer notes that there are still some open questions but feels those are beyond the scope of the paper.

Reviewer #2 thinks that the added data do not directly address the specificity question that reviewer #4 and she/he raised. The reviewer is therefore disappointed but admits that this might require a lot more time/effort and has offered some constructive suggestions for future work. Despite this, the reviewer still thinks that this is an important paper.

Reviewer #4 says that you have not adequately addressed their points. The reviewer is disappointed by your rebuttal, much like reviewer #2. However, they agree that the manuscript is important and interesting. As a minimum, this reviewer thinks that you need to comprehensively tone down any claims regarding specificity and better highlight the caveats of your data interpretation in the Discussion.

After a careful discussion within the editorial team, we've decided that we'll be happy in principle to publish your manuscript in Nature Genetics, pending a textual revision to satisfy the referees' final requests and to comply with our editorial and formatting guidelines.

Since the current version of your manuscript is in a PDF format, please email us (natgen@us.nature.com) a copy of the file in an editable format (Microsoft Word) - we cannot proceed with PDFs at this stage.

We will then be performing detailed checks on your paper and will send you a checklist detailing our editorial and formatting requirements. Please do not upload the final materials and make any revisions until you receive this additional information from us.

Thank you again for your interest in Nature Genetics. Please do not hesitate to contact me if you have any questions.

Congratulations!

Sincerely,

Tiago

Tiago Faial, PhD
Senior Editor
Nature Genetics
<https://orcid.org/0000-0003-0864-1200>

Reviewer #1 (Remarks to the Author):

The authors addressed the raised points/comments, and I am satisfied with the revised version of the manuscript, which is ready for publication. Indeed, although there are still some important open questions, I agree that further dedicated studies would be required to address them, and these experiments go beyond the goal of this manuscript.

Reviewer #2 (Remarks to the Author):

My major criticism from revision round #1 was that the mechanistic basis of the preferential SE-interference by ERV RNA vs. control GFP-RNA was not sufficiently investigated and that the specificity of this effect remained unclear.

In revision round #2, the authors have added data showing that IAP-EZ ERV RNA has a higher % of retention in the nuclear fraction compared to the control GFP RNA molecules (Sup. Fig. 17f). Since the authors have already shown that RNA can mediate PS effects in vitro, the model that they put forth hinges on the ERV RNA being preferentially retained in the nucleus (compared to control GFP RNA). These additional data do not comprehensively answer the question of molecular mechanism and specificity through which ERV RNA leads to SE interference through transcriptional condensate highjacking per se, but the authors (quite understandably) realize that this is probably another paper. I feel like they were **so** close to the home-run molecular mechanism, and I still feel that such an addition would fundamentally close the knowledge-gap that I want this paper to do!

On the other hand, it is possible that their highly useful piggy-back ERV transgene system (Fig. 4) has been pushed to the limit of its utility, as some co-transcriptional features of ERV RNA (m6A for example) may only be partially recapitulated in this system and endogenous ERV RNA transcribed in situ may be impacted or modified by their local chromatin environment. For future studies, the authors may want to focus on whether the ERV RNA retained in the nucleus is actually transcriptionally terminated or whether it is still tethered to the RNA polymerase machinery and is actually a nascent transcript. The presence of a poly-A tail could help them distinguish this. Additionally, the inclusion of mRNA export sequences (such as HIV Rev-responsive elements + Rev protein expression or an analogous HERVK-Rec responsive elements) into their piggy-back transgenes may aid in their dissection of whether the ERV RNA has a nascent transcript functionality (RNA export signal immune) or can act in trans after transcriptional termination/nuclear retention (RNA export signal responsive).

Reviewer #4 (Remarks to the Author):

The authors have performed additional experiments and/or text clarifications to address the remaining comments. They disregarded most of my comments though. There are substantial claims on specificity of RNA regulatory impact, which are not supported by the data presented. All my comments have been meant to help the authors strengthen their manuscript, not to destroy it, and thus I was quite disappointed by their responses. The cellular fractionation, while supportive for IAP enrichment in the nucleus, cannot formally rule out that other RNAs play a role in their phenotypic characterisation. The authors should at the very least acknowledge themselves the limitations of their analyses and, for the sake of a constructive and balanced scientific contribution, incorporate those by a discussion in their manuscript. I remain of the opinion that the manuscript is important and interesting, but a balanced discussion without overstatements would only help the authors to avoid causing misinterpretations in the field.

Author Rebuttal, second revision:

General response to reviewers

We thank the referees for their valuable comments and suggestions throughout the revision process. We are pleased that the reviewers unanimously appreciate the importance of our findings that RNA transcripts produced by endogenous retroviruses play essential roles in the distribution of nuclear condensates and cell identity in pluripotent cells. We also appreciate the sentiment of the reviewers that the RNA-based condensate hijacking model clearly inspires further interesting questions and opens up new directions of future investigations. In particular the question of the source of RNA specificity will be most interesting, but contend that such investigations be done in future studies.

Reviewer #1 (Remarks to the Author):

The authors addressed the raised points/comments, and I am satisfied with the revised version of the manuscript, which is ready for publication. Indeed, although there are still some important open questions, I agree that further dedicated studies would be required to address them, and these experiments go beyond the goal of this manuscript.

Reviewer #2 (Remarks to the Author):

My major criticism from revision round #1 was that the mechanistic basis of the preferential SE-

interference by ERV RNA vs. control GFP-RNA was not sufficiently investigated and that the specificity of this effect remained unclear.

In revision round #2, the authors have added data showing that IAP-EZ ERV RNA has a higher % of retention in the nuclear fraction compared to the control GFP RNA molecules (Sup. Fig. 17f). Since the authors have already shown that RNA can mediate PS effects in vitro, the model that they put forth hinges on the ERV RNA being preferentially retained in the nucleus (compared to control GFP RNA). These additional data do not comprehensively answer the question of molecular mechanism and specificity through which ERV RNA leads to SE interference through transcriptional condensate hijacking per se, but the authors (quite understandably) realize that this is probably another paper. I feel like they were *so* close to the home-run molecular mechanism, and I still feel that such an addition would fundamentally close the knowledge-gap that I want this paper to do!

On the other hand, it is possible that their highly useful piggy-back ERV transgene system (Fig. 4) has been pushed to the limit of its utility, as some co-transcriptional features of ERV RNA (m6A for example) may only be partially recapitulated in this system and endogenous ERV RNA transcribed in situ may be impacted or modified by their local chromatin environment. For future studies, the authors may want to focus on whether the ERV RNA retained in the nucleus is actually transcriptionally terminated or whether it is still tethered to the RNA polymerase machinery and is actually a nascent transcript. The presence of a poly-A tail could help them distinguish this. Additionally, the inclusion of mRNA export sequences (such as HIV Rev-responsive elements + Rev protein expression or an analogous HERVK-Rec responsive elements) into their piggy-back transgenes may aid in their dissection of whether the ERV RNA has a nascent transcript functionality (RNA export signal immune) or can act in trans after transcriptional termination/nuclear retention (RNA export signal responsive).

We thank the reviewer for the enthusiasm and suggestions of useful future experiments! Indeed using the PiggyBac system to dissect the basis of RNA specificity, and looking at whether the RNAs are tethered to RNAPII are great future experiments.

Reviewer #4 (Remarks to the Author):

The authors have performed additional experiments and/or text clarifications to address the remaining comments. They disregarded most of my comments though. There are substantial claims on specificity of RNA regulatory impact, which are not supported by the data presented. All my comments have been meant to help the authors strengthen their manuscript, not to destroy it, and thus I was quite disappointed by their responses. The cellular fractionation, while supportive for IAP enrichment in the nucleus, cannot formally rule out that other RNAs play a

role in their phenotypic characterisation. The authors should at the very least acknowledge themselves the limitations of their analyses and, for the sake of a constructive and balanced scientific contribution, incorporate those by a discussion in their manuscript. I remain of the opinion that the manuscript is important and interesting, but a balanced discussion without overstatements would only help the authors to avoid causing misinterpretations in the field.

We definitely did not disregard comments, and do appreciate the helpful guidance to improve the manuscript! Our data on the PiggyBac system clearly suggest that there is some specificity of ERV RNA in cells. Indeed, forced expression of *IAPez* and *MMERVK10c* fragments reduced super-enhancer transcription, while forced expression of the same copy number of *GFP* from the same genomic loci did not. The fact is that we do not know what the molecular basis of this specificity is, and we have seen no evidence of RNA specificity in the in vitro reconstitution system, which likely because we simply assay three components. We made this very clear in the discussion.

Final Decision Letter:

In reply please quote: NG-A58214R2 Hnisz

26th May 2022

Dear Denes,

I am delighted to say that your manuscript "Hijacking of transcriptional condensates by endogenous retroviruses" has been accepted for publication in an upcoming issue of Nature Genetics.

Your paper will be published online after we receive your corrections and will appear in print in the

next available issue. You can find out your date of online publication by contacting the Nature Press Office (press@nature.com) after sending your e-proof corrections. Now is the time to inform your Public Relations or Press Office about your paper, as they might be interested in promoting its publication. This will allow them time to prepare an accurate and satisfactory press release. Include your manuscript tracking number (NG-A58214R2) and the name of the journal, which they will need when they contact our Press Office.

Please note that *Nature Genetics* is a Transformative Journal (TJ). Authors may publish their research with us through the traditional subscription access route or make their paper immediately open access through payment of an article-processing charge (APC). Authors will not be required to make a final decision about access to their article until it has been accepted. [Find out more about Transformative Journals](https://www.springernature.com/gp/open-research/transformative-journals)

Authors may need to take specific actions to achieve [compliance with funder and institutional open access mandates](https://www.springernature.com/gp/open-research/funding/policy-compliance-faqs). If your research is supported by a funder that requires immediate open access (e.g. according to [Plan S principles](https://www.springernature.com/gp/open-research/plan-s-compliance)) then you should select the gold OA route, and we will direct you to the compliant route where possible. For authors selecting the subscription publication route, the journal's standard licensing terms will need to be accepted, including [self-archiving-and-license-to-publish](https://www.nature.com/nature-portfolio/editorial-policies/self-archiving-and-license-to-publish). Those licensing terms will supersede any other terms that the author or any third party may assert apply to any version of the manuscript.

Please note that Nature Portfolio offers an immediate open access option only for papers that were first submitted after 1 January, 2021.

To assist our authors in disseminating their research to the broader community, our SharedIt initiative provides you with a unique shareable link that will allow anyone (with or without a subscription) to

read the published article. Recipients of the link with a subscription will also be able to download and print the PDF.

An online order form for reprints of your paper is available at https://www.nature.com/reprints/author-reprints.html. Please let your coauthors and your institutions' public affairs office know that they are also welcome to order reprints by this method.

Sincerely,

Tiago

Tiago Faial, PhD
Senior Editor
Nature Genetics
<https://orcid.org/0000-0003-0864-1200>